# Degradation of complex arabinoxylans by human colonic Bacteroidetes

Gabriel V. Pereira [1,2,3], Ahmed M. Abdel-Hamid[1,4], Soumajit Dutta[5], Corina N. D'Alessandro-Gabazza[1,6], Daniel Wefers [1,3], Jacob A. Farris[1,3,7], Shiv Bajaj[1,7], Zdzislaw Wawrzak[8], Haruyuki Atomi [9], Roderick I. Mackie[1,2,3,10], Esteban C. Gabazza[1,6], Diwakar Shukla[5], Nicole M. Koropatkin[1,11] & Isaac Cann [1,2,3,7,9,10,12,13 ✉]

Some Bacteroidetes and other human colonic bacteria can degrade arabinoxylans, common polysaccharides found in dietary fiber. Previous work has identified gene clusters (polysaccharide-utilization loci, PULs) for degradation of simple arabinoxylans. However, the degradation of complex arabinoxylans (containing side chains such as ferulic acid, a phenolic compound) is poorly understood. Here, we identify a PUL that encodes multiple esterases for degradation of complex arabinoxylans in *Bacteroides* species. The PUL is specifically upregulated in the presence of complex arabinoxylans. We characterize some of the esterases biochemically and structurally, and show that they release ferulic acid from complex arabinoxylans. Growth of four different colonic Bacteroidetes members, including *Bacteroides intestinalis*, on complex arabinoxylans results in accumulation of ferulic acid, a compound known to have antioxidative and immunomodulatory properties.

[1] Carl R. Woese Institute for Genomic Biology (Microbiome Metabolic Engineering Theme), University of Illinois at Urbana-Champaign, Urbana, IL 61801, USA. [2] Department of Animal Science, University of Illinois at Urbana-Champaign, Urbana, IL 61801, USA. [3] Energy Biosciences Institute, University of Illinois at Urbana-Champaign, Urbana, IL 61801, USA. [4] Department of Botany and Microbiology, Faculty of Science, Minia University, 61519 El-Minia, Egypt. [5] Department of Chemical and Biomolecular Engineering, University of Illinois at Urbana-Champaign, Urbana, IL 61801, USA. [6] Department of Immunology, Mie University, Tsu City, Mie 514, Japan. [7] School of Molecular and Cellular Biology, University of Illinois at Urbana-Champaign, Urbana, IL 61874, USA. [8] Argonne National Laboratory, 9700 S. Cass Avenue, Argonne, IL 60439, USA. [9] Top Global University Program, Department of Synthetic Chemistry and Biological Chemistry, Graduate School of Engineering, Kyoto University, Katsura, Nishikyo-ku, Kyoto, Japan. [10] Division of Nutritional Sciences, University of Illinois at Urbana-Champaign, Urbana, IL 61801, USA. [11] Department of Microbiology and Immunology, University of Michigan. Medical School, Ann Arbor, MI 48109, USA. [12] Center for East Asian & Pacific Studies, University of Illinois at Urbana-Champaign, Urbana, IL 61801, USA. [13] Department of Microbiology, University of Illinois at Urbana-Champaign, Urbana, IL 61801, USA. ✉email: icann@illinois.edu

The capacity of a microbial community to colonize an ecosystem is intimately linked to nutrient availability and acquisition in that environment. The human gut microbiome reflects this concept, as its members are able to acquire carbon and energy from host gastrointestinal tract (GIT)-derived glycans and the dietary components undegradable by the host[1–4]. Many of these nutrient sources, however, exist as polymeric structures and require enzymatic depolymerization to release their metabolizable components. In humans, a microbial consortium comprised mostly of two bacterial phyla, the Bacteroidetes and Firmicutes, has evolved to degrade complex polysaccharides in the lower GIT[5–8]. The process of polysaccharide degradation has been best studied in the Bacteroidetes.[5] Furthermore, the Bacteroidetes are known to harness a unique polysaccharide degradation strategy, where the genes encoding the proteins that sense, degrade, and transport nutrients are co-localized in clusters known as polysaccharide-utilization loci (PULs)[3]. Unique among the genes in the PULs are the *susC/susD* gene pairs that encode membrane-localized transporters and the genes encoding the Hybrid Two-Component System (HTCS)[6,9].

The HTCS is multi-modular polypeptides predicted to sense nutrient availability to regulate the expression of the genes in their associated PULs[10,11]. Biochemical, genetic, and structural analyses have been used to demonstrate that the sensor module of a *Bacteroides thetaiotaomicron* HTCS binds specifically to monomeric fructose to activate a fructan degradation PUL[12]. Furthermore, Martens et al. reported that larger forms of the sensor module from diverse HTCS (i.e., ~700–900 amino acid-polypeptides versus ~300 amino acid-polypeptides in the fructose HTCS sensor) in *B. thetaiotamicron* and *Bacteroides ovatus* recognize oligosaccharides derived from polysaccharides uniquely targeted by a cognate PUL for degradation[3]. The crystal structure of the sensor module of an HTCS together with its Y_Y_Y domain has provided the first look at a homodimeric structure of a Bacteroidetes HTCS and therefore allowed the proposal of a mechanism for signal transduction in this class of sensor regulators[13].

In orchestrating the initial enzymatic attack on polysaccharides in the GIT, members of the Bacteroidetes and the Firmicutes act as primary polysaccharide degraders, making nutrients available for their own metabolic processes and also to cross-feed the colonic microbiome members that lack the requisite hydrolytic enzymes[4,14–19]. In support of this observation, strains of the colonic bacterium *Faecalibacterium prausnitzii* are known to require the fermentation end product acetate for growth, and strains ATCC 27766 and L2–6 failed to grow in the absence of this short-chain fatty acid[20]. Furthermore, *Methanobrevibacter* strains isolated from humans were shown to grow strictly on carbon dioxide and hydrogen, both end products of fermentation of polysaccharide degraders, such as *Bacteroides intestinalis*[21]. Aside from cross-feeding other microbes[22,23], a large portion of microbial fermentation end products are absorbed by the host for various metabolic processes[24].

Previous reports show that the colonic Bacteroidetes have the capacity to degrade arabinoxylan, a structurally-heterogeneous polysaccharide rich in foods such as maize, oat, rye, and wheat[4,16,25–27]. In addition, degradation of arabinoxylan by some Bacteroidetes, including *B. intestinalis* and *B. cellulosilyticus*, suggests that they function as primary polysaccharide degraders in the colon[25,28,29]. Importantly, simple or soluble arabinoxylan (i.e., without ferulic acid side chains) induced expression of two unique PULs in *B. intestinalis* without upregulating a PUL containing several predicted esterases that likely enhance degradation of complex arabinoxylans (i.e., with ferulic acid side chains) by this colonic bacterium[25]. This observation led us to hypothesize

that depending on the nature of the arabinoxylan, i.e., simple or complex, the Bacteroidetes deploy different PULs for degradation of the polysaccharide.

In this study, we culture diverse members of the human colonic Bacteroidetes on arabinoxylans of different complexities and demonstrate that these bacteria are able to distinguish simple from complex arabinoxylans by deploying different PULs for their degradation. We identify an esterase genes-enriched (EGE) PUL that targets complex arabinoxylan degradation and systematically characterize the enzymes associated with the PUL to delineate the linkages cleaved by the different enzymes. We further determine the three-dimensional structures of side-chain cleaving esterases that likely relieve steric hindrance to enhance depolymerization of the polysaccharide. We then use molecular dynamics simulations to provide insights to the molecular determinants, in the active sites, that yield different substrate specificities in three similarly folded polypeptides. Finally, we demonstrate that the metabolism of complex arabinoxylans by members of the colonic microbiome leads to the accumulation of ferulic acid, a phenolic compound with reported immunomodulatory, antioxidant, and anti-inflammatory activities.

## Results

**An esterase genes-enriched (EGE) PUL in colonic Bacteroidetes.** In an earlier report that examined how *B. intestinalis* degrades simple or soluble wheat arabinoxylan (i.e., arabinoxylan without ferulic acid linkages, Supplementary Fig. 1a), we found that the bacterium upregulates a number of genes, mostly in two PULs containing either one or two carbohydrate esterase (CE) genes[25]. In the present study, we discover that *B. intestinalis* and related colonic bacteria use a putative EGE PUL to extensively degrade complex arabinoxylans. Here, we define an EGE PUL as a PUL with more than two genes encoding either known or putative carbohydrate esterase family proteins. Thus, in the *B. intestinalis* EGE PUL, there are five putative esterases (BACINT_01033-CE1, BACINT_01034-putative esterase, BACINT_01038-CE6/CE1, BACINT_01039-CE1, and BACINT_01040-putative esterase), and in the corresponding PUL in *B. cellulosilyticus* four putative esterases (BACCELL_02154-CE1, BACCELL_02153-putative esterase, BACCELL_02145-CE1, BACCELL_02144-CE1). Only two putative esterases were found in the EGE PUL of *B. oleiciplenus* (HMPREF9447_02532-CE1 and HMPREF9447_02531-putative esterase); however, due to the conservation of the critical glycoside hydrolase (GH) genes in the PUL, it was included in our analyses (Figs. 1a, 2).

To determine whether more complex arabinoxylans (Supplementary Fig. 1b) will trigger the expression of the putative EGE PULs, we grew *B. intestinalis* DSM 17393 and two other colonic *Bacteroides* spp (*B. cellulosilyticus* DSM 14838 and *B. oleiciplenus* YIT 12058), encoding the putative EGE PUL on their genomes (Fig. 1a), on soluble or simple wheat arabinoxylan (sWAX) and three forms of complex arabinoxylan that maintained their side-chain decorations (Supplementary Fig. 1a,b). The complex substrates were feruloylated arabinoxylan oligosaccharides (FAOX), de-starched wheat bran (DWB), and insoluble wheat arabinoxylan (InWAX). While the structure of the DWB is unknown, we assume that it is similar to InWAX (Supplementary Fig. 1b). In addition, we present the structures of the FAOX in Supplementary Figs. 2,3. Using qRT-PCR, we compared the expression of the *susC/sucD* gene pair in the putative EGE PULs during growth on the complex substrates versus a mixture of their constituent sugars (i.e., xylose and arabinose). Growth on the complex substrates led to ~25 – 100-fold increases in the transcript level of the *susC/sucD* gene pair in the EGE PUL compared to growth on the constituent sugars. In contrast, upregulation of transcription of the *susC/susD* gene pair in the

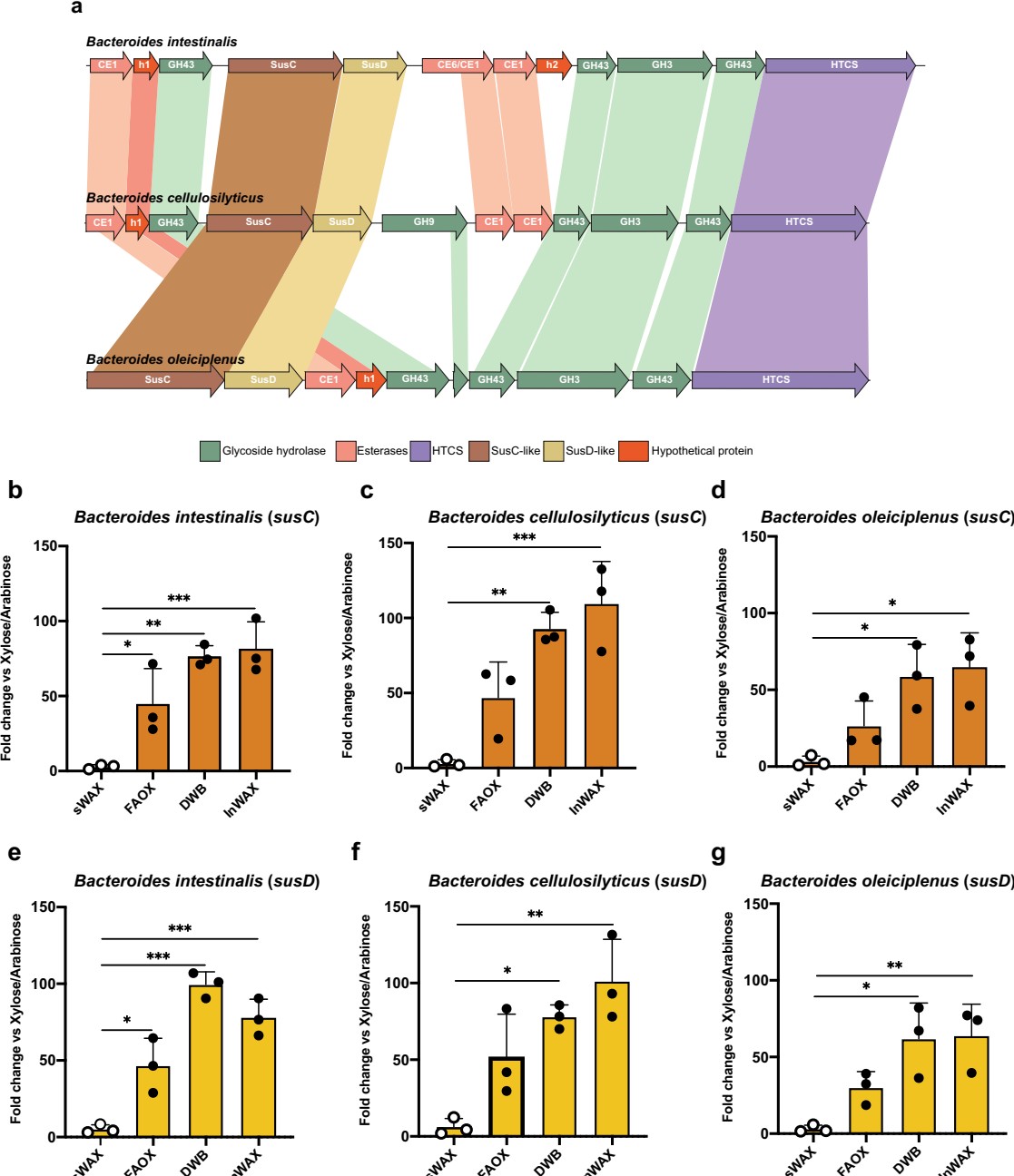

**Fig. 1 The esterase genes-enriched (EGE) PUL, upregulated by feruloylated arabinoxylans, is conserved in different members of the colonic Bacteroidetes. a** A MAUVE alignment of three members of the human colonic microbiome shows high conservation and synteny of an esterase genes-enriched (EGE) polysaccharide-utilization loci (PUL), with all the genes needed for extensive depolymerization of complex arabinoxylans. **b–d** The qRT-PCR of the *susC* gene normalized with the 16S rRNA gene expression, show differential expression of the EGE PULs in *B. intestinalis, B. cellulosilyticus, and B. oleiciplenus* grown on different arabinoxylan substrates compared to growth on a mixture of xylose and arabinose. **e–g** The qRT-PCR of the *susD* gene normalized with the 16S rRNA gene expression, show differential expression of the EGE PULs in *B. intestinalis, B. cellulosilyticus, and B. oleiciplenus* grown on different arabinoxylan substrates compared to growth on a mixture of xylose and arabinose. sWAX: soluble wheat arabinoxylan, FAOX: feruloylated arabinoxylan oligosaccharides, DWB: de-starched wheat arabinoxylan, InWAX: insoluble wheat arabinoxylan, GH: glycoside hydrolase, CE: carbohydrate esterase, HTCS: Hybrid Two-component system, Sus: starch utilization system. h: Hypothetical protein or protein of unknown function. In (**b–g**), the results are presented as means ± standard deviations of biological triplicates ($n = 3$). Statistical analysis was performed using one-way Analysis of Variance (ANOVA) with Tukey's multiple comparisons posttest. *$p < 0.033$, **$p < 0.002$, ***$p < 0.001$. The source data underlying (**b–g**) are provided in the Source Data file.

EGE PUL was barely observed with the growth of the three different bacteria on the simple or soluble arabinoxylan (sWAX). Furthermore, the differences in the expression of the EGE PUL between the simple and complex arabinoxylans were, in general, shown to be statistically significant (Fig. 1b–g).

We carried out whole-genome transcriptional analyses of the three *Bacteroides* spp. grown on complex arabinoxylan to determine if their EGE PUL associated genes will be upregulated, as observed with the *susC/susD* gene pair, during growth on the complex arabinoxylan substrates. All three bacteria upregulated

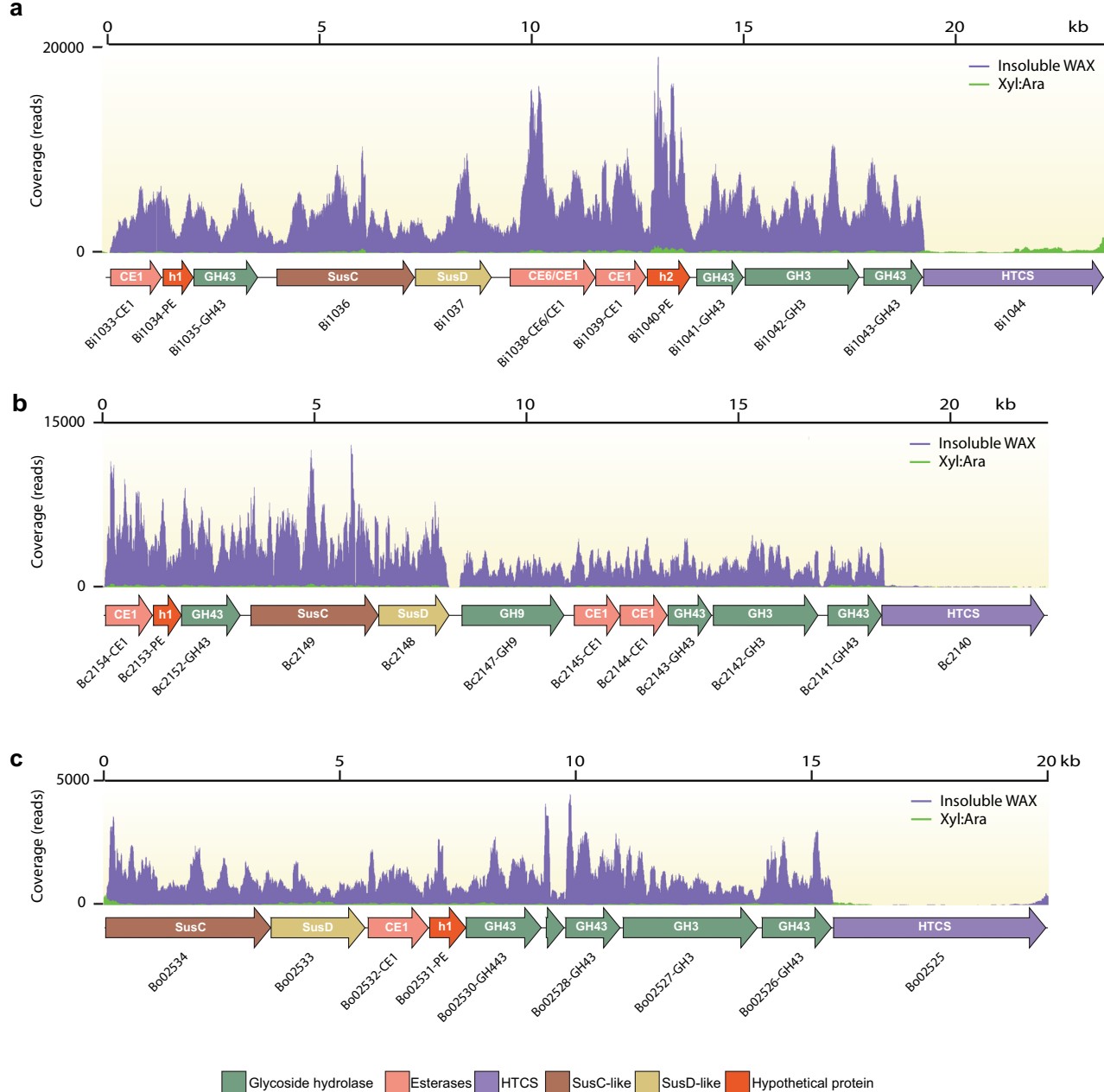

**Fig. 2 Comparative transcriptomics analysis of *Bacteroides* spp grown on insoluble wheat arabinoxylan (InWAX) and a xylose/arabinose mixture (Xyl: Ara).** Transcriptome map showing RNASeq coverage of the esterase genes-enriched (EGE) polysaccharide-utilization loci (PUL) during growth on insoluble wheat arabinoxylan (InWAX, purple) compared to growth on the component sugars (Xyl:Ara, green) by (**a**), *B. intestinalis* (**b**), *B. cellulosilyticus*, and (**c**), *B. oleiciplenus*. The genes in the PULs are color coded based on their predicted functions: glycoside hydrolase (GH, green), carbohydrate esterase (CE, salmon), Hybrid Two-Component System (HTCS, purple), Starch Utilization System proteins (SusC/SusD, brown and yellow) and hypothetical proteins (red). RNAseq experiments involved two biological replicates per species and representative traces are shown for each bacterium in the figure. For the name Sus or <u>S</u>tarch <u>u</u>tilization <u>s</u>ystem, note that although this name originally derives from the first functionally characterized homolog or prototypic system, this name is still maintained for these proteins although it is known that similar systems transport different polysaccharides, including xylan, pectin, mannan, and others.

expression of their respective EGE PUL on the complex substrate compared to the monosaccharide mixture. *Bacteroides intestinalis* showed the highest relative expression of its EGE PUL, followed by *B. cellulosilyticus* and then *B. oleiciplenus* (Fig. 2a,b,c). Other GH and CE genes on the genomes of the three bacteria were also upregulated, although at a far lower level (Supplementary Data 1), suggesting that the EGE PUL is important for complex arabinoxylan degradation. Thus, other enzymes outside of the EGE PUL likely participate in the degradation of the complex

polysaccharide, since the nature of the arabinoxylans encountered in the diet is known to be more intricate than the complex arabinoxylans used in the present study. As an example, arabinoxylans and xylans are known to crosslink with cellulose, another plant cell wall component. This strategy of using PULs together with non-PUL associated enzymes, encoded in other regions of the genome, to degrade complex arabinoxylan is consistent with our findings on soluble arabinoxylan degradation, where two major PULs were highly upregulated, with several

genes, some unassociated with PULs, also upregulated at lower levels[25]. In fact, a gene encoding a GH9 family enzyme, known to be involved in cellulose degradation, is located in the EGE PUL of *B. cellulosilyticus* (Fig. 1a). This gene was upregulated together with the other EGE PUL associated genes during the degradation of complex arabinoxylan (Fig. 2b). In the natural environment, a cellulose targeting enzyme would likely enhance the accessibility of substrate to the arabinoxylan-degrading enzymes to facilitate degradation.

**Bioinformatics analyses of the EGE PUL in *B. intestinalis*.** To determine the role of the individual enzymes encoded in the EGE PUL, the *B. intestinalis* gene cluster, with the highest number of putative esterases, was selected for further analyses. In Fig. 3a, we show putative annotations of the twelve gene products associated with the EGE PUL. There are four glycoside hydrolases (GH), i.e., three GH43 polypeptides (BACINT_01035-GH43, BACINT_01041-GH43, BACINT_01043-GH43) and one GH3 polypeptide (BACINT_01042-GH3) in the cluster. We also identified three putative carbohydrate esterase family 1 (CE1) polypeptides (BACINT_01033-CE1, BACINT_01038-CE6/CE1, and BACINT_01039-CE1), with BACINT_01038-CE6/CE1 further harboring a CE6 module. BACINT_01038-CE6/CE1 was, therefore, designated a bifunctional esterase, either with the same activity or different activities in the two modules. In addition, two polypeptides designated hypothetical proteins (h1 and h2), i.e., BACINT_01034-PE-h1 (or Bi1034-PE-h1) and BACINT_01040-PE-h2 (or Bi1040-PE-h2) were each designated a putative esterase (PE). As observed in canonical PULs, the EGE PUL harbors an HTCS gene (BACINT_01044-HTCS) that is predicted to control expression of the genes in the PUL, and a *susC/susD* gene pair (BACINT_01036 and BACINT_01037) that is predicted to encode a transporter. The GH and CE polypeptides were composed of single modules, except for two GH43 polypeptides that harbor C-terminally appended family 13 (BACINT_01035-GH43) and family 6 (BACINT_01043-GH43) carbohydrate binding modules (CBM), respectively (Fig. 3b).

**The esterases in the EGE PUL target arabinoxylan side chains.** Except for BACINT_01034-h1, recombinant polypeptides of the *B. intestinalis* EGE PUL were successfully produced and purified (Fig. 3a, c). Using a panel of feruloylated oligosaccharides[28], we showed that BACINT_01033-CE1 (or Bi1033-CE1) exhibits low ferulic acid cleavage activity on feruloylated monosaccharide (FA) and about 10 times higher activity on a feruloylated trisaccharide (FAXX) (Fig. 3d). While BACINT_01038-CE6/CE1 (or Bi1038-CE6/CE1) showed some activity on both FA and FAXX, these activities were far lower compared with Bi1033-CE1 (Fig. 3d). We only observed very little activity on the feruloylated oligosaccharides with BACINT_01039-CE1. By contrast, the hypothetical protein BACINT_01040-PE or (Bi1040-PE) showed high cleavage activity on each feruloylated oligosaccharide (Fig. 3d). The structures of the feruloylated oligosaccharides and their hydrolytic traces from Bi1040-PE activity are shown in Supplementary Fig. 2 and Supplementary Fig. 3. Furthermore, Bi1040-PE released ferulic acid from InWAX (Supplementary Fig. 1b and Supplementary Fig. 3d). The foregoing results demonstrated that BACINT_01040-PE (Bi1040-PE) or hypothetical protein 2 (h2 in Fig. 3a) is a ferulic acid esterase, and thus the polypeptide was designated Bi1040-FAE.

Arabinoxylans may harbor side-chain acetyl groups[30,31]. Therefore, we prepared acetylated oatspelt xylan[32,33] and tested the esterases for cleavage of acetyl groups. All four enzymes released acetyl groups from the xylan backbone (Fig. 3e). Bi1038-CE6/CE1 exhibited the highest activity followed by Bi1039-CE1,

i.e., the polypeptides with the least ferulic acid esterase activities. Recombinant Bi1038-CE6/CE1 degraded into two polypeptides (Fig. 3c) during purification, and we hypothesized that the two fragments derive from a proteolytic cleavage in a linker between its two modules (Fig. 3b,c). Expression of the recombinant protein in the presence of a protease inhibitor resulted in purification of the full-length polypeptide (Supplementary Fig. 4) and, therefore, supporting our hypothesis of a proteolytic cleavage.

Enzymatic assays with synthetic substrates, i.e., pNP acetate and methyl-ferulate, also resulted in Bi1038-CE6/CE1 and Bi1039-CE1 exhibiting higher acetyl-xylan esterase activities than Bi1033-CE1 and Bi1040-FAE (Supplementary Fig. 5). Thus, the highest ferulic acid esterase and acetyl-xylan esterase activities on the synthetic substrates were from Bi1040-FAE and Bi1038-CE6/CE1, respectively (Supplementary Fig. 5), further confirming our observation that Bi1040-FAE is a versatile ferulic acid esterase.

**The GH enzymes in the EGE PUL hydrolyze arabinoxylans.** Each purified GH polypeptide was incubated with three different xylan substrates (glucuronoxylan, rye arabinoxylan, and wheat arabinoxylan). The GH43 (Bi1035-GH43) encoded closest to the *susC/susD* gene pair showed far superior hydrolytic activities than the other enzymes (Supplementary Fig. 6a). Furthermore, the products (reducing ends) released from the arabinoxylans were about three times that released from the glucuronoxylan. The Bi1041-GH43 hydrolytic trend was similar to that of Bi1035-GH43; however, the activities were low compared with Bi1035-GH43 (Supplementary Fig. 6a). Bi1042-GH3 and Bi1043-GH43 each released less reducing ends from the polysaccharide substrates compared to Bi1035-GH43 and Bi1041-GH43. Further analyses showed that Bi1042-GH3 cleaves xylo-oligosaccharides to xylose or X1 (Supplementary Fig. 6b), while Bi1043-GH43 cleaves arabinose off InWAX (Supplementary Fig. 6c). Thus, Bi1042-GH3 and Bi1043-GH43 likely function as a β-xylosidase and an α-arabinofuranosidase, respectively, during the degradation of complex arabinoxylans. Importantly, note that Bi1043-GH43 did not release arabinose from mixed linkages with xylose (Supplementary Fig. 6c).

**The EGE PUL associated enzymes depolymerize arabinoxylans.** We investigated whether the GH enzymes work with the esterases to degrade complex arabinoxylans. All esterases were analyzed for their optimum pH and optimum temperatures (Supplementary Fig 7). Different enzyme combinations were then tested for the release of ferulic acid and reducing ends from two different complex arabinoxylans., i.e., de-starched wheat bran (DWB) and insoluble wheat arabinoxylan (InWAX). As individual enzymes, only Bi1035-GH43 and Bi1041-GH43 released appreciable amount of products (Fig. 3f, g). The β-xylosidase Bi1042-GH3 and the α-arabinofuranosidase Bi1043-GH43, in binary reactions with the ferulic acid esterases (Bi1033-CE1 or Bi1040-FAE) released very little reducing ends (Fig. 3f,g), as expected, since Bi1042-GH3 and Bi1043-GH43 are not endo-acting enzymes (Supplementary Fig. 6b, c). In contrast, combinations of the endo-acting enzymes (Bi1035-GH43 and Bi1041-GH43) and the ferulic acid esterases yielded large amounts of both ferulic acid and reducing ends, suggesting that the side-chain decorations impose steric hindrance on the activities of the endo-acting enzymes (Fig. 3f, g). Note that in some of the incubations with individual GH enzymes (Fig. 3f, g), small amounts of ferulic acid, above the control, appeared to be released. These are likely artifacts of the assay, as GH enzymes are not known to harbor esterase activities.

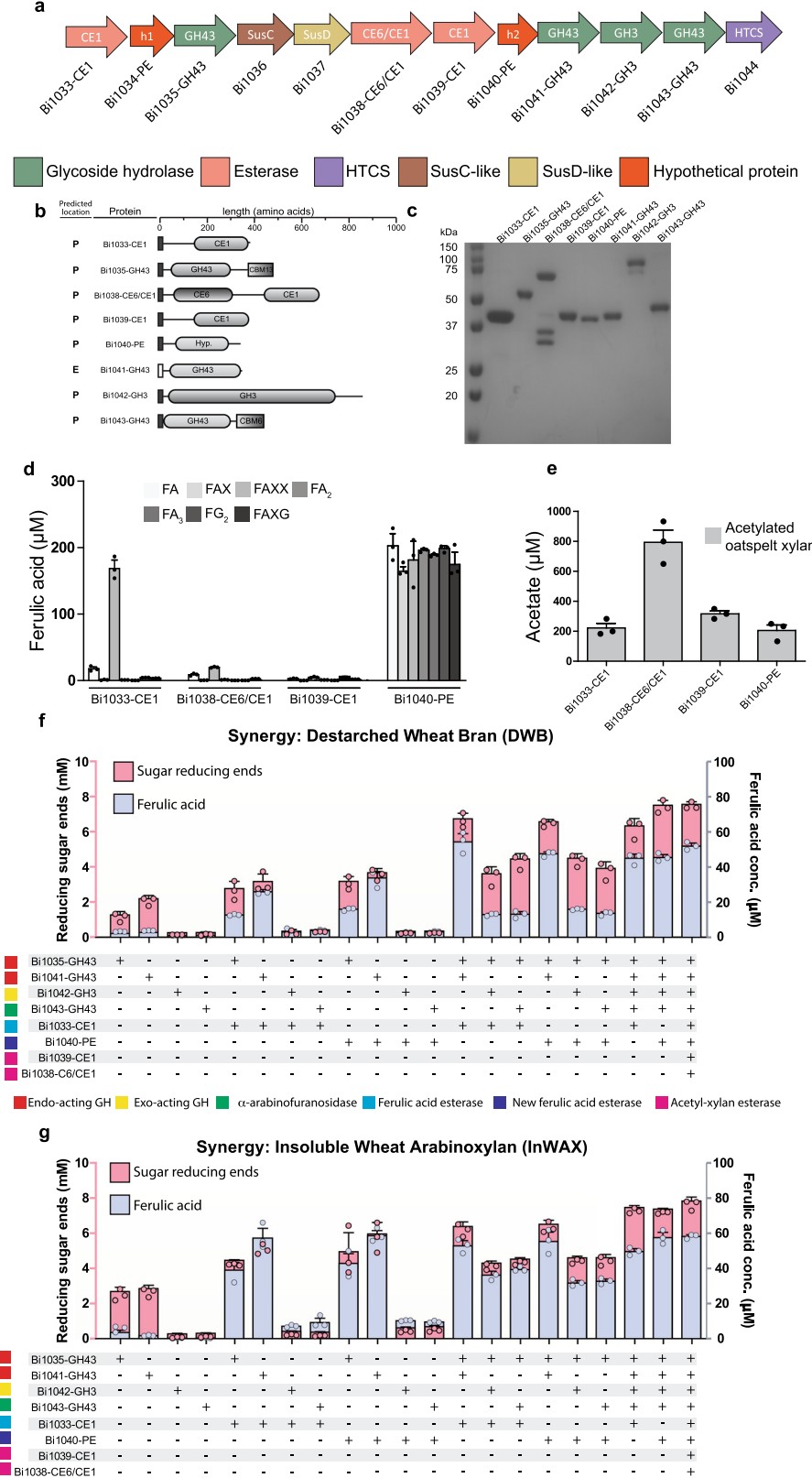

To obtain a clearer view of the functions of the GH enzymes, we subjected the reaction end products to HPAEC-PAD analyses. The contribution of each GH enzyme to the function of the EGE PUL is clearly observed on incubation with the complex arabinoxylan polysaccharides. Bi1035-GH43 and Bi1041-GH43 each released a range of oligosaccharides (from xylobiose to xylohexaose) from the arabinoxylans, and thus confirming their endo-acting activities (Fig. 4a, g). Bi1042-GH3 released the monomeric sugar xylose, likely from free ends in the polysaccharide substrates, as expected of a β-xylosidase (Fig. 4a, g). Furthermore, Bi1043-GH43 released arabinose from each of the two substrates (DWB and InWAX), confirming its

**Fig. 3 The esterase genes enriched (EGE) PUL of *B. intestinalis* encodes esterases with different specificities. a** The genomic context of the EGE PUL is color coded to show the predicted functions of the encoded polypeptides. **b** A schematic representation of the predicted domain architecture of the putative esterases and putative glycoside hydrolases in the EGE PUL. The N-terminally appended dark boxes indicate the predicted signal peptide is Type I (SPI) and therefore a likely periplasmic localization, while the open box indicates predicted signal peptide Type II (SPII) and therefore a predicted outer membrane or extracellular localization. **c** A 12% SDS-PAGE showing the purified (recombinant) putative esterases and glycoside hydrolases. **d** Substrate specificities of the putative esterases in the cluster on feruloylated monosaccharide (FA), feruloylated disaccharides (FAX, FA$_2$, FG$_2$), and feruloylated trisaccharides (FAXX, FAXG, FA$_3$). **e** Acetyl-xylan esterase activity towards acetylated oat spelt xylan. The synergistic activities of the enzymes encoded by the EGE PUL were assessed by incubating the enzymes and their different combinations with: (**f**) de-starched wheat bran and (**g**) insoluble wheat arabinoxylan. The released reducing ends were determined by the *para*-hydroxybenzoic acid hydrazide assay and the released ferulic acid was determined by HPLC-DAD. GH: glycoside hydrolase, CE: carbohydrate esterase, HTCS: Hybrid Two-Component System, and Sus: starch utilization system. In **c**, on purification, each protein was resolved on SDS-PAGE to ensure that they migrated according to their predicted molecular mass, and finally a single SDS-PAGE was ran with all proteins migrating to their individually detected positions relative to the protein molecular mass markers. In **d–g**, the bars are means ± standard deviations of three independent reactions (*n* = 3). The source data underlying (**d–g**) are provided in the Source Data file.

α-arabinofuranosidase activity (Fig. 4a, g). Importantly, the presence of the Bi1042-GH3 β-xylosidase in reactions that also contained one of the endo-acting enzymes (Bi1035-GH43 or Bi1041-GH43) or both resulted in large releases of xylose as the main product, suggesting that the xylo-oligosaccharides released by the endo-acting enzymes are cleaved by the Bi1042-GH3 β-xylosidase to the monomeric sugar (Fig. 4d, e, j, k). In contrast, all reactions containing multiple enzymes, but without Bi1042-GH3, accumulated oligosaccharides, i.e., multiple peaks larger than xylose or X1 (Fig. 4b–e,h–k). The depolymerization of the two complex arabinoxylans is clearly seen where combinations of multiple enzymes that included, at least an endo-acting enzyme (Bi1035-GH43 or Bi1041-GH43), the β-xylosidase (Bi1042-GH3) and the α-arabinofuranosidase (Bi1043-GH43) led to the accumulation of the constituent sugars arabinose and xylose (Fig. 4f,l). These sugars should then be available for fermentation by *B. intestinalis* and therefore other EGE PUL containing Bacteroidetes. Note, however, that although the roles of the esterases are not clearly depicted here, their contributions are easily discerned in the results reported in Fig. 3f, g.

**A homolog of Bi1040-FAE is fused to a GH43 enzyme.** Due to the extensive ferulic acid linkage-cleaving activity of Bi1040-FAE, we had interest in determining the three-dimensional structure of this esterase to gain insight into its enzymatic versatility. However, the expression of Bi1040-FAE was low (Fig. 3c), making crystallization a challenge. Thus, we searched the publicly available databases for homologous proteins, and the best candidate was a larger polypeptide composed of a GH43 family protein fused to a Bi1040-FAE-homologous polypeptide sequence (Supplementary Fig. 8a, b). This polypeptide was present in *Bacteroides eggerthii* (Genbank protein accession number WP_050793236.1) and *Bacteroides stercoris* (Genbank protein accession number WP_117741097.1). The *B. eggerthii* gene was expressed, and the polypeptide initially designated BeGH43/Hyp was purified (Supplementary Fig. 8c) and incubated with different substrates. Arabino-oligosaccharides and xylo-oligosaccharides were not cleaved by BeGH43/Hyp (Supplementary Fig. 8d, e), and the enzyme instead exhibited arabinofuranosidase activity by cleaving arabinose side-chains from InWAX (Supplementary Fig. 8d). Upon further analyses, BeGH43/Hyp exhibited all the activities observed in Bi1040-FAE (Supplementary Figs. 9 and 10), and the enzyme was, therefore, named BeGH43/FAE. To further explore the versatility of the characterized ferulic acid esterases in the present study, we tested the enzymes for their capacity to release ferulic acid from sugar beet pulp (Supplementary Fig. 11), a natural substrate known to contain cellulose, hemicellulose and pectin, with the ferulic acid in more complex linkages in the pectin. Unlike the arabinoxylans, which generally have the ferulic acid esterified to the C-5 position of the arabinose

in the polysaccharides (Supplementary Fig. 1b), in sugar beet pulp, the ferulic acid is linked to the core α-1,5-arabinan chains and the galactopyranosyl residues of the core β-1,4-linked type I galactan chains[34,35]. Only Bi1040-FAE and BeGH43/FAE released ferulic acid from sugar beet pulp, further demonstrating the versatility of this new ferulic acid esterase.

**Molecular structures of the esterases in the EGE-PUL.** The crystal structure of BeGH43/FAE (residues 23–796) was determined to 2.7 Å ($R_w$ = 22.0, $R_f$ = 26.7) (Fig. 5a). The X-ray data collection and refinement statistics are presented in Supplementary Table 1. The N-terminal GH43 domain (residues 23–523) and the C-terminal esterase domain (residues 524–796) of BeGH43/FAE share a minimal number of interactions that may impart some rigidity to their juxtaposition. The four monomers in the asymmetric unit could be overlaid with an RMSD of 0.6–1.1 Å for all residues without significant deviations.

The GH43 domain is most similar to the arabinofuranosidase (GH43) of *Humicola insolens* (PDB 3ZXL, $Z$ = 41.3, RMSD = 2.1 Å for 465 amino acids[36]) and the β−1,4-xylosidase Xyl of *Geobacillus thermoleovorans* (PDB 5Z5I, $Z$ = 39.3, RMSD = 2.1 Å for 454 residues[37]). Homology is shared between the BeGH43 and *G. thermoleovorans* Xyl enzyme over canonical five-bladed β-propeller fold of the catalytic domain (residues 23–313) and the β-sandwich domain (residues 314–523) (Fig. 5a). An overlay of the active sites of BeGH43 and the Xyl complexed with xylose and arabinose reveals conservation of most residues, including the catalytic Asp-Asp-Glu triad (Supplementary Fig 12a, b).

The C-terminal ferulic acid esterase (FAE) domain of BeGH43/FAE shares 76% amino acid sequence identity (87% similarity) with Bi1040-FAE and displays an α/β fold with a 7-stranded central β-sheet flanked by 8 α-helices (Fig. 5a). The FAE domain of BeGH43/FAE revealed structural similarity to a *Lactobacillus plantarum* tannase and an overlay of the S163A catalytically inactive mutant of the *L. plantarum* tannase complexed with gallic acid (PDB 4JUI, 2.0 Å RMSD for 246 Cα pairs) allowed the identification of the putative residues in BeGH43/FAE that discriminate substrate (Fig. 5a, b). Tannases are a family of esterases that target the galloyl ester bond in hydrolysable tannins to release gallic acid, and the *L. plantarum* enzyme was reported to display an α/β structure with a catalytic triad constituted of Ser/His/Asp[38]. Despite differences in the specificities of these enzymes, there is considerable conservation of the active site architecture, including placement of the catalytic nucleophile (BeGH43/FAE-S634 and *L. plantarum* tannase S163A inactive mutant), the hydrophobic pocket created by Val744/Val745 (BeGH43/FAE) underneath of the phenolic substrate, and W563 (BeGH43/FAE) towards the exterior of the binding site (Fig. 5b). These are conserved as Ile206 and Y78 in the tannase structure.

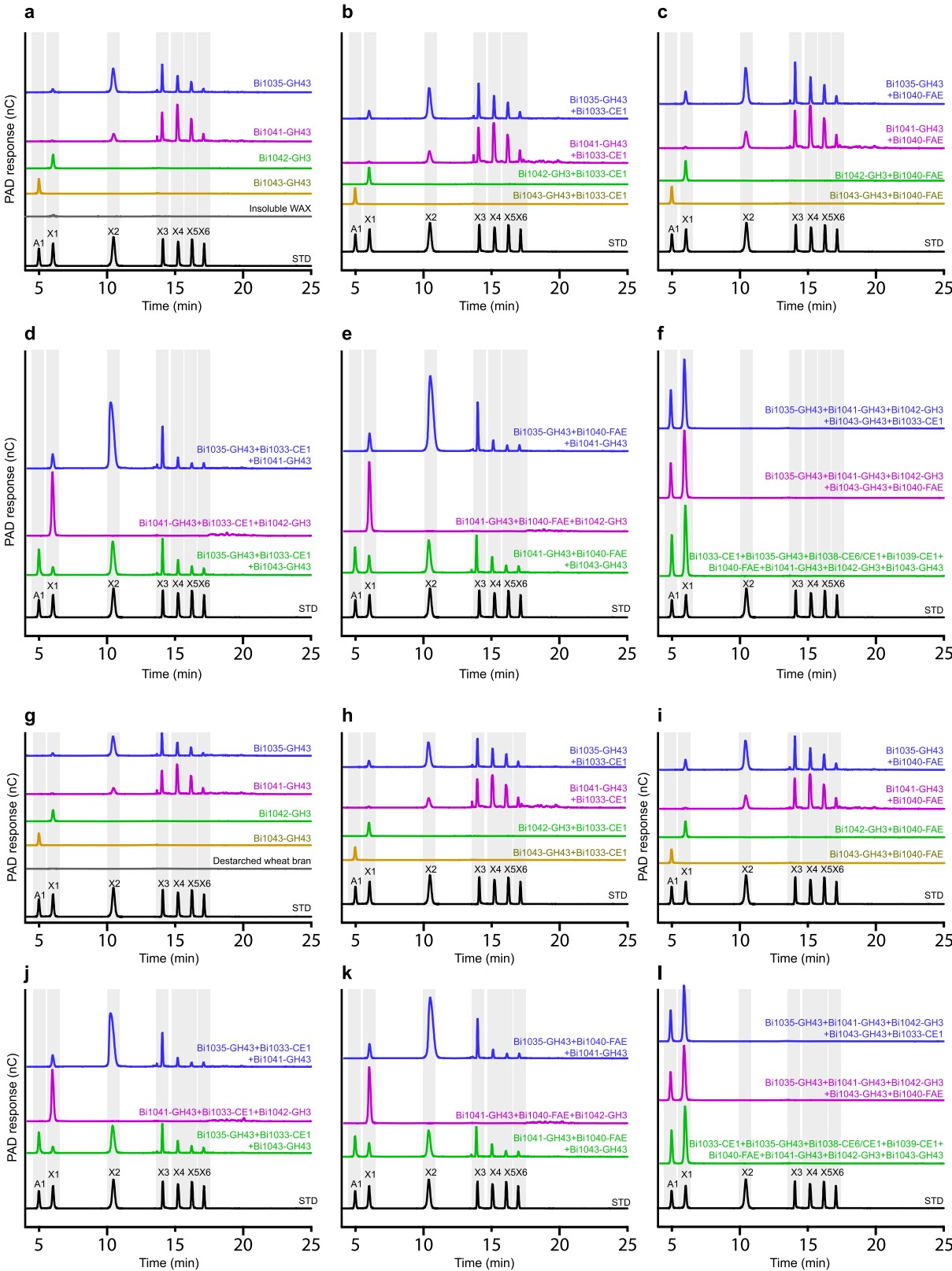

The crystal structures of Bi1039-CE1 and Bi1033-CE1 were also determined to observe the differences in enzyme specificity (see Supplementary Table 1 for the X-ray data collection and refinement statistics). The structure of Bi1039-CE1 included residues 25–382 (1.74 Å, $R_w = 18.2$, $R_f = 21.6\%$) with two molecules in the asymmetric unit as a dimer. The structure of Bi1033-CE1 was determined in two space groups: $P3_121$ (2.24 Å, $R_w = 19.0\%$ $R_f = 23.6\%$) and $P6_422$ (1.71 Å, $R_w = 17.1\%$, $R_f = 19.1\%$), the former of which displays a dimer in the asymmetric unit. The dimer of Bi1039-CE1 can be superimposed with the dimer of Bi1033-CE1 with 1.5 Å RMSD over 648 Cα pairs and 51.7% sequence identity (Fig. 5c). The most obvious structural

**Fig. 4 Hydrolytic activities of individual GH enzymes and their combinations with other enzymes encoded in the _B. intestinalis_ EGE PUL. a** Hydrolytic traces of individual GH enzymes showing their end products on incubation with insoluble wheat arabinoxylan (InWAX). **b–f** Hydrolytic traces of multiple combinations of the enzymes encoded in the EGE PUL of _B. intestinalis_ showing their end products on incubation with InWAX. **g** Hydrolytic traces of individual GH enzymes showing their end products on incubation with de-starched wheat bran (DWB). **h–l** Hydrolytic traces of multiple combinations of the enzymes encoded in the EGE PUL of _B. intestinalis_ showing their end products on incubation with DWB. The end products of hydrolysis were identified and quantified by HPAEC-PAD for the three independent reactions reported in Figs. 3f, g, and one set of chromatograms is presented. Arabinose (A1), xylose (X1) and xylo-oligosaccharides (X2 to X6) were mixed and analyzed by HPAEC-PAD to serve as standards for the assignment of the released products.

difference between the two enzymes is the loop from residues 221–233 in Bi1039-CE1 with the more extended α-helix in loop spanning residues 220–240 in Bi1033-CE1 (Fig. 5c, red box). In the monomeric structure of Bi1033, there is a chain break from 223–237 that could not be modeled, suggesting some flexibility. The helix in Bi1033-CE1 (residues 231–237) is proximal to the predicted active site, perhaps restricting access, whereas this loop is smaller and appears to leave the active site more open in Bi1039-CE1 (Fig. 5d). However, the precise contribution of this loop to activity and substrate selectivity is unknown.

Both Bi1039-CE1 and Bi1033-CE1 are comprised of an N-terminal Ig-like fold of ~100 residues followed by the esterase domain, and both domains are involved in dimerization. The best structural homology of the N-terminal domains is to the early set/ fibronectin-like domains of some GH13 enzymes. The larger C-terminal domains of Bi1033-CE1 (residues 126–383) and Bi1039-CE1 (residues 126–382) display a canonical α/β fold with an 8-stranded central β-sheet flanked by α-helices (Fig. 5c). In the structure of Bi1039-CE1, chain B displays an unidentified density at the active site S266 (Fig. 6a, b). The shape of the density does not match any of the crystallization reagents. Automated ligand identification in Phenix suggests this density is an amino acid, possibly lysine, though this does not refine well. The density is somewhat reminiscent of 2-(N-morpholino)ethanesulfonic acid (MES) or ferulic acid, yet these compounds were not added to the crystallization, and in addition PEG failed to refine. Furthermore, the phenolic moiety of ferulic acid clashed with the protein in either orientation. This density is wedged between S266 and S183, both of which have their hydroxyl side chains ~2.3–2.7 Å from the center of the density, making covalent modification unlikely. Because we could not unambiguously model this density, it was left empty in the final structure.

The C-terminal domain of BeGH43/FAE can be superimposed onto Bi1033-CE1 with a RMSD of 3.8 Å for 248 Cα atom pairs (14.9% sequence identity), and onto Bi1039-CE1 with a RMSD of 3.6 Å over 208 Cα atom pairs (13.9% sequence identity) (Fig. 6c). While Bi1039-CE1 is an acetyl-xylan esterase and Bi1033-CE1 a ferulic acid esterase, the enzymes share striking structural homology to each other and the ferulic acid esterase domain of the cellulosomal xylanase Z of _Clostridium thermocellum_ complexed with ferulic acid (PDB 1jt2; Fig. 6d, e). To resolve why Bi1039-CE1 cannot accommodate ferulic acid, we overlaid the active sites of Bi1033-CE1 and Bi1039-CE1 with the _C. thermocellum_ enzyme complexed with ferulic acid (Fig. 5c–e). While most of the active site architecture is conserved among the three proteins, the loop comprising residues 292–300 in Bi1039-CE1 is only ordered in chain B, which displays the unidentified solvent molecule. This loop shapes one side of the active site (Fig. 5d, see Y295/P296 of Bi1039 in purple). In this chain, Y295 has limited mobility and would sterically hinder the binding of ferulic acid based upon the placement of this molecule from the overlay (Fig. 5e). As stated before, we cannot determine from these data if the region from 292–300 is ordered in Bi1039-CE1 chain B because of the captured solvent or due to an artifact of crystal packing; certainly, the observation that this region could not be modeled in chain A

suggests some degree of conformational flexibility. It is possible that the capture of this solvent here demonstrates the degree to which the enzyme can close down upon the active site and possibly restrict the accommodation of larger esterified oligosaccharides that contain feruloyl-like moieties. In both Bi1033-CE1 and the _C. thermocellum_ enzyme, there is fairly conserved placement of A299–G300 and A198–P199, respectively, which does not appear to hinder ferulic acid placement (Fig. 5f).

**Site-directed mutagenesis to identify active site residues.** We used the polypeptide sequences of Bi1040-FAE (EDV05955.1), Bi1033-CE1 (WP_007660993.1 or EDV05948.1), and Bi1039-CE1 (WP_007661004.1 or EDV05954.1) to search the publicly available databases to determine if they are restricted to the phylum Bacteroidetes or whether they are widely distributed. Alignments of the three esterases and some of their homologs are presented (Supplementary Fig. 13a–c) with their predicted catalytic triads highlighted. Thus, for Bi1040-FAE (EDV05955.1)/BeGH43/FAE and their homologs, the predicted catalytic triad is Ser634-His774-Asp742 (BeGH43/FAE numbering), for Bi1039-CE1 and its homologs the predicted catalytic triad is Ser266-His364-Glu332 (Bi1039-CE1 numbering), and for Bi1033-CE1 and its homologs, the predicted catalytic triad is Ser273-His365-Glu336 (Bi1033-CE1 numbering).

To interrogate the contributions of the predicted members of the putative catalytic triad in each esterase group, we subjected each of the amino acid residues to site-directed mutagenesis by substitution with alanine and assessing the catalytic activities of the derived mutant. Thus, for BeGH43/FAE, we made the recombinant mutants S634A, H774A, and D742A; for Bi1033-CE1, the recombinant mutants S273A, H365A, and E336A were made; and for Bi1039-CE1, we made the recombinant mutants S266A, H364A, and E332A.

As shown in the circular dichroism scans (Supplementary Fig. 14a–c), the mutations did not severely impact the secondary structural elements of the mutants compared to their wild-type proteins. These observations were, in general, confirmed by the lack of statistical significance among the comparisons of the secondary structure compositions of the wild-type protein with its mutants (Supplementary Table 2). Using insoluble wheat arabinoxylan (InWAX), a feruloylated oligosaccharide (FAXX), and _para_-nitrophenol acetate (_p_NP acetate) as substrates, we observed that the alanine mutants of the catalytic triad in BeGH43/FAE exhibited severely impaired catalytic activity on InWAX (Supplementary Fig. 14d, BeGH43/FAE panel). In the case of Bi1033-CE1, shown above to exhibit activity on FAXX (Fig. 3d), the S273A and H365A mutations abolished activity, whereas about 30% of activity was still detectable in the E336A mutant (Supplementary Fig. 14e, Bi1033-CE1 panel). We hypothesize that the highly conserved E335 (Supplementary Fig. 13c) can substitute for the function of E336 when this residue is mutated, or E335 is the right member of the catalytic triad. In contrast, the activity of the wild-type Bi1039-CE1, observed with _p_NP acetate as the substrate, was lost by each alanine mutant of its predicted catalytic triad (Supplementary Fig. 14e, Bi1039-CE1 panel).

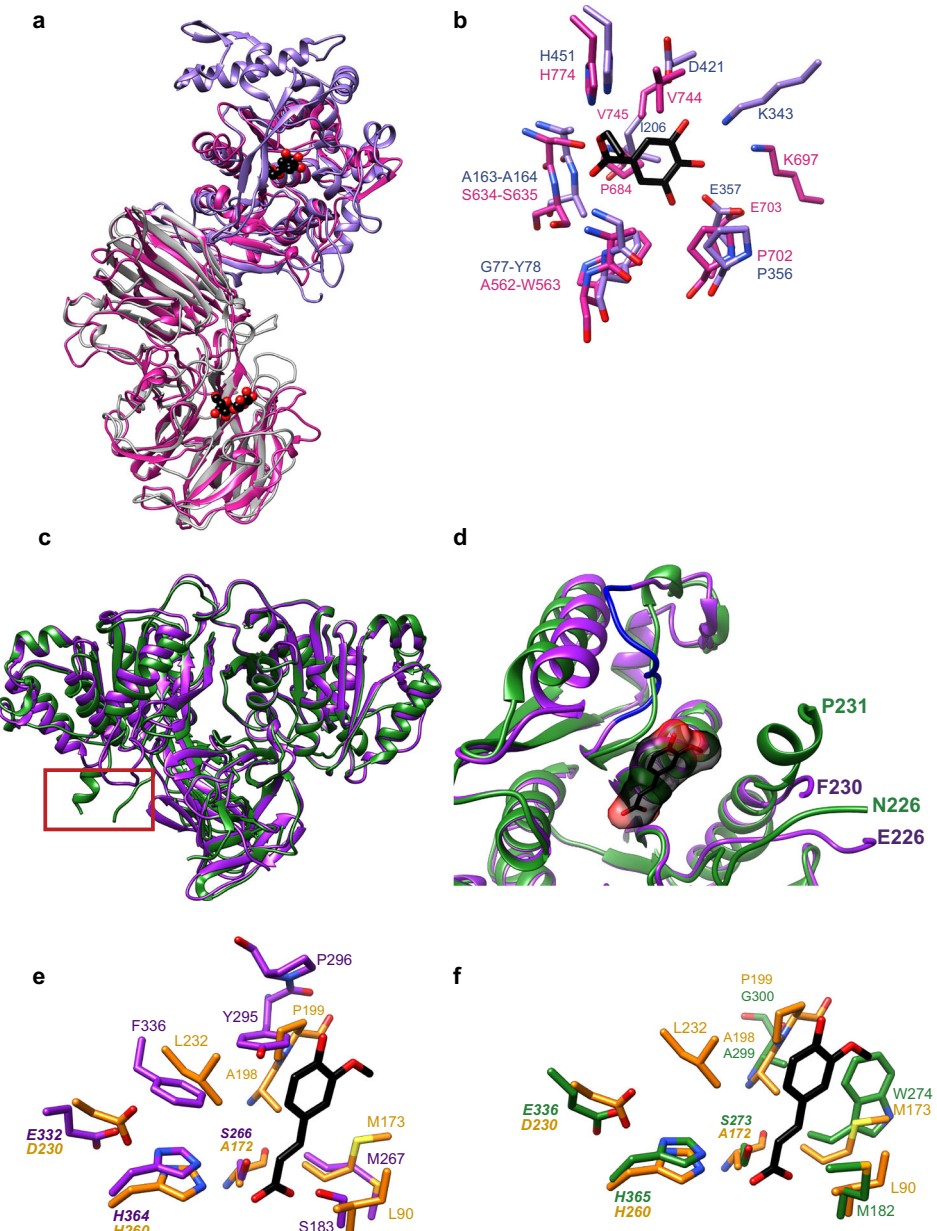

**Fig. 5 Molecular structures of *Bacteroides* spp. esterases. a** Molecular structure of the BeGH43/FAE bifunctional enzyme (pink ribbon). At the N-terminus, the structure of the β,1,4-xylosidase (GH43) from *Geobacillus thermoleovorans* (gray, PDB 5z5i, RMSD = 2.1 Å for 454 residues) has been overlayed, and at the C-terminus the *Lactobacillus plantarum* tannase (purple, PDB 4jui, RMSD = 2.0 Å for 243 residues) has been overlayed with (FAE) of BeGH43/FAE. Ligands for the 5z5i and 4jui structures are shown in spheres to label the active sites. **b** Close-up view of the overlay of the active site of the *L. plantarum* tannase (purple, PDB 4jui) with the C-terminal ferulic acid esterase (FAE) domain of BeGH43/FAE. Residues within 4.0 Å of gallic acid in the tannase and structurally homologous in the *B. eggerthii* enzyme are displayed. **c** Ribbon diagram overlay of the dimeric structures of Bi1033-CE1 (green) and Bi1039-CE1 (purple). The red boxed area highlights the most significant structural deviation between the enzymes, comprising residues 221–233 in Bi1039-CE1 and 220–240 in Bi1033-CE1 which includes a chain break from 227 to 230. **d** Close-up overlay of the active sites of Bi1033-CE1 (green) and Bi1039-CE1 (purple; chain B which includes a small break as indicated). An overlay of Bi1033-CE1 and Bi1039-CE1 with the cellulosomal xylanase Z C-terminal ferulic acid esterase domain from *C. thermocellum* (PDB 1jt2) complexed with ferulic acid (black and red) was performed to highlight the active site. The region spanning G291-P300 in blue from Bi1039-CE1 is highlighted as it deviates from Bi1033-CE1 in contributing to the active site. **e** Close-up view of the putative active site residues in Bi1039-CE1 (purple) and the cellulosomal xylanase Z C-terminal ferulic acid esterase domain from *C. thermocellum* (yellow, PDD 1jt2) complexed with ferulic acid (black and red). **f** Close-up view of the putative active site residues in Bi1033-CE1 (green) and the *C. thermocellum* enzyme with ferulic acid (yellow, black, and red).

**Structural dynamics and substrate accessibility of the esterases.** We followed up our structural analyses aimed at gaining insight into the molecular basis for the substrate specificities of the three esterases with Molecular Dynamics (MD) simulations. The three esterases show large differences in their substrate specificities in cleaving ferulic acid from feruloylated oligosaccharides (Fig. 3d). Bi1040-FAE exhibits strong cleavage activity on diverse substrates, whereas Bi1039-CE1 displayed very minimal activity towards breaking esterase bonds in feruloylated oligosaccharides. Furthermore, Bi1033-CE1 only exhibited catalytic activities on a

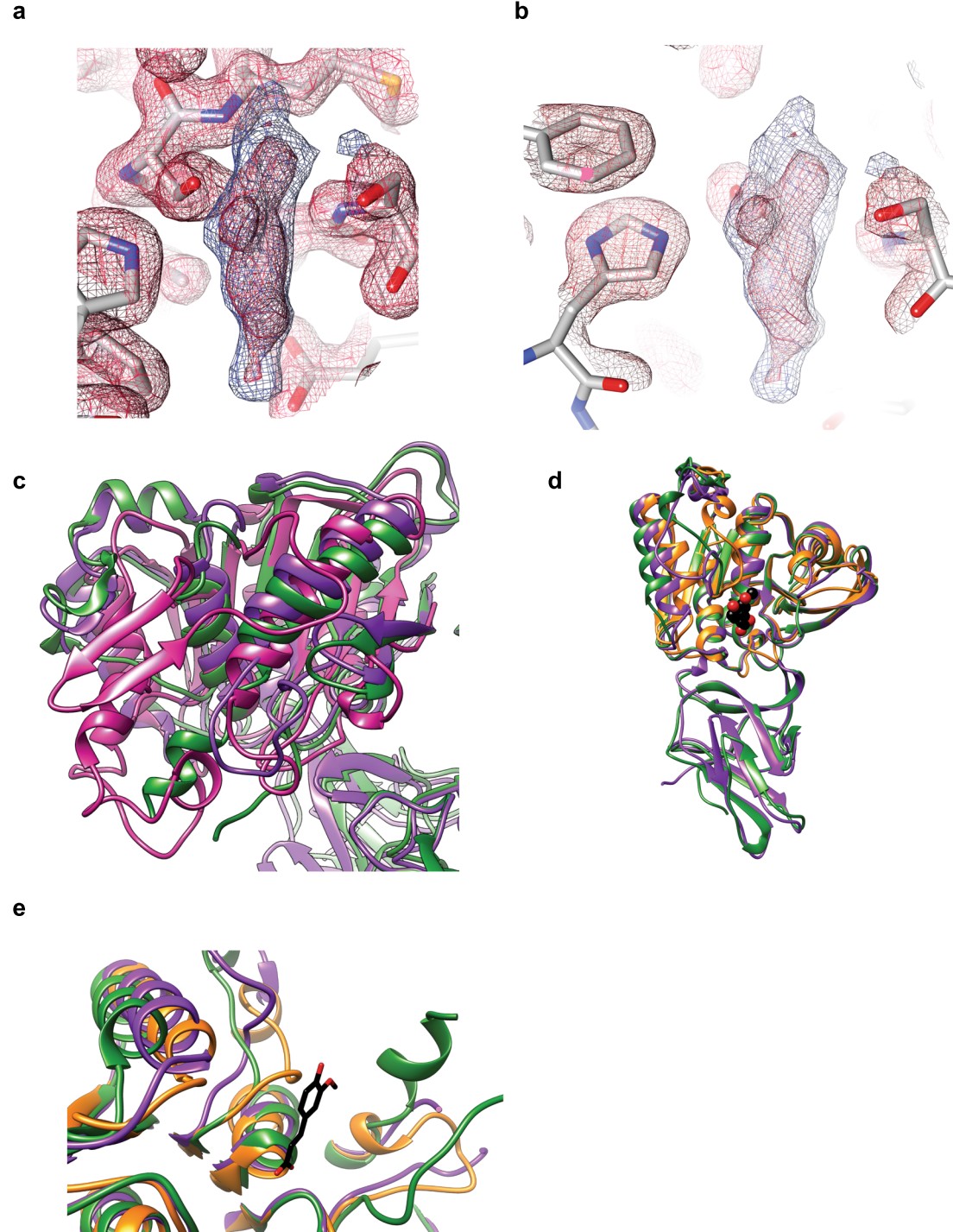

**Fig. 6 Structure comparisons of the Bacteroides esterase enzymes. a**, **b**, Close-up view of the unidentified solvent density at chain B of the Bi1039-CE1 structure. The $F_o$–$F_c$ map is displayed in blue and contoured at 3.5 σ while the 2Fo-Fc for the surrounding residues is shown in red and contoured at 1.5 σ. Maps were calculated in Phenix and rendered in Chimera. **c** Overlay of the C-terminal domain of the BeGH43/FAE enzyme (pink) with Bi1033-CE1 (green) and Bi1039-CE1 (purple). **d** Overlay of the Bi1033-CE1 (green), Bi1039-CE1 (purple) and the ferulic acid esterase domain of cellulosomal xylanase Z from *Clostridium thermocellum* with ferulic acid (yellow, PDB 1JT2). Ferulic acid bound to xylanase Z is shown in black and red spheres. **e** Close-up view of the active sites from a superposition of Bi1033-CE1 (green), Bi1039-CE1 (purple) and the ferulic acid esterase domain of cellulosomal xylanase Z from *C. thermocellum* with ferulic acid (yellow, PDB 1JT2). Ferulic acid bound to xylanase Z is shown in black and red sticks.

limited set of feruloylated oligosaccharides (FA, FAXX). In esterase enzymes, catalytic triad residues [Nucleophile (SER), Base (HIS), and Acid (ASP/GLU] are essential for the catalytic action of the protein (confirmed by our mutagenesis experiments in Supplementary Fig. 14). The structural dynamics and substrate accessibility of these residues are, therefore, critical for understanding the selectivity differences among the enzymes.

To explore this observation further, we proposed two hypotheses to explain the different substrate specificities of the esterase enzymes. First, the catalytic action of the enzymes

depends on attack of the nucleophile (SER) on the carbonyl carbon of the ester. Feruloylated oligosaccharides are large and their binding pocket volume near the catalytic triad must be large enough to accommodate these large substrates. Therefore, an enzyme with a large binding pocket volume likely can accommodate different large substrates and exhibit catalytic triad activity towards diverse set of substrates. The second hypothesis is based on the structural stability of the catalytic triad. During the catalytic reaction, the acid (GLU/ASP) residue forms a stable hydrogen bond with the base (HIS) residue to make it more electronegative[39] and thereby increase its capacity to take up a proton from the nucleophile (SER). Therefore, the stability of acid (GLU/ASP)—base (HIS) hydrogen bond is imperative for catalytic action of the esterase enzyme.

To test these hypotheses, we performed Molecular Dynamics (MD) simulations of the three esterases (Bi1033-CE1, Bi1039-CE1, and the Bi1040-FAE homolog of BeGH43/FAE) to observe the differences in their binding pocket volume and the stability of the hydrogen bond between acid and base in the catalytic triad. We anticipated that such data would offer insight into how three enzymes of similar polypeptide fold (Fig. 6c) exhibit very different catalytic properties (Fig. 3d, e and Supplementary Fig. 5). For these simulations, it was assumed that the N-terminal residues of the esterase enzymes do not affect the structural dynamics of the esterase domain. Under this assumption, 2 μs long simulations of the esterase domain of each enzyme were performed.

Simulations of enzymes (without substrate) revealed that the esterase domain adopts stable conformation (root mean square deviation (RMSD) < ~ 3 Å) within the duration of the simulation (Supplementary Fig. 15a–c). To compare the substrate accessible volume near the active site for each protein, pocket volume calculations were performed on the ensemble of protein structures obtained from the simulation. Our calculations show that the FAE domain of BeGH43/FAE (mean binding pocket volume: 97.3 Å$^3$) provides a larger substrate accessible volume as compared to Bi1039-CE1 (mean binding volume: 81.2 Å$^3$) and Bi1033-CE1 (mean binding volume: 62.8 Å$^3$) (Fig. 7a–d). This observation supports our first hypothesis that increase in the binding pocket volume is correlated with high catalytic activity of BeGH43/FAE (and Bi1040-FAE) to break ferulic acid side chain in diverse large substrates, including feruloylated oligosaccharides.

To further interrogate our first hypothesis, we performed substrate binding simulations for Bi1039-CE1 and the FAE module of BeGH43/FAE with FA (feruloylated arabinose) as a substrate. Ten μs of simulation were performed with ~8 mM concentration of FA in the solution for each protein system. For catalytic action of the esterase, nucleophilic residue (SER) needs to attack the carbon atom (C9 atom for FA) of the carbonyl (C = O) bond in FA. Markov state model (MSM) were used to build weighted free energy landscapes of substrate binding. These landscapes reveal the distances of the catalytic triad residues from FA in the binding pocket (Fig. 7e–g and Fig. 8). The results show that in the binding pose of FA, the C9 atom of FA is closer to SER residue in the FAE module of BeGH43/FAE as compared to Bi1039-CE1. Higher substrate accessibility in the FAE module of BeGH43/FAE (and therefore Bi1040-FAE) allows FA to adopt a catalytically competent binding pose. Therefore, our simulation results support the hypothesis that substrate accessibility regulates enzyme activity in the esterases.

To test the second hypothesis, the acid (GLU)-base (HIS) side-chain distances were calculated for each enzyme (Supplementary Fig. 16). In this analysis, the formation of stable hydrogen bond is decided by the distance cut-off of 2.5 Å and the angle greater than 120° between donor hydrogen and acceptor atoms[40]. According to this criterion, over 99% frames of the trajectory forms stable

acid-base hydrogen bond in Bi1033-CE1 (ferulic acid esterase), whereas in the case of Bi1039-CE1 (acetyl-xylan esterase) it is ~64%. The breaking of this critical interaction between catalytic triad residues could explain the lower catalytic activity of Bi1039-CE1 for feruloylated oligosaccharides in comparison with Bi1033-CE1. Therefore, simulation results support the proposed hypotheses and are able to explain different substrate specificities of the three esterase enzymes.

**The cleaved ferulic acid accumulates during fermentation**. Our previous results[28] and the present study show that *B. intestinalis* is rich in enzymes that target cleavage of ferulic acid off complex arabinoxylans. We, therefore, sought to determine whether the phenolic compound is metabolized by the bacterium. The cells of *B. intestinalis* were, therefore, grown on de-starched wheat bran (DWB) or wheat bran (WB), with the amounts of ferulic acid in the media monitored. After 24 h of incubation, ferulic acid amounts about doubled and tripled in the WB medium and the DWB medium, respectively (Supplementary Fig. 17a), with concomitant production of succinate, propionate, and acetate (Supplementary Fig. 18a). Furthermore, increased protein contents of the media supported bacterial fermentation and cell proliferation during growth on the substrates (Supplementary Fig. 18b). We further confirmed in separate experiments that when we culture *B. intestinalis* on ferulic acid in the presence of the constituent sugars of arabinoxylan, the cells only metabolize the sugars and not the added ferulic acid after 24 h of incubation. In addition, cell proliferation was not detected during culturing on medium with ferulic acid as the only energy source (Supplementary Fig. 17b).

As shown in Fig. 2, other members of the colonic Bacteroidetes harbor the EGE PUL. Thus, we investigated if the release of ferulic acid into the culture medium is common to these arabinoxylan degraders from the human colonic microbiome. *Bacteroides eggerthii*, from which we discovered the variant of Bi1040-FAE, i.e. BeGH43/FAE, was also examined. In all three *Bacteroides* spp., i.e., *B. eggerthii*, *B. oleiciplenus,* and *B. cellulosilyticus*, we observed doubling of ferulic acid concentration in their growth medium, with *B. intestinalis* tripling the concentration of the phenolic compound after 24 h of incubation (Fig. 9). This observation showed that ferulic acid release and accumulation during complex arabinoxylan metabolism is common to these four colonic *Bacteroides* spp.

## Discussion

The phenolic compound ferulic acid is abundant in arabinoxylan-rich human foods[41], and is absorbed in both the small and large intestines. However, the main site of absorption is the colon, where it is released by microbial esterases[42,43]. The microbes responsible for ferulic acid release in the colon and their fermentation strategies are not completely understood. However, due to the expansive health benefits of ferulic acid and the potential to manipulate the microbiome for higher yields, there is increased interest in a mechanistic understanding of ferulic acid metabolism. Here, we show that several *Bacteroides* spp. from the human colonic microbiome can discriminate complexity in arabinoxylans and deploy different PULs for efficient arabinoxylan metabolism.

Administration of arabinoxylan lacking esterified ferulic acid residues led to upregulation of expression of two PULs in *B. intestinalis*[25] (in a range of 7-fold to 210-fold), with the EGE PUL investigated in the present study barely upregulated (<3-fold) on this substrate (Supplementary Data 2). A cardinal feature of the PULs used by *B. intestinalis* in metabolizing the less complex arabinoxylan is the presence of a unique GH10 endoxylanase with

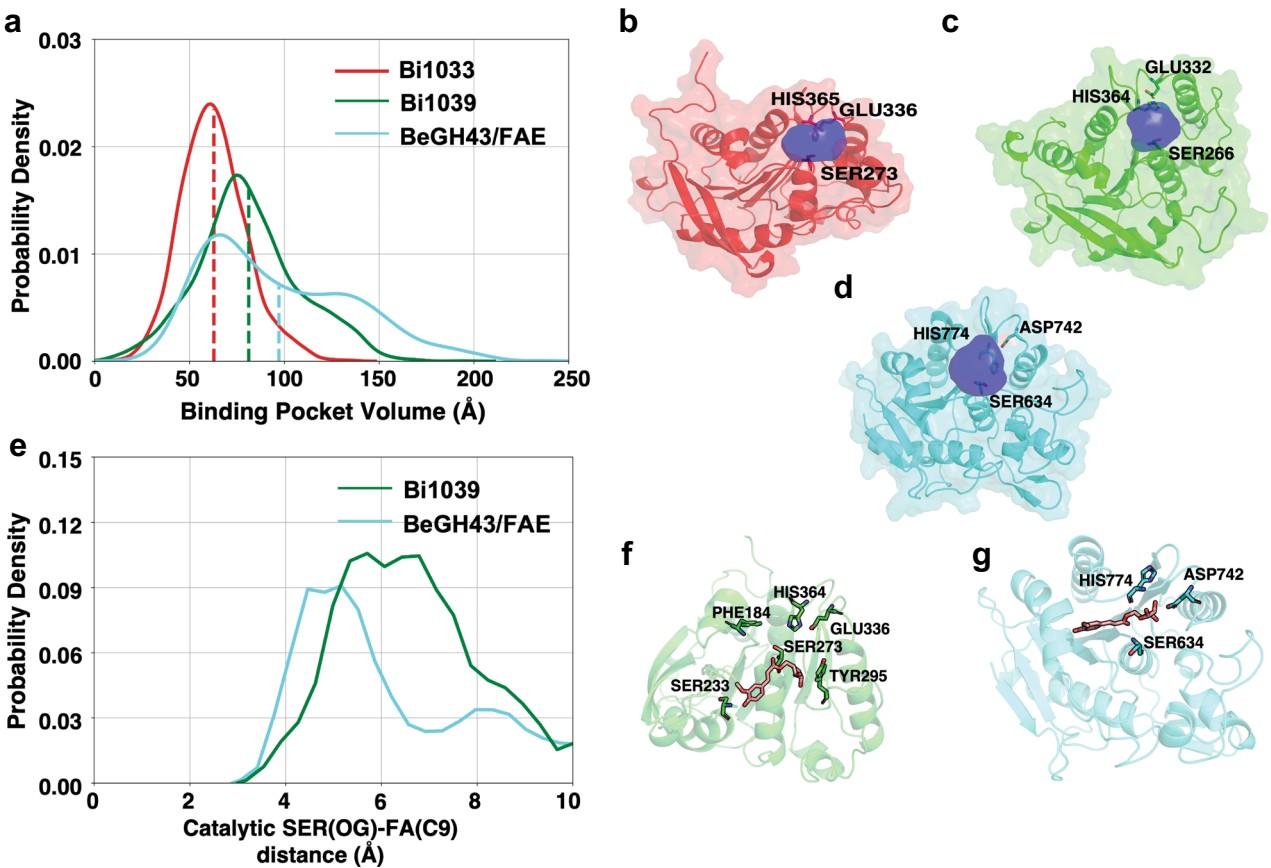

**Fig. 7 Larger substrate accessible volume is correlated with selectivity differences. a** Binding pocket volumes of Bi1033-CE1 (red), Bi1039-CE1 (green), and the FAE domain of BeGH43/FAE (cyan) trajectories are represented as histogram. Pocket volumes were calculated for the frames with 1 ns interval of each trajectory. Mean binding pocket volumes are represented as vertical lines. **b**, **c**, **d** Cartoon and surface representation of esterase domains of Bi1033-CE1, Bi1039-CE1, and BeGH43/FAE, respectively, are presented. Catalytic triad residues are shown as sticks. Representative mean pocket volumes are displayed as transparent blue surface. **e** Binding distance of FA(C9) & SER(OG) are represented as Markov state model (MSM) weighted 1-D histogram. **f**, **g** Cartoon representation of stable binding poses of FA substrate close to active site in Bi1039-CE1 and in the FAE domain of BeGH43/FAE, respectively.

carbohydrate binding module (CBM) insertions in the catalytic module[25], an observation confirmed with another colonic bacterium, *Bacteroides ovatus* (Cann et al. unpublished data). This uniquely configured GH10 polypeptide appears to be invariable, in terms of its structure and position in this PUL, which is conserved in diverse Bacteroidetes that likely have the capacity to ferment less complex arabinoxylans[44]. In contrast, growth on complex arabinoxylan, with intact ferulic acid side chains, led to a strong upregulation of the EGE PUL (>60–>100-fold) in *B. intestinalis*. Interestingly, however, some of the genes in one of the PULs that were upregulated on the less complex arabinoxylan were also upregulated on the complex arabinoxylan, albeit less than the EGE PUL. The associated transporter (SusC/SusD pair) of the less complex arabinoxylan-targeting PUL was, however, upregulated 5–6-fold compared to the 68–98-fold upregulation of the SusC/SusD pair of the EGE PUL (Supplementary Data 2). By examining expression of the EGE PUL in multiple colonic *Bacteroides* spp., we discovered that this gene cluster is conserved in several colonic members of this genus, likely with the capacity to sense and degrade complex arabinoxylans. Consistent with our observation, earlier work by Rogowski and co-workers also demonstrated that *Bacteroides* spp are able to sense the complexity between glucuronoxylan and arabinoxylan[26]. These findings collectively suggest that the Bacteroidetes PULs are finely tuned to sense complexity in different polysaccharides, a regulatory strategy that likely enhances energy conservation by eliminating inefficiencies in PUL expression.

The analyses of the enzymatic activities associated with the EGE PUL demonstrated that GH43 family enzymes are critical to the degradation of the xylose-linked backbone, in contrast to the enzymes that are fundamental to the degradation of the simple or soluble arabinoxylan[25]. We further demonstrate that the *Bacteroides* spp investigated in the present study, especially *B. intestinalis* and *B. cellulosilyticus*, are well equipped for extensive degradation of complex arabinoxylans. The domain architectures of the enzymes suggested that BACINT_01041-GH43 (or Bi1041-GH43) is anchored on the cell surface to initiate degradation of the complex arabinoxylans, and as an endo-acting enzyme, it generates oligosaccharides that are then transported through the SusC/SusD transporter system into the periplasm. Within the periplasm, the transported products are then further degraded into the unit components.

The activities detected in the individual enzymes and their combinations suggest that the esterases (Bi1033-CE1, Bi1038-CE6/CE1, Bi1039-CE1, and Bi1040-FAE) remove the side-chain acetyl and feruloyl groups, to suppress the steric hindrance imposed by the side chains on the endo-acting enzymes. The resulting products are then attacked by the α-arabinofuranosidase Bi1043-GH43 to remove the arabinose side-chains, leaving the xylo-oligosaccharides for degradation by Bi1035-GH43 and the β-xylosidase Bi1042-GH3 to xylose. While the order of the enzymatic steps described here may differ from the in vivo steps, it is clear that the enzymes in the EGE PUL collectively can depolymerize complex arabinoxylans to their monomeric sugars (xylose

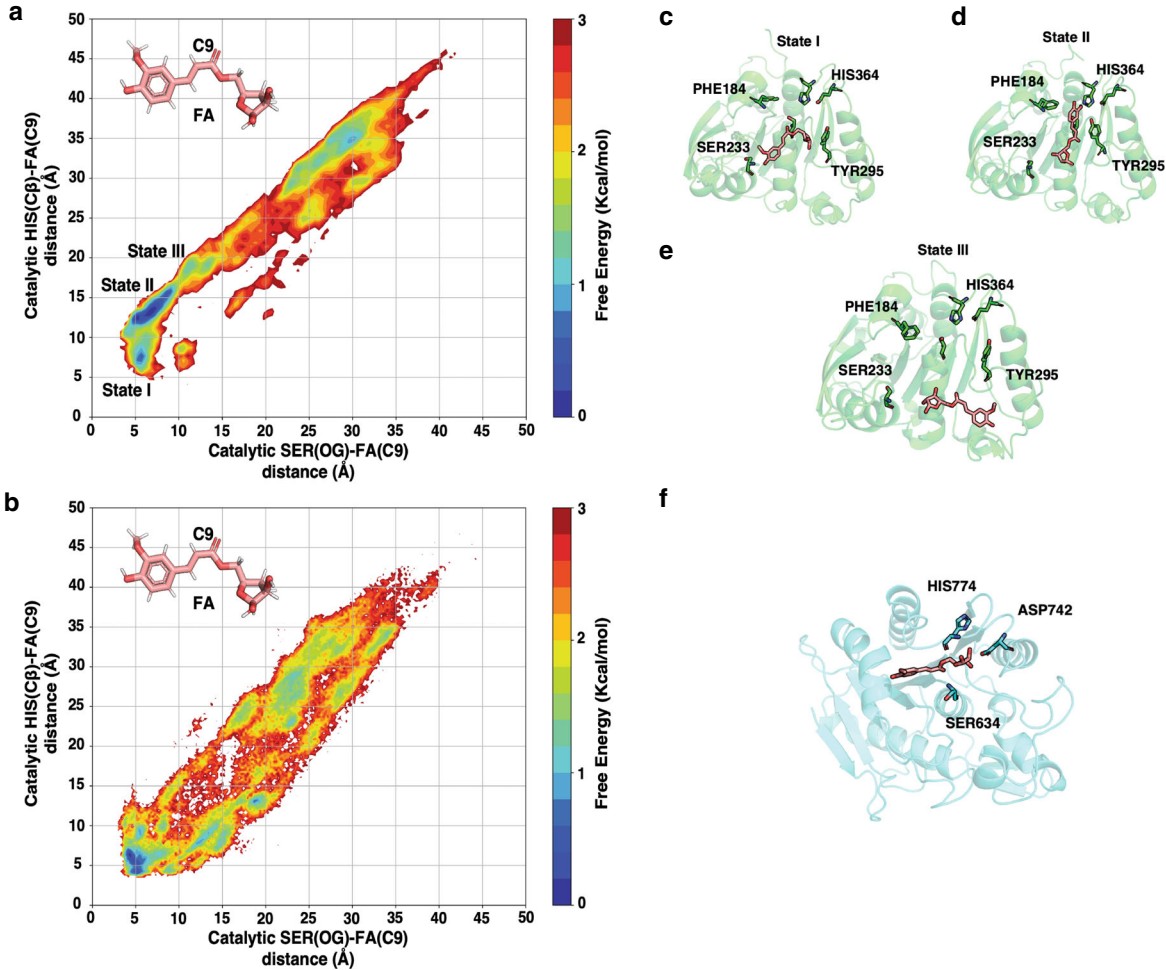

**Fig. 8 Accessibility of reaction center of the substrate by catalytic triad. a, b** Markov state model (MSM) weighted free energy landscapes are projected as 2-D plots with respect to FA(C9) & HIS (Cβ) distance and FA(C9) & SER(OG) distance. **c–f** Cartoon representation of different stable binding poses of FA substrate close to active site in Bi1039-CE1 (**c, d, e**) and the FAE domain of BeGH43/FAE (**f**). In the stable binding state I (**c**), the aromatic ring of FA binds far away from catalytic HIS364 residue compared to state II (**d**) in Bi1039-CE1. State III (**e**) represents the pose where ligand is entering the pocket.

and arabinose, Fig. 4) for transport into the cytoplasm and metabolism to release energy and cellular building blocks.

Based on our insights, we present a model which localizes all the esterases in the periplasm (Fig. 10a, b); however, the model contradicts the accumulation of ferulic acid in the spent medium during complex arabinoxylan metabolism, unless the *Bacteroides* spp under study efflux the phenolic compound. It has been known for decades, however, that lipoproteins of Gram-negative bacteria, although generally considered to be present in the inner membrane, are unexpectedly localized on the outer membrane[45–47]. Furthermore, a recent analysis on lipoproteins in the human gut bacterium *Bacteroides fragilis* demonstrated that surface exposure of lipoproteins is very common[48]. In fact, of 229 proteins determined by LC-MS/MS to be outer membrane proteins in this bacterium, close to half were predicted to be lipoproteins. The authors further provided evidence suggesting that these lipoproteins are secreted by a type I or type VI secretion system found in *B. fragilis*[48]. More importantly, the SusD proteins of the Bacteroidetes, which are known to be involved in substrate binding, are lipoproteins. However, they are located on the cell surface to perform their function[47,49]. It is, thus, conceivable that the esterases, especially the ferulic acid esterases described in this work, are surface exposed and therefore cleave and accumulate the ferulic acid extracellularly during catabolism of complex arabinoxylans (Fig. 10c).

While it is likely that the human colonic microbiome harbors organisms with ferulic acid metabolizing capacity, which can lead to bio-transformations of the phenolic compound[17], we posit that enough of the phenolic compound is released from complex arabinoxylan-rich diets with some amount absorbed by the host. The initiation of complex arabinoxylan degradation in the extracellular environment further suggests that these arabinoxylan degraders contribute to the important function of cross-feeding members of the microbiome, especially those organisms lacking the hydrolytic enzymes that target the polysaccharides[16,18]. Furthermore, the importance of ferulic acid and its pervasiveness and diversity of linkages in the human diet is underscored by the abundance and also diversity of ferulic acid esterases in the colonic *Bacteroides* spp.[17,28].

Short chain fatty acids, derived from colonic fermentation by the microbiome, are already known to contribute to the health and nutrition of the human host. Thus, significant in the present report is our observation that ferulic acid accumulates during metabolism of complex arabinoxylans by diverse colonic *Bacteroides* spp. (Fig. 9 and Supplementary Fig. 17a). Furthermore, *B. intestinalis* did not use ferulic acid when the phenolic compound was added as the sole carbon source to the medium (Supplementary Fig. 17b). These observations suggest that these *Bacteroides* spp. release the phenolic compound

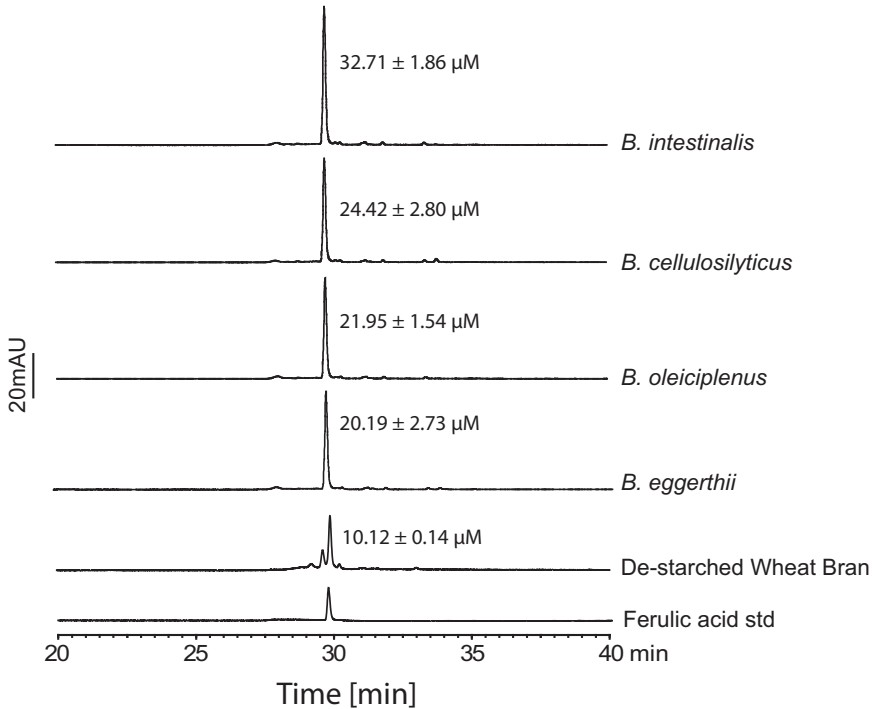

**Fig. 9 Growth of diverse *Bacteroides* spp. in minimal medium containing de-starched wheat bran as the sole carbon source.** HPLC-DAD chromatogram of *B. intestinalis*, *B. cellulosilyticus*, *B. oleiciplenus*, and *B. eggerthii* spent medium from growth on de-starched wheat bran as the sole carbon source showing release of ferulic acid into the medium. The de-starched wheat bran trace shows concentration of free ferulic acid in medium without bacterial inoculation (as control). The results are presented as the mean ± the standard deviation of three independent reactions ($n = 3$), and the traces are representatives of the three different reactions. The source data underlying Fig. 9 are provided in the Source Data file.

without further metabolism or do not metabolize it in the presence of xylose and or arabinose. Ferulic acid exerts its antioxidant activity through possession of several motifs that endow it with a strong free radical scavenging properties[50], including its unsaturated C–C bond which stabilizes free radicals and therefore inhibit free radical attack, and its carboxylic group, which can act as a lipid anchor and, therefore, protect against lipid peroxidation[51]. Both free radical attack and lipid peroxidation can lead to cell damage and death. The accumulation of ferulic acid during metabolism of arabinoxylan by the colonic *Bacteroides* spp could then potentially provide the molecular basis for the observed impact of arabinoxylan on human health[52–55].

Colonic microbiome research has emphasized analyses of large genomic datasets, and functional approaches are only recently receiving attention. Using a multidisciplinary approach, we provide mechanistic insights into the expression and importance of the Bacteroidetes EGE PUL in arabinoxylan metabolism. Furthermore, we characterize genes hitherto of unknown function, but widespread in nature (Figs. 3,4 and Supplementary Fig. 13) and demonstrate their importance in complex arabinoxylan degradation. The fusion of the promiscuous ferulic acid esterase to an arabinofuranosidase by some colonic *Bacteroides* spp. should increase the efficiency of complex arabinoxylan degradation, with the release of large amounts of ferulic acid. Exploring combinations of these organisms as probiotics with complex arabinoxylans, as the accompanying prebiotics, should yield significant health benefits to the host, in congruence with the proposal that regular ingestion of ferulic acid may provide substantial protection against various oxidative stress-related diseases[50]. Furthermore, the results reported herein should help establish the molecular basis for manipulating ferulic acid, an abundant compound in human diets, to benefit the human host.

## Methods

**Bacterial strains and other materials.** *Bacteroides intestinalis* 341 (DSM 17393) was a kind gift from Jeffrey I. Gordon (Washington University in St. Louis, Missouri) and was originally obtained from the DSMZ (German Collection of Microorganisms and Cell Cultures, Braunschweig). *Bacteroides eggerthii* 1_2_48FAA, *Bacteroides cellulosilyticus* DSM 14838 and *Bacteroides oleiciplenus* YIT 12058 were kindly provided by Eric Martens (University of Michigan, Ann Arbor, Michigan). The NEB® 5-alpha Competent *Escherichia coli* (High Efficiency) and Phusion® High-Fidelity DNA Polymerase were purchased from New England Biolabs (Ipswich, MA). The *E. coli* BL21-CodonPlus (DE3) RIL competent cells were obtained from Agilent (Santa Clara, CA). The pET-28a plasmid used for cloning has the kanamycin resistance gene replaced with the ampicillin resistance[56]. The QIAprep Spin Miniprep and QIAquick PCR Purification kits were obtained from Qiagen, Inc. (Valencia, CA). Amicon Ultra-15 centrifugal filter units with 10 kDa molecular mass cutoffs (MMCOs) were obtained from Millipore (Billerica, MA). The feruloylated oligosaccharides from arabinoxylans (FA, FAX, FAXX, FAXG) were kindly provided by Rachel Schendel (University of Kentucky) and the pectins (FA₂, FA₃, FG₂) were previously purified from Driselase and acid hydrolysates of wheat, maize, and sugar beet insoluble fiber[28,57]. Soluble wheat arabinoxylan with medium viscosity (sWAX), insoluble wheat arabinoxylan (InWAX) and xylo-oligosaccharides were obtained from Megazyme (Bray, Ireland), and oat spelt xylan (OSX) was purchased from Sigma-Aldrich (St. Louis, MO).

**Growth conditions.** Bacterial growth was carried out inside an anaerobic chamber with a gas mixture of 85% $N_2$, 10% $CO_2$, and 5% $H_2$. *Bacteroides* spp cells were cultured from a single colony inoculated into a medium supplemented with Brain-heart infusion (BHIS) at 37 g/L, hemin chloride (1.9 μM), cysteine hydrochloride at 4 mM and sodium bicarbonate at 9.5 mM. For all growth experiments, 100 μL of cells of an overnight culture in the BHIS medium were inoculated into a defined medium containing a specified carbon source and passed through three dilutions to remove any residues derived from the growth in the BHIS medium. The defined medium was composed of sodium chloride (15 mM), dipotassium hydrogen phosphate (5 mM), monopotassium hydrogen phosphate (5 mM), sodium bicarbonate (9.5 mM), cysteine hydrochloride (4 mM), magnesium (II) chloride heptahydrate (0.1 mM), calcium (II) chloride dihydrate (54 μM), iron (II) sulfate heptahydrate (1.4 μM), hemin chloride (1.9 μM), vitamin K3 (5.8 μM), vitamin B12 (7.3 nM), and a specified carbon source (5 mg/mL).

**Gene cloning, expression, and protein purification.** The Polysaccharide-Utilization Loci Database (PULDB) server (http://www.cazy.org/PULDB/) was used to identify gene clusters similar to the esterase genes-enriched (EGE) PUL of

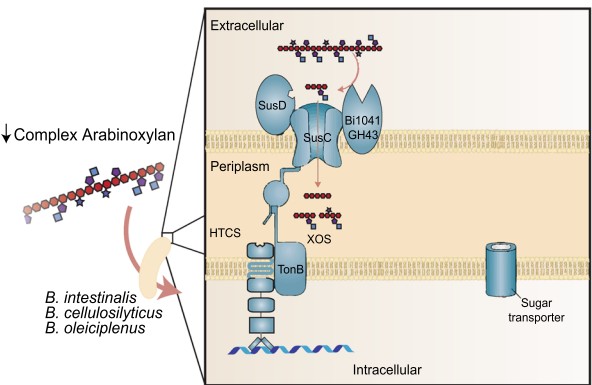

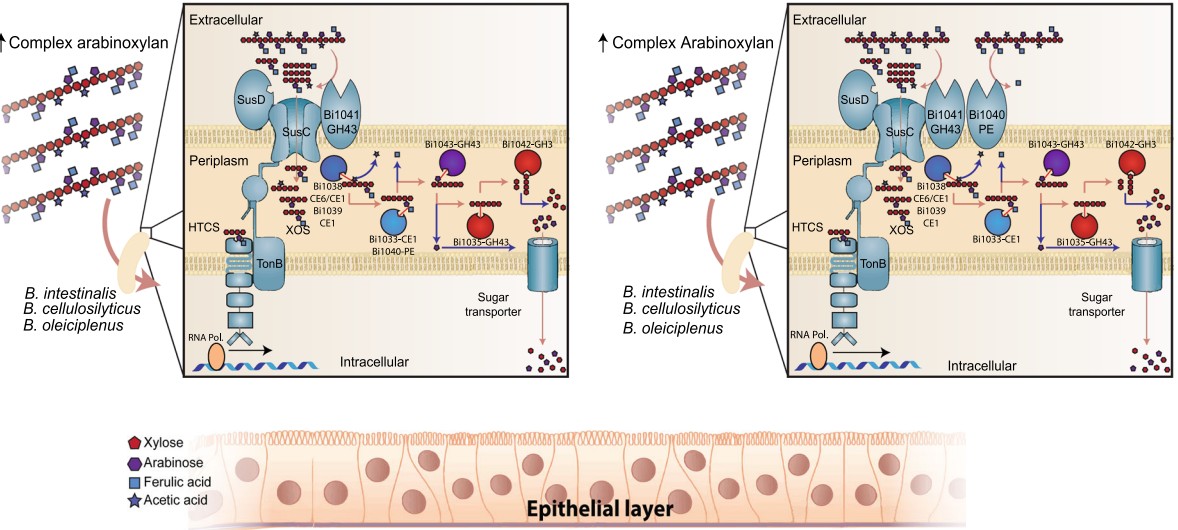

**Fig. 10 A model predicting the regulation of the EGE PUL in *B. intestinalis*. a** In the absence or during low amounts of complex arabinoxylan (↓), the *Bacteroides* spp. containing the EGE PUL express basal levels of core polypeptides, i.e., the SusC/SusD transporter, the endo-acting enzyme (Bi1041-GH43), and the hybrid two-component system (HTCS), which regulates expression of the genes in the PUL. Under this condition, the helix-turn-helix motif of the HTCS is bound to a yet to be determined DNA sequence in the gene cluster to block transcription of the EGE-PUL. Note that binding of an HTCS to DNA has been demonstrated (Cann et al. unpublished). **b** In the presence of increased amounts of complex arabinoxylan (↑), increased degradation of the polysaccharide by Bi1041-GH43, and the transport of products into the periplasm lead to binding (sensing) of the products (branched oligosaccharides) by the sensor module of the HTCS[3, 98]. The sensor domain of the HTCS sends a signal to the C-terminal helix-turn-helix (DNA-binding domain of response regulator), triggering dissociation from the DNA. This allows the RNA polymerase to bind a promoter within the PUL and freely express the genes in the EGE PUL. Other than Bi1041-GH43, the signal peptides present in the rest of the enzymes in the EGE PUL suggest that they are periplasmic in location. In such a model, the ferulic acid will be cleaved in the periplasmic space and likely pumped extracellularly by an efflux pump. **c** In an alternate hypothesis, some of the gene products predicted to be periplasmic in location are found to be extracellularly located, as observed in previous reports[47, 48]. We hypothesize, as shown in "**c**" that during upregulation of EGE PUL expression, a ferulic acid esterase (e.g., Bi1040-FAE) localizes extracellularly and cleaves off the ferulic acid side chains, resulting in the accumulation of the phenolic compound in the spent medium. The ferulic acid made available extracellularly in either "**b**" or "**c**" is metabolized by both members of the microbiome and the host, with the potential for immune modulation in the host.

*B. intestinalis*. Signal peptides and lipoprotein signal sequences were predicted using SignalP v5.0 (http://www.cbs.dtu.dk/services/SignalP/) and LipoP v1.0 (http://www.cbs.dtu.dk/services/LipoP/), respectively. The gene for each putative esterase (BACINT_01033-CE1, BACINT_01034-h1, BACINT_01038-CE6/CE1, BACINT_01039-CE1, BACINT_01040-h2, and WP_050793236) and each putative glycoside hydrolase (BACINT_01035-GH43, BACINT_01041-GH43, BACINT_01042-GH3, and BACINT_01043-GH43) was PCR-amplified from the genomic DNA of *B. intestinalis* DSM 17393 or *Bacteroides eggerthii* 1_2_48FAA, using the NEB Phusion® High-Fidelity DNA Polymerase. The primer pairs for the PCR amplification of each gene is listed in Supplementary Table 3. The PCR products were digested with NdeI and XhoI and cloned into the pET-28a vector, digested with the same restriction enzymes. The products of the ligation were transformed into NEB® 5-alpha Competent *E. coli* cells, and the transformants were selected on Lysogeny Broth (LB) agar plates supplemented with 100 μg/mL

ampicillin. The DNA inserts in the recombinant plasmids were confirmed by nucleotide sequencing (W. M. Keck Center for Comparative and Functional Genomics, University of Illinois at Urbana Champaign) and introduced into *E. coli* BL21-CodonPlus (DE3) RIL competent cells by heat shock transformation. Transformants were cultured overnight on LB agar plates supplemented with ampicillin (100 μg/ml) and chloramphenicol (50 μg/ml) at 37 °C. A single colony for each gene was picked and inoculated into 10 ml LB medium containing the same antibiotics at the same concentrations as the agar plates and incubated at 37 °C with vigorous shaking (250 rpm/min) for 6 h. The 10 ml cultures were then used to inoculate a 1 liter LB medium in a 2.8 liter Fernbach flask and culturing continued at 37 °C until the absorbance at 600 nm (OD$_{600}$) reached 0.3–0.6. Expression of the recombinant proteins were initiated by adding isopropyl β-D-thio-galactopyranoside (IPTG) to the culture at a final concentration of 0.1 mM. The culturing was continued for 16 h at 16 °C with shaking at a speed of 225 rpm/min.

The cells were harvested by centrifugation, resuspended in ice-cold binding buffer (pH 7.5, 50 mM Tris-HCl, 300 mM NaCl) and cell contents released by passage through an EmulsiFlex C-3 cell homogenizer (Avestin, Ottawa, Ontario, Canada) as previously described[25]. Each clarified supernatant, prepared through centrifugation at 12,857×g for 30 min at 4 °C, was loaded onto a HisTrap™ 5 mL column (GE Healthcare, Piscataway, NJ) fitted to an ÄKTAxpress fast protein liquid chromatography (FPLC) system. The column was washed with five column volumes of binding buffer supplemented with 10 mM imidazole. The bound proteins were eluted with a gradient of elution buffer (pH 7.5, 50 mM Tris-HCl, 300 mM sodium chloride, 500 mM imidazole) and 1.5 ml fractions collected and resolved through 12% sodium dodecyl sulfate-polyacrylamide gel electrophoresis (SDS-PAGE). The protein bands were visualized by staining with Coomassie brilliant blue G-250. The purest fractions were pooled, concentrated into 2 mL volumes and centrifuged at 25,000×g for 5 min to precipitate any denatured proteins. The recombinant proteins were further purified through size exclusion chromatography by applying to a HiLoad 16/60 Superdex 200 column (GE Healthcare, Piscataway, NJ). The chromatography was developed with a buffer composed of 50 mM Tris-HCl, 150 mM sodium chloride, pH 7.5. Eluted fractions were analyzed by 12% SDS-PAGE, and the highly purified protein fractions were used for enzymatic assays.

**Carbohydrate esterase activities on complex substrates**. The release of ferulic acid from the complex substrates, i.e., insoluble wheat arabinoxylan and sugar beet pulp, were monitored by incubating different concentrations (0.5, 1.0, and 1.5 µM) of each putative carbohydrate esterase with 10 mg/ml of each substrate in a buffer composed of 50 mM sodium citrate, 150 mM NaCl, pH 6.5 with the substrate pre-swollen for 10 min at 80 °C. The reactions were carried out for 2 h at 37 °C with shaking at 1200 rpm, after which each sample was heated at 99 °C for 10 min to inactivate the enzyme, cooled to room temperature, and one volume of methanol added and vortexed for 30 s. The reaction mixture was then centrifuged at 15,000 rpm for one minute, and the clear supernatant was analyzed by an High-Performance Liquid Chromatography with Pulse Amperometric Detection (HPLC-DAD) system (LC-20 AD pumps, SIL-20A autosampler, SPD-M20A PDA detector; Shimadzu, Kyoto, Japan) equipped with a Luna C18 column (250 mm × 4.6 mm i. d., 5 µm particle size; Phenomenex, Torrance, CA). The flow rate was 0.5 ml/min, and the following gradient composed of (A) water with 0.1% formic acid and (B) acetonitrile with 0.1% formic acid was used at 45 °C: 0–20 min, isocratic 85% A and 15% B; 20–35 min, linear to 100% B; 35–40 min, isocratic 100% B; and 40–45 min, isocratic 85% A and 15% B. UV detection was carried out at 325 nm, and a standard curve was used to determine the ferulic acid content of the hydrolysates[28].

**Feruloylated arabinoxylan oligosaccharides**. The preparation of de-starched wheat bran (DWB) and purification of the feruloylated oligosaccharide mixture was done following a previous protocol[28]. In brief, to prepare the DWB, 60 g of milled wheat bran (<0.5 mm) were suspended in 600 ml of phosphate buffer (80 mM, pH 6.2) and incubated with 4.5 ml of thermostable α-amylase (Termamyl 120 L, EC 3.2.1.1, from *Bacillus licheniformis*, 20,000–60,000 U/mL) at 92 °C for 20 min. The insoluble fraction was obtained by centrifugation at 4000 rpm for 10 min, washed twice with water, twice with ethanol, and twice with acetone, and dried at 40 °C. For the preparation of feruloylated oligosaccharides from the DWB, 30 g of the de-starched wheat bran were suspended in 3 L of water and autoclaved for 40 min. Subsequently, 1.5 g of Driselase enzyme preparation was added to the suspension and incubated at 37 °C at 80 rpm for 48 h. The mixture was then heated at 95 °C for 10 min and any undigested material was removed by centrifugation at 4000 rpm for 30 min. The non-phenolic products were removed by passage of the mixture through an Amberlite XAD-2-column (bed volume: 55 × 2.6 cm³), which was preconditioned with methanol and water. The hydrolysate was applied to the column, which was then washed with 500 ml of water. The feruloylated oligosaccharides were eluted using 500 ml of methanol/water (50:50, v/v). Subsequently, the methanol was removed by evaporation, and the aqueous eluate was freeze dried.

**Preparation of acetylated xylan**. The acetylated xylan was prepared in accordance with Fontana[33]. In summary, oat spelt xylan (Sigma Aldrich) was slowly added to 250 ml of dimethyl sulfoxide with gentle stirring at room temperature until 4% wt/volume was achieved. After the xylan was evenly suspended, the preparation was heated at 55 °C for 20 min to solubilize the mixture. Two hundred milliliters of acetic anhydride, pre-heated to 60 °C, was added with stirring over 5 min. The reaction mixture was placed in a dialysis tube and dialyzed for 5 days under running water at 4 °C. After further dialysis for 24 h against double-distilled $H_2O$, the substrate was freeze dried and used as the acetylated xylan substrate.

**Determination of esterase activity and specificity**. The substrate specificity of each esterase was assessed by incubating 200 µM of each feruloylated oligosaccharide (FA, FAX, FAXX, FAA, FAAA, FGG, and FAXG) with 50 nM of each putative carbohydrate esterase (BACINT_01033-CE1, BACINT_01038-CE6/CE1, BACINT_01039-CE1, and BACINT_01040-h2) in citrate buffer (50 mM sodium citrate, 150 mM NaCl, pH 6.5) at 37 °C for 10 min. The enzymes were inactivated by heating the reaction at 95 °C for 10 min, and the solutions were analyzed by HPLC-DAD as described above for esterase activities towards complex substrates.

The feruloylated oligosaccharides and the end products of hydrolysis were detected at an absorbance of 325 nm. For the acetyl-xylan esterase activity, the acetylated oat spelt xylan (0.5% wt/vol) was incubated with each putative carbohydrate esterase in citrate buffer pH 6.5 at 37 °C for 1 h, and the released acetic acid was measured using an acetic acid detection kit (Megazyme, Bray, Ireland) according to the manufacturer's instructions. The formation of NADH was monitored continuously at an absorbance of 340 nm using a Synergy 2 microplate reader (BioTek, Winooski, VT) with a path length correction feature. For the hydrolysis of the artificial substrates, methyl-ferulate and *para*-nitrophenol (*p*NP) acetate, 50 nM of enzyme was incubated with 1 mM of each substrate in 0.1 M MOPS buffer pH 7.5 at 37 °C. The amount of released *p*NP was continuously monitored at 400 nm and calculated by using a standard curve derived from various concentrations of *p*NP. The degradation of methyl-ferulate was continuously monitored at 350 nm, and a standard curve derived with various concentrations of methyl-ferulate was used to determine the concentrations[28].

**Optimal pH and temperatures of esterases**. The optimal pH of the esterases BACINT_01033-CE1, BACINT_01038-CE6/CE1, BACINT_01039-CE1, and BACINT_01040-FAE were determined by incubating each enzyme at 50 nM final concentration with 2 mM *p*NP acetate at 37 °C at a pH range of 4.0–6.5 (50 mM sodium citrate and 150 mM NaCl) and 6.0–8.5 (50 mM sodium phosphate and 150 mM NaCl), and the released *para*-nitrophenol was continuously monitored at an absorbance of 400 nm using a Cary 300 UV–vis spectrophotometer (Agilent, Santa Clara, CA). The optimal temperature of each enzyme was determined in its corresponding optimal buffer at temperatures ranging from 20 °C to 75 °C.

**Substrate specificities of the glycoside hydrolases**. To investigate the enzymatic specificities of the four glycoside hydrolases in the esterase genes-enriched cluster, the purified polypeptides were incubated with different polysaccharide substrates (wheat arabinoxylan, rye arabinoxylan, and glucuronoxylan) individually at pH 6.5 and a temperature of 37 °C at a final enzyme concentration of 500 nM. Substrates concentrations were at 0.5% wt/vol. Aliquots were removed after 16 h from the reaction mixtures, and the concentrations of reducing ends released by the enzymes were assayed using the *para*-hydroxybenzoic acid hydrazide assay[58]. For BACINT_01042-GH3 and BACINT_01043-GH43, a similar end point assay was performed against xylo-oligosaccharides (X2, X3, and X4), arabinoxylan oligosaccharides (A²X and A³XX), and insoluble WAX obtained from Megazyme (Bray, Ireland), and the end products were analyzed by High-Performance Anion Exchange Chromatography with Pulse Amperometric Detection (HPAEC-PAD) on an ICS-5000 system (Thermo Scientific Dionex, Sunnyvale, CA) equipped with a CarboPac PA-100 column (250 mm × 2 mm; Thermo Scientific Dionex)[28].

**Synergistic activities of the EGE PUL associated enzymes**. To determine if the esterases and the GH enzymes function in synergy, the enzymes were analyzed in a combinatorial fashion for their hydrolysis of insoluble wheat arabinoxylan and de-starched wheat bran at 0.5% wt/vol final concentration, at pH 6.5 and temperature of 37 °C for 14 h. The ferulic acid and reducing sugar ends released were then determined as described above for reducing ends and also by high-performance liquid chromatography (HPLC). Briefly, the monomeric and oligomeric end products were analyzed by HPAEC-PAD on an ICS-5000 system (Thermo Scientific Dionex, Sunnyvale, CA) equipped with a CarboPac PA-100 column (250 mm × 2 mm; Thermo Scientific Dionex). The flow rate was 1 mL/min, and a gradient composed of the following eluents was used at 25 °C: (A) 0.1 M sodium hydroxide, (B) 0.1 M sodium hydroxide + 1 M sodium acetate. Before every run, the column was washed with 100% A for 10 min. After injection, the following gradient was applied: 0–5 min, isocratic 100% A; 5–24 min linear gradient to 50% A and 50% B; 24–34 min 100% B. Standard curves were constructed for arabinose, xylose, xylobiose, xylotriose, xylotetraose, xylopentaose, and xylohexaose to calculate the concentrations of the end products in the reaction mixture.

**Quantitative reverse transcription-PCR (qRT-PCR)**. The bacterial cells (*B. intestinalis*, *B. oleiciplenus*, and *B. cellulosilyticus*) were grown in 5 mL of minimal medium containing the polysaccharide of interest [(i.e., soluble wheat arabinoxylan (sWAX), feruloylated arabinoxylan oligosaccharides (FAOX), de-starched wheat bran (DWB), and insoluble wheat arabinoxylan (InWAX)] and cultured to mid-log phase. The cells were collected by centrifugation and rapidly mixed with RNAprotect bacterial reagent, as described by the manufacturer and used for subsequent RNA extraction. RNA was extracted using the RNeasy mini kit (Qiagen, Valencia, CA) according to the manufacturer's instructions. Contaminating DNA was removed in-column using NEB DNase I (Ipswich, MA), and the quality of the RNA was assessed by using a Bioanalyzer 2100 with a RNA 6000 Nano Assay Reagent kit from Agilent (Santa Clara, CA). The RNA quantity was determined by nanodrop (Agilent) and Qubit (Thermo Fisher Scientific) methods. The qRT-PCR was performed using a Thermo light cycler 480 (Roche, San Francisco CA) and SYBR green master mix (Roche, San Francisco, CA). The cDNA was generated from 1 µg total RNA using a transcriptor High-fidelity cDNA synthesis kit according to the manufacturer's instructions (Roche, San Francisco, CA). The cDNA was 100× diluted for qRT-PCR (corresponding to a starting amount of 10 ng RNA). The relative expression of the *susC/susD* gene pair (in the

esterase genes-enriched PULs) of *Bacteroides intestinalis*, *Bacteroides cellulosilyticus* and *Bacteroides oleiciplenus* during growth on the substrates (sWAX, FAOX, DWB, and InWAX) compared to their component sugars (xylose:arabinose mixture) were determined for the cells. The fold-change in gene expression (xylan substrates *versus* monosaccharides mixture) was calculated from three biological replicates, using the 16S rRNA reference gene for normalization. The primer pairs for the qRT-PCR of each gene is listed in Supplementary Table 4. Log2 fold-change at mid-log phase were considered as significantly different at $p < 0.01$ (Student's *t* test).

**Transcriptional analysis.** Cells grown in the monosaccharides mix (xylose:arabinose) and insoluble wheat arabinoxylan were collected in two volumes of RNAprotect Bacteria Reagent (Qiagen, Germantown, MD). Cells were pelleted by centrifugation at $10,000 \times g$ for 10 min and resuspended in lysis buffer (10 mM Tris, 1 mM EDTA, 1 mg/ml lysozyme, 0.1 mg/ml Proteinase K) for 10 min. Total microbial RNA was extracted using the RNeasy Mini Kit from Qiagen following the manufacturer's protocol with an on-column DNase treatment step. RNA quality was assessed using an Agilent 2100 Bioanalyzer (Agilent, Santa Clara, CA) with all samples achieving a RNA Integrity Number (RIN) of 9 or higher. The High-Throughput Sequencing and Genotyping Unit of the Roy J. Carver Biotechnology Center at the University of Illinois at Urbana Champaign performed high-throughput sequencing using an Illumina 4000 sequencing platform.

The resulting sequence data was analyzed for differentially expressed genes following a previously published protocol[59]. Briefly, reads were filtered for quality using Trimmomatic v0.33[60]. Reads were aligned to the genome using BowTie2 v2.3.3.1[61]. Reads mapping to gene features were counted using htseq-count (release_0.9.1)[62]. Differential expression analysis was performed using the edgeR v3.6 package in R v3.4.2 (with the aid of Rstudio v1.1.383). The TMM method was used for library normalization[63]. Coverage data was visualized using Integrated Genome Viewer (IGV)[64].

**Comparative genome analysis.** To determine the prevalence of the esterase genes-enriched (EGE) PUL, a comparative genomic method was used to search the genomes of more than 20 *Bacteroides* common to the human gut, using Geneious version 9.0.2. for a genome-wide alignment. The *B. intestinalis* genome was used as the reference genome to identify the potential EGE PUL.

**Growth and end products analyses.** The capacity of the *Bacteroides* spp. to grow on the complex insoluble substrates, i.e., de-starched wheat bran (DWB), wheat bran (WB) and insoluble wheat arabinoxylan (InWAX) was determined through calculations of the initial and final protein concentrations and the fermentation end products in the medium. The total protein quantification was performed using a Bradford microtiter assay (Biorad, Hercules, CA) according to the manufacturer's instructions, and the fermentation end products (initial and final supernatant acetate, succinate, and propionate concentrations) with a high-performance liquid chromatography (HPLC, Agilent Technologies 1200 Series, Mississauga, CA) instrument equipped with a refractive index detector and fitted with a Rezex ROA-Organic Acid H$^+$ (8%) column (Phenomenex Inc., Torrance, CA). Elution was carried out with 0.005 N H$_2$SO$_4$ at a flow rate of 0.6 ml/min at 50 °C.

**Crystallography studies.** The crystallization conditions for BACINT_01033-CE1, BACINT_01039-CE1, and BeGH43/FAE (WP_050793236) were evaluated by using the sparse matrix sampling method and commercial screens. For preparation of seleno-methionine-labeled BACINT_01033-CE1 and BACINT_01039-CE1, transformed *E. coli* BL21 (DE3) cells were grown in M9 medium (containing 1 mM MgSO$_4$, 0.4% glucose, 0.5 mg/L thiamine, and 100 μg/mL ampicillin) at 37 °C to an OD$_{600}$ of 0.4. Following the addition of an amino acid mixture (lysine, threonine, phenylalanine, leucine, isoleucine, and valine) and seleno-methionine, protein overexpression was induced by adding IPTG[65]. The cultures were then grown for 24 h at 16 °C, followed by purification of the protein as described above.

Two crystal forms of seleno-methionine-labeled BACINT_01033-CE1 were obtained, both via sitting drop vapor diffusion at room temperature. The protein was concentrated to 20 mg/ml in 50 mM Tris pH 7.5, 150 mM NaCl prior to crystallization. Crystals of the seleno-methionine-labeled BACINT_01033-CE1 of the P6$_4$22 space group were obtained from the Hampton Research SaltRx screen in condition 22 [1.2 M Sodium citrate tribasic dehydrate, 0.1 M Tris pH 8.5]. Crystals of the seleno-methionine-labeled BACINT_01033-CE1 of the P3$_1$2 space group were prepared by sitting drop vapor diffusion from Hampton Research PEGRx condition 20 [0.1 M BIS-Tris pH 6.5, and 20% PEG1.500]. Both crystal forms were flash frozen by plunging into liquid nitrogen after a brief soak in cryoprotectant comprised of 80% of the original crystallization solution plus 20% ethylene glycol. X-ray diffraction maxima were collected at the Life Sciences Collaborative Access Team beamline 21 (ID-F) at the Advanced Photon Source in Argonne, IL. X-ray data were processed and scaled with Xia2[66] within the CCP4[67] suite of programs. AutoSol[68] within the Phenix[69] suite was used to locate the positions of the selenium atoms, and Autobuild[70] was used to create the initial models of the proteins. Alternate rounds of manual model building in Coot[71] and refinement with Phenix.refine was used to complete the structures.

Crystals of the seleno-methionine-labeled BACINT_01039-CE1 were prepared by using the hanging drop diffusion technique. The protein at 20 mg/mL in 50 mM Tris,150 mM NaCl (pH 7.5) was mixed 1:1 with a precipitant solution from Hampton Research Crystal screen condition 41 [10 mM ammonium sulfate, 80 mM Sodium cacodylate pH 6.5, and 21% PEG4000]. Crystals were briefly soaked in a cryoprotectant comprised of 80% of the crystallization solution plus 20% ethylene glycol and flash frozen by plunging into liquid nitrogen. X-ray diffraction maxima were collected at the Life Sciences Collaborative Access Team beamline 21 (ID-D) at the Advanced Photon Source in Argonne, IL. X-ray data were processed and scaled with HKL2000[72]. AutoSol within the Phenix suite was used to locate the positions of the selenium atoms, and Autobuild was used to create the initial models of the proteins. Alternate rounds of manual model building in Coot and refinement with Phenix.refine was used to complete the structures.

The crystals of BeGH43/FAE were prepared by using the hanging drop diffusion technique. The protein at 10 mg/mL in 50 mM Tris,150 mM NaCl (pH 7.5) was mixed 1:1 with a crystallization solution comprised of 0.2 M AmSO4, 0.1 M MES pH 6.5, 30% PEG 5000 at room temperature. Initial crystals from this solution were then streak seeded back into the same conditions to obtain larger single crystals. Crystals were flash frozen in liquid nitrogen after briefly incubating in a cryoprotectant comprised of a solution containing 80% crystallization solution and 20% ethylene glycol. X-ray diffraction maxima were collected at the Life Sciences Collaborative Access Team (LS-CAT) beamline ID-F at the Advanced Photon Source in Argonne, IL. X-ray data were processed and scaled in HKL2000. The structure was determined via molecular replacement using MrBUMP[73] and Phaser[74] in the CCP4 program suite, using PDB 3ZXL[36] as the initial model. AutoBuild within Phenix was then used to automatically build the structure, followed by several rounds of manual model building in Coot and refinement with Phenix refine. Data collection and refinement statistics are presented in (Supplementary Table 1).

**Multiple sequence alignments.** The esterases with the structures determined in the present study were used to search the Genbank protein database (https://www.ncbi.nlm.nih.gov/protein), and the polypeptide sequences of high similarity were retrieved. The homologous polypeptides were then aligned using Clustal Omega Multiple Sequence Alignment program (https://www.ebi.ac.uk/Tools/msa/clustalo) and manually shaded.

**Site-directed mutagenesis.** Mutagenesis was performed using a Q5 site-directed mutagenesis kit from New England Biolabs (Ipswich, MA). First, mutagenesis primers were designed using the NEBaseChanger tool (New England Biolabs). For each mutation, a pair of primers were designed (Supplementary Table 5). The first primer or mutagenic primer was designed with the desired mutation in the center of the primer and 10 bases of correct sequence on either side. The second primer contained no mismatch, and it begins at the base next to the 5′ end of the mutagenic primer and proceeds in the opposite direction. Reaction mixtures were prepared according to the manufacturer's recommendation with pET28-BeGH43/FAE and pET28-BACINT_01033-CE1 and pET28-BACINT_01039-CE1 as the DNA templates for generation of the corresponding mutants. After cycling of the reaction mixture 25 times in a thermal cycler, the mixture was treated with kinase, ligase and DpnI (KLD) mixture. The resulting DNA was transformed into chemically competent NEB 5-alpha *E. coli* by heat shock. Recombinant plasmids were extracted, and their gene inserts with mutations were sequenced to confirm the presence of the desired mutations. Expression of the wild-type and mutant recombinant proteins was performed as described above. Cells were harvested by centrifugation, resuspended in ice-cold equilibration buffer (50 mM sodium phosphate, 300 mM NaCl, pH 7.4), and lysed as described above. Cell-free extract was obtained by centrifugation at 12,000×g for 30 min at 4 °C and mixed with 1 ml of equilibrated Talon resin (Takara Bio USA). The mixture was incubated at 4 °C for 20 min with gentle shaking. The resin with bound proteins was collected by centrifugation (700×g, 2 min) at room temperature and washed twice with 10 ml of equilibration buffer and the proteins were eluted with elution buffer (50 mM sodium phosphate, 300 mM NaCl, pH 7.4). The eluted proteins were analyzed by 12% SDS-PAGE and visualized by Coomassie blue staining to confirm their purity and used for circular dichroism analysis and enzymatic assays.

**Circular dichroism (CD) spectra.** Determination of CD spectra for the BeGH43/FAE, BACINT_01033-CE1 and BACINT_01039-CE1 wild-type proteins and their site-directed mutant proteins was carried out using a J-815 CD spectropolarimeter (Jasco, Tokyo, Japan). Protein samples were prepared at a concentration of 0.2 mg/ml in 10 mM potassium phosphate buffer (pH 7.5)[75]. For the measurements, a quartz cell with a path length of 0.1 cm was utilized. CD scans were carried out at 25 °C from 190 nm to 260 nm at a speed of 50 nm/min with a 0.1-nm wavelength pitch, with five accumulations. Data files were analyzed on the DICHROWEB online server (http://dichroweb.cryst.bbk.ac.uk/html/home.shtml) using the CDSSTR algorithm with reference set 4, which is optimized at 190 nm to 240 nm[76].

**Molecular dynamics simulation set-up.** The 3-D coordinates of esterase domains were acquired from the crystal structures of the three esterase proteins reported in this study (Bi1033-CE1 residue range: 126–382, Bi1039-CE1 residue range:

126–383, and the BeGH43/FAE residue range: 524–796 or FAE module). The engineered seleno-methionine (MSE) residues were mutated back to original Methionine residues. The three missing residues (227–229) of protein Bi1039-CE1 were added using MODELLER v9.21[77]. The proteins were solvated in an orthogonal TIP3P water box[78]. To make the systems neutral, 150 mM NaCl was added using PACKMOL v18.169[79]. To observe the binding of Ferulic Arabinose (FA) to Bi1039-CE1 and the FAE module of BeGH43/FAE, two separate systems were built with 8 mM concentration of FA. The FA molecule was created using MarvinSketch v19.23 (ChemAxon, 2019, https://chemaxon.com/products/marvin) and parameterized using AMBER GAFF forcefield[80]. Amber ff14SB forcefield[81] was used in modeling esterase enzymes. The integration time-step was chosen as 4 fs via hydrogen mass repartitioning (HMR)[82], which redistributes mass between hydrogen atoms and neighboring covalently bonded atoms of protein to allow simulations with a 4 fs time-step.

The simulations were performed using Amber18 software package[83]. Before the production run, all the systems were subjected to minimization and equilibration[84]. Energy minimization was performed for 15000 steps with 5000 steps of conjugate gradient minimization and remaining steps with the steepest descent algorithm. The systems were heated from 0 K to 300 K in NVT ensemble. The heating step was conducted for 5 ns using Langevin thermostat[85]. Protein backbone atoms were constrained using a spring force (in form of $k(\Delta x)^2$)[83] and movement of hydrogen atoms were constrained by SHAKE algorithm[86]. The systems were then simulated for 5 ns in a NPT ensemble with a pressure of 1 bar applied using Berendsen barostat[87] and constrained backbone atoms. In the last equilibration step, all the constraints were removed, and the systems were allowed to equilibrate for 20 ns at 300 K and 1 bar in NPT ensemble.

For each of the three esterase (without FA) systems, 2 μs of single-long simulated trajectory was obtained. For each of the ligand binding systems, 10 μs of aggregate simulation was performed using an adaptive sampling approach[88]. This method improves sampling efficiency by reducing the amount of time system spent in local minima. In this method, multiple short simulations are performed in parallel to explore the conformational landscape. The starting frames of each round of simulation are obtained through clustering of data from the previous round of simulations using K-means algorithm[89] and selecting frames from the least-populated clusters. For the binding simulation, FA distance from the catalytic residues was used to featurize the simulation data and obtained least-populated clusters. Finally, the sampling bias in the simulation was removed by constructing a Markov state model (MSM)[90]. MSM connects the clusters (or conformational states of the protein) by calculating a transition probability matrix between different conformational states of the protein to remove the sampling bias. The eigenvalues and eigenvectors of the transition probability matrix provide the estimates of the timescales and the population shifts caused by the different conformational transitions in the protein. The eigenvector corresponding to the largest eigenvalue provides the equilibrium population of the conformational states[91].

**Markov state model construction**. The python package PyEMMA 2.5.6[92] was used to build MSMs. Residue pair distances and residue-ligand distances were used to featurize the simulation data. Number of clusters for the MSM were optimized by maximizing the VAMP score[93]. The lag time of 5 ns was chosen from the implied time scale plot and Chapman-Kolmogorov test[90] was performed to validate the Markovian behavior of the MSMs.

**Trajectory analysis**. Analysis of trajectory data was performed using the CPPTRAJ v18.01[94] module in AMBER18 and the python package MDTraj v1.9.3[40]. The 2-D plots were generated using the python package Matplotlib[95]. MD snapshots were visualized and analyzed using VMD v1.9.3[96] and pyMOL v2.1 (https://pymol.org/2/). Pocket volume calculations were performed by python package POVME v3.0[97]. To calculate the volume, we selected the center of mass of HIS (Cβ) and SER (Cβ) atoms as a center and computed the accessible volume within 5 Å

**Statistical analysis**. Data were expressed as the mean ± standard deviation of the means (S.D.). The statistical difference among three variables or more by one-way Analysis of Variance (ANOVA, F-test $p < 0.05$ declared significant) with multiple comparisons using Tukey's test. Statistical analyses were performed using the Graph-pad Prism v8.4.3 (Graph-pad Software, San Diego, CA).

**Reporting summary**. Further information on research design is available in Nature Research Reporting Summary linked to this article.

## Data availability

RNASeq data generated from this study are deposited at NCBI GEO under accession number GSE161471 (raw reads and differential expression files). Sequence Read Archive (SRA) entries containing raw reads of the individual samples can be found at BioProject PRJNA678359. Protein Databank (PDB) IDs of the structures presented in this study are as follows: 6MOU (BACINT_01033-CE1 dimer), 6MOT (BACINT_01033-CE1 monomer), 6NE9 (BACINT_01039-CE1), and 6MLY (BEGH43/FAE). PDB entries

analyzed in this study include: *Humicola insolens* GH43 glycoside hydrolase PDB 3ZXL, *Geobacillus thermoleovorans* beta-1,4-xylosidase PBD 5Z5I, *Lactobacillus plantarum* tannase PBD 4JUI and *Clostridium thermocellum* xylanase 1JT2. Other data supporting the findings of this study are available within this manuscript and its supplementary files. All datasets and recombinant plasmids generated and/or analyzed during the current study are available from the corresponding author on reasonable request. Source data are provided with this paper.

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

## Acknowledgements

The research was supported by the National Institute of General Medical Sciences of the National Institutes of Health under Award Number RO1GM140306 to R.I.M., N.M.K., and I.C., and in part by funding from the Microbiome Metabolic Engineering Theme (MME) of the Carl R. Woese Institute for Genomic Biology, University of Illinois at Urbana Champaign. The funders had no role in the study design, data analyses, decision to publish, or preparation of the manuscript. I.C. would like to acknowledge support during sabbatical by the Top Global University Program, Kyoto University, Japan.

## Author contributions

G.V.P.: study design for biochemical, transcriptomics, structural studies and preparation of the manuscript; A.M.A.: biochemical studies and preparation of manuscript; S.D.: molecular dynamics simulations; C.N.D.G.: preparation of manuscript, D.W., J.A.F., and S.B.: biochemical studies and provision of reagents; Z.W.: structural studies; H.A.: mutagenesis studies and interpretation of data; R.I.M.: design of growth experiments and transcriptomics analysis; E.C.G.: interpretation of data, preparation and correction of the manuscript; D.S.: molecular dynamics simulations, interpretation of data and preparation of the manuscript; N.M.K.: structural studies, interpretation of data, and preparation of manuscript; I.C.: overall design of the study, analyses of the data and interpretation, preparation, and correction of the manuscript.

## Competing interests

The authors declare no competing interests.

## Additional information

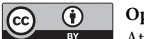

