## [Peer Review File · Nature Communications]

Reviewers' comments:

Reviewer #1 (Remarks to the Author):

Pereira et al. have discovered a PUL involved in sensing and cleaving off side-chains found on arabinoxylans. These esterases vary in their structure and substrate-specificity, with one having broad activities. They then go on to show that bacterial spent media following incubations with arabinoxylans induces inflammatory cytokines in cell lines and mouse models.

I found the first few sections far stronger than the link to immunity, which seems premature given the data and a bit confusing given the literature on the anti-inflammatory effects of ferulic acid. This literature is mentioned but the conflict with the current data is addressed. One explanation might be that the concentrations used are not physiologically relevant. Another might be due to the oral delivery of ferulic acid (and other metabolites) which is not the typical site of production.

The immunology studies are not sufficiently controlled, especially given the elegant genetic tools used earlier. Isogenic strains with or without specific esterases are needed to control for the effects of other components of the spent media. Ideally, mono-colonized mice would be used to avoid the issues of oral delivery of microbial products. Even if this were done, I'd be concerned about downstream effects of these enzymes, since presumably they boost the efficiency of microbial fermentation leading to other end products that might have effects on host immunity independent of ferulic acid.

I've also never been a huge fan of using Caco-2 cells (a cancer cell line) to study bacterial interactions with the immune system. The authors might consider DC or T cell-based assays, which are likely more physiologically relevant. The same issue is true for the HIEC-6, which are small intestinal not immune cells.

Minor points:

Line 59 – typo – “nutrients acquisition”

Line 299 – I'm not sure this data can be used to conclude that ferulic acid is not metabolized. Maybe accumulates during growth instead?

Figure 1b-d needs stats.

Figure 2 – the upregulation is striking, how does this look genome-wide? Is this the most up-regulated PUL?

Figure 3c – why are there multiple bands in some of the protein lanes?

Figure 3d – Any idea what's going on with Bi1040????

Figure 5 – bacterial genetics would really help make this more definitive.

Figure 6 – a lot of these comparisons aren't significant, especially for “WB”. The fact that changes are seen in spleen and lymph node are consistent with the concern that oral gavage leads to rapid absorption of these compounds, inducing systemic inflammation that might not be seen with the same level of compound in the colon.

Reviewer #2 (Remarks to the Author):

The paper by Pereira et al describes the identification of a predicted carbohydrate esterase enriched PUL from a gut *Bacteroides* spp that is only expressed on 'complex' arabinoxylans (defined as the fraction of purified arabinoxylan that has ferulate cross links). The carbohydrate active enzymes from the *B. intestinalis* EGE PUL are biochemically and structurally characterised and several are shown to act together to release ferulic acid from complex xylans. Growth of *B. intestinalis* on complex xylans results in a build-up of free FA and the authors use cell lines and a mouse model to test the effect of the released FA on intestinal cell immune function. While the proposed effect of FA on intestinal cells has been described, the finding that FA is released from dietary xylans by prominent gut bacteria is both novel and interesting. I have however some comments to be addressed.

Main comments:

1. For both the cell line and mouse experiments the effect of spent media, while significant in some cases, seems very small making me question if it is real. In addition a key control of spent media from Ara or Xyl and/or simple WAX grown cells (ie. no FA present) is missing – this is needed to be sure the effect seen is due to FA released from xylans as proposed and not just say SCFAs and/or LPS that would be present in all spent media samples.
2. Line 111-112. How do you know the EGE PUL is required for degradation of complex AX? What data is this statement based on?
3. In Table S7 it shows that at least one of the simple WAX PULs is also upregulated on complex WAX (InWAX - Bi04193-04205). The authors should comment on this and what this might mean for the degradation of InWAX as the MS is currently written from the perspective that only the EGE PUL is involved in degradation of complex WAX.
4. In relation to this in Table S7 it is shown that there are multiple esterases (both CE1 and CE6) in the other two simple AX PULs. Do the authors know what the roles of these enzymes are? Why have esterases in these xylan PULs if not to de-esterify xyans?
5. All locus tags should be further annotated throughout the MS (text and Figs) with Cazy family or activity for clarity (e.g. Bi1035-GH43) otherwise the reader has to constantly refer to Fig3a to have any idea what activity the locus tag refers to.
6. Lines 185-188. Clarify why only 1042 was tested against xylo-oligosaccs and 1043 against InWAX by HPAEC-PAD? Was it because they showed low levels of reducing sugar release against xylans compared to 1035 and 1041? Ideally test all enzymes against xylo-oligos and WAX by HPAEC-PAD or at least explain why only certain enzymes assayed.
7. Line 190. The data does not show '...enzymes function to completely hydrolyze arabinoxylans.' The data show that the enzymes are able to hydrolyse arabinoxylans, but not necessarily to completion (ie. only free monosaccharides, acetate and FA remaining). Please reword or present data that shows the enzymes hydrolyse AX to completion.
8. Line 195. How do the GH43s (Bi1035 and 41) release free FA from arabinoxylan as shown in Fig 3f and g? (see 1035 and 1041 only lanes – small amount, but above control, of FA released with all except 1041 vs InWAX). Are you proposing that the enzymes are dual acting GH/esterases? Please provide an explanation for this activity or if it is an artefact of the assay used.
9. Line 200. How do you know these enzymes are endo-acting? Is it just because they release the most reducing ends from xylan of all the GHs? This is very circumstantial evidence. This should be clarified and ideally present some data that directly shows endo activity (e.g. HPAEC-PAD or TLC showing range of oligosaccharides being produced during digestion of AX).
10. Line 226. If the closest structural homologue of the GH43 is the double Ara specific *H. insolens* enzyme (3ZXL) have you tested to see if the *Be* enzyme is double or single Ara specific? The only data I can see is for InWAX (Fig S8d) which will have both double and singly substituted side chains so not clear from this.
11. It looks like several of the signal peptide predictions are wrong (see Fig 3b for enzymes proposed to have signal peptide) and therefore the model (Fig S18) proposed for complex xylan breakdown incorrect. For example Bi1035 has a strongly predicted Type I signal sequence (see below - using either LipoP or the new SigP v5) which would mean in *Bacteroides* spp it was much more likely to be periplasmic than on the cell surface. Furthermore there is no experimental data presented regarding the enzyme localisation and so the whole model is based on these predictions.

These need to be corrected and the model changed.

e.g. >Bi1035

```
MIMKTKAILLGTAALMFTLTSEKAIAQIGTPYIHPSTIMECDGKYYTFGTGGGGLISED
GWTWNGGGVVRPGGGAAPDAVKIGDRYLVAYSATGGGLGGGHSKVLTMWNKTLDPNSPDF
KYTEPIEVASSAYDEDCAIDAGLLLDPTDGRWLWSYGYFGFIRLVELDPATGKRMEGN
KEIDIAIDCEATDLIYRDGWYLLGTHGTCCDGNSTYNIVVGRSRKVTGPYVDNMGRKM
LEGGGKMVIAAGNRQTGPGHFGLFKVADGVEKMSCHFADFDRGGRSVLGIRPLLWKNW
PVAGEVFKEGTYEIESERRGYALELAVDFVRMEYTRHRFEKDDTPVVPLKPQTLEDVIG
TWPQKINVRIGDYMFRPHQKWITAVPNGGGYLGPPYKIVIEGTNRALAATAGAEIMT
VPEFTGAPEQLWRIDQLTDGTYRIMPKKVPGTDKELVLSVGDSTPSLGFDMNSDNSKW
NFRDH
```

LipoP output for Bi1035:

Sequence SpI score=14.7597 margin=14.960613 cleavage=26-27

Cut-off=-3

Sequence LipoP1.0:Best SpI 1 1 14.7597

Sequence LipoP1.0:Margin SpI 1 1 14.960613

Sequence LipoP1.0:Class CYT 1 1 -0.200913

Sequence LipoP1.0:Class TMH 1 1 -0.744742

Sequence LipoP1.0:Signal CleavI 26 27 14.6568 # EKAIA|QIGTP

Sequence LipoP1.0:Signal CleavI 23 24 10.6123 # LTSEK|AIAQI

Sequence LipoP1.0:Signal CleavI 24 25 7.30313 # TSEKA|IAQIG

Sequence LipoP1.0:Signal CleavI 21 22 7.20032 # FTLTS|EKAIA

Sequence LipoP1.0:Signal CleavI 22 23 4.65163 # TLTSE|KAIAQ

Sequence LipoP1.0:Signal CleavI 25 26 2.9738 # SEKAI|AQIGT

Sequence LipoP1.0:Signal CleavI 20 21 1.95987 # MFTLT|SEKAI

Sequence LipoP1.0:Signal CleavI 28 29 1.24524 # AIAQI|GTPYI

SigP v5 output for Bi1035:

Protein type: Likelihood

Signal peptide (Sec/SPI): 0.9907

TAT signal peptide (Tat/SPI): 0.0048

Lipoprotein signal peptide (Sec/SPII): 0.0037

Most of the enzymes appear to be Type I signals and therefore periplasmic. The only exception is Bi1041-Gh43 which is a predicted lipoprotein and so may be on the cell surface.

In this new model 1041 would be in the place of 1035 on the cell surface and most of the other enzymes in the periplasm, including the newly discovered FA esterase Bi1040. FA would therefore be released in the periplasm and have to be exported.

Bi1033-CE1 – SPI - periplasm

Bi1038-CE6/CE1 – SPI - periplasm

Bi1039-CE1 – SPI - periplasm

Bi1040-HP – SPI – periplasm. NB. N-term Met likely wrong here as database seq (starts MSLL) predicts no signal peptide (likely due to several Asn/Gln residues), but from downstream Met (MKLV) is well predicted Type I signal peptide.

>Bi1040

```
MSLLRFFLNQCQLFNKLN
```

```
MKLVFGFVCFALFYSFMSFGQITQWTDINYANDSLEGHKLDIYLPDGGQTEY
```

```
KVVVLIYGSAWFANNMKQMAFQAMGKPLLDGGFAVVSINHRSSGDAKFPAQINDVKA AVR FIRAHAD EYR
```

```
LDTSGITGFSSGGHLSLAGTTNGVKVYKVGDTMEDIENVDGCTSFSSRVDVAVDWFPGIDMTRMEN
```

```
CATTKGADSP EALIGGTPAEHMDVLTLLNPMYIDEKDPKFLVIHGADTVVPHCQSVFFKDTLSAKGR
```

```
LEEFITVPQGQHGPITFNEQTFKKMTEFFRKQAAMDLC PANITLVPHIRPGSKKNPPQKQFY TLY
```

Bi1041-GH43 – SPII – cell surface

Bi1042-GH3 – SPI - periplasm

Bi1043-GH43 – SPI - periplasm

12. Supplementary Fig 18. I have a few queries about the model (in addition to pt 11 above):

a. It is not clear on the fig what the role of each of the enzymes is. Please make clearer what each activity is – possibly by showing an arrow from the enzyme to the bond cleaved? Otherwise the reader is left trying to cross reference info from the rest of the paper to understand how you came to propose this model.

b. Why not show all enzymes? Where are Bi1033 and 1038? From your data (Fig 3e) 1038 releases the most acetate from xylan – should this not be shown?

c. With regard to this what is the rationale for multiple CE1 and CE6 enzymes in the PUL if only need a single xylan esterase and FA esterase to de-esterify complex xylan? Some comment on this should be made.

d. What is the outer membrane 'sugar transporter'? This seems to have appeared from nowhere. What is the evidence for this and why do you think the bacterium needs this activity? (import of surface released Ara?)

e. Line 1386. The identification of the branched oligos as the products sensed by the HTCS is likely correct based on previous studies (e.g. Martens et al 2011), but you should state this is speculation for this system.

Other comments:

13. Line 64 – what does 'Unique consortium' mean? Unique to what – humans or the colon? Rephrase this as I think you mean the consortium is unique to each individual at the species and strain level. But Bacteroidetes and Firmicutes common theme amongst most mammals.

14. Line 67. Clunky start. Change to 'best studied in the Bacteroidetes' ie. infers less studied in Firmicutes.

15. Line 75. Missing some key refs for HTCS as oligosaccharide sensors. Martens et al 2011 (PMID: 22205877) first showed the periplasmic domains of Bacteroides HTCS bind oligosaccharide ligands and this is the inducing signal for their cognate PUL, while Lowe et al 2012 (PMID: 22532667) showed the first structure of a Reg_prop class HTCS bound to its target ligand and proposed a mechanism for signal transduction in this class of sensor-regulators.

16. Line 79. Add Ze et al 2012 (PMID: 22343308) and Cartmell et al 2018 (PMID: 30349080) as key refs on cross feeding in the microbiota.

17. Line 92. Missing Rogowski et al 2015 (PMID: 26112186). Key ref on xylan utilisation by Bacteroides.

18. Lines 95-97. Vague. What is 'arabinoxylan lacking side chains' – do you mean linear xylan? This is insoluble unless short dp. Also what are '...all the signatures for complex arabinoxylan degradation'? Please clarify both terms.

19. Line 99 – what does 'with primary degrader signatures' mean? Explain.

20. Line 111. Define EGE PUL here – more than one predicted esterase? What family of esterases? Give locus tags.

21. Line 118-120. Fig S1 only shows structures of simple AX (sWAX) and insoluble AX (InWAX), but the text refers to FAOX and DWB. The authors should describe the structures of these other two polysacchs for clarity.

22. Fig 2 legend Line 893. Clarify that Sus-like proteins are named after the prototypic system but are not all involved in starch utilisation. This can be confusing to non-specialist readers.

23. Line 148. Refer to Fig 3b here. Also in Fig 3b define the signal peptides and state if Type I or Type II – see main comment on signal peptides.

24. Line 155 – refer to Fig 3d at end of sentence (after FAXX).

25. Fig 3. Label locus tags with Cazy family for clarity e.g. Bi1033-CE1. This is needed throughout MS – see main comments.

26. Line 174. Add ref to Fig S5 at end of sentence.

27. Line 182 – add ref to data (Fig S6) here.

28. Supplementary Fig 6a – annotate locus tags with Cazy family for clarity e.g. Bi1035-GH43.

29. Supplementary Fig 6. Add some description of the structures of the different xylans to the legend. Also need more detail about the assays used here and the stats.

30. Fig 3f and g. Very hard to follow with just the locus tags. You should colour code enzymes into

- classes (e.g. exo acting GH/ endo acting GH/esterase/FA esterase) to make it easier to understand.
31. Lines 204-205. Define the robust activity of 1040 here for clarity.
 32. Line 206. Expression of the 'recombinant' Bi1040 was low (refer to Fig 3c in text).
 33. Supplementary Fig 8d – why only test with arabino-oligos and WAX?
 34. Line 216. What is structure of sugar beet pulp? Describe here or in legend to S11 and also explain why this was first tested here and not earlier i.e. with the *B. intestinalis* enzymes?
 35. Line 240. Describe reaction tannase carries out to clarify differences in specificities claimed.
 36. Line 242-245. Not clear which residues are Be and which tannase from the text. Make this clearer. Also line 242 - what is S163A? Is it the inactive mutant of the tannase? If so this should be clarified.
 37. Line 254-255. Label on Fig 4 where the loop difference is between these enzymes. Also – is it near active site? Any speculation on why this region is different?
 38. Line 307 refer to Fig S15a after 'acetate.'
 39. Line 333. HIEC-6 cells are from small intestine. This is fine but should be stated as section heading states 'colonic cells'.
 40. Line 346. Sentence ending '...either spent medium.' Is confusing – change to '...spent medium from either DWB or DW grown cells.'
 41. I assume if Bi1040 is a new esterase family it will be submitted to Cazy.org?
 42. Line 376-379 should cite Rogowski et al 2015 as the ability of *Bacteroides* spp. to sense the complexity of the xylan present was shown here (glucuronoxylan vs arabinoxylan).

Reviewer #3 (Remarks to the Author):

The manuscript by Pereria and colleagues describes the functional dissection of a complex polysaccharide utilization locus involved in complex xylan degradation.

There have been a lot of similar papers in recent times - analysing what enzymes are activated for which substrate. It can be quite hard to be novel. But I found this paper enjoyable to read and informative. This is an unusual and novel system in which complexity is recognised and a panel of side-chain removing enzymes up-regulated.

As far as this referee is concerned, the paper is very close to publication-ready. It is well referenced, well written and technically well performed. I was especially asked to comment on structural aspects. The structural biology appears to have been carried out to a very high standard and is reported to expected current norm. Could not the mystery density be modelled as PEG? Partially distorted PEG fragments are very common in enzyme active sites and that was (in contrast to what is written?) a component of the crystallisation. If it looks like lysine and makes H bonds to –OH groups, then PEG might fit the bill better than an alkyl chain?

The precision in Supplemental table 10 displays no common sense (and might lead a reader to think that the authors didn't really understand what these numbers are or how they are generated?)

B values to 1/100th of an Angstrom squared (are the authors really claiming they can work at that level of precision)

R values to the fourth decimal place.

Angles to 1/100th of an Angstrom precision.

I am not a fan of chemical structures drawn without their stereo-chemical shading. I would prefer them all with shading and perhaps following the nature group chemdraw template (available at Nat Chem Biol, may need to be modified somewhat for sugars where hydroxyls are close!). But that is up to the authors and editors to decide if it is worth it.

This is not an enzymology paper per se, and it makes no difference to any conclusion - so I shan't

insist on this change. But enzymologists would much prefer k_{cat}/K_m pH profiles, which can at least be somewhat interpreted in terms of mechanism; which an elementary % activity vs pH cannot.

Line 212. I'm not a fan of the grammatical construction "which rather exhibited.." maybe that is common in USA? "instead" sounds so much nicer.

Response to Queries of Reviewer #1

Comment 1

Pereira et al. have discovered a PUL involved in sensing and cleaving off side-chains found on arabinoxylans. These esterases vary in their structure and substrate-specificity, with one having broad activities. They then go on to show that bacterial spent media following incubations with arabinoxylans induces inflammatory cytokines in cell lines and mouse models.

I found the first few sections far stronger than the link to immunity, which seems premature given the data and a bit confusing given the literature on the anti-inflammatory effects of ferulic acid. This literature is mentioned but the conflict with the current data is addressed. One explanation might be that the concentrations used are not physiologically relevant. Another might be due to the oral delivery of ferulic acid (and other metabolites) which is not the typical site of production.

Response

We appreciate the reviewer for taking the time to carefully read our manuscript and making suggestions that have vastly improved our manuscript.

It is very clear that we did not clearly explain the rationale behind the immunological experiments. The anti-inflammatory effect of the compound ferulic acid (administered as a pure substance) has been well documented in the literature, as noted by the reviewer. The models used in the literature are either in “physiologically healthy states” or in “diseased conditions”. Thus, in the present study, we aimed at demonstrating that the **ferulic acid** cleaved by the gut *Bacteroides* spp. **from complex arabinoxylan** into their spent medium is in a form capable of eliciting the reported effects. To make things clearer, we have carried out new experiments where we demonstrate the effect of the bacteria-mediated release of ferulic acid under healthy conditions by gavaging a mouse model and making immunological measurements (compared to the control, i.e., gavaging pure ferulic acid). In addition, as suggested by the reviewer, we use mouse dendritic cells to test the effects of bacteria-released ferulic acid in a “disease model” prepared by exposing the dendritic cells to lipopolysaccharide (LPS) and comparing the immunological effects with the control (pure ferulic acid). **The new information can be found on page 25, 26, and 27 line 596 to line 646**, as described below:

*“.....The spent medium of *B. intestinalis* modulates the immune response under physiological and pathological conditions. The beneficial effect of dietary arabinoxylans in the healthy population has been well-documented,⁴² and this effect has been linked to their antioxidant properties and the stimulatory effect on the innate and adaptive immunity⁴³⁻⁴⁵. While the antioxidant activity of arabinoxylans is known to depend*

on the presence of hydroxycinnamic acids, such as ferulic acid⁴², the mechanism of immune response activation by this dietary component remains unclear. Here, we investigated whether administering ferulic acid can stimulate immune response under physiological conditions by orally gavaging healthy mice with the phenolic compound and comparing the effects with that of mice receiving bacterial spent media from growth on wheat bran (WB) or growth on de-starched wheat bran (DWB). Each of the two spent media was shown to contain ferulic acid cleaved by the *B. intestinalis* cells (Supplementary Fig. 19a). The administration of the compound (ferulic acid) significantly increased the relative number of monocytes and granulocytes in both the mesenteric lymph nodes and spleen, an observation recapitulated with the DWB spent medium (Fig.7, Supplementary Table 4 and 5), which contained twice the amount of ferulic acid compared with the WB spent medium (Supplementary Fig. 19a). The plasma concentrations of the T helper type 1 (Th1) cell-derived cytokines IL-2 and IL-12 were also significantly higher with the ferulic acid and also the spent media treatment (Fig. 7); however, similar to the observation with the granulocytes and monocytes, statistical significance was detected only for the ferulic acid by itself and the spent medium from the DWB culture. Importantly, note that the DWB, with the starch removed, is more similar to the dietary component that flows to the colon, since the starch is digestible by the enzyme amylase secreted in the stomach of the human host. Hence, the DWB likely has more and easily accessible ferulic acid moieties.

In contrast to the healthy conditions examined above, under diseased conditions, including diabetes mellitus, cancer, and microbial infection, ferulic acid has been demonstrated to act as a potent anti-inflammatory factor⁴². We, therefore, evaluated the effect of ferulic acid in a disease model prepared *in vitro* by culturing mouse bone marrow-derived dendritic cells in the presence of lipopolysaccharide (LPS). A group of dendritic cells was stimulated with control media containing arabinose/xylose (Ara/xyl, the sugar components of arabinoxylan), simple or soluble wheat arabinoxylan (sWAX) or insoluble WAX (InWAX) in the presence or absence of LPS. Another group of dendritic cells was stimulated with the spent media of *B. intestinalis* grown in the Ara/xylose medium (Bi + Ara/xylose), *B. intestinalis* grown in sWAX (Bi + sWAX) or *B. intestinalis* grown in InWAX (Bi + InWAX) in the presence or absence of LPS. The difference between sWAX and InWAX is the absence of ferulic acid side chains in sWAX (Supplementary Fig. 1a,b). As expected, the concentrations of the anti-inflammatory cytokines interleukin-10 (IL-10) and transforming growth factor β 1 (TGF β 1) were significantly increased, whereas the concentration of the pro-inflammatory cytokine tumor necrosis factor α (TNF α) was significantly decreased in the cell supernatant from dendritic cells treated with ferulic acid compared to untreated controls (Fig. 8). The concentrations of IL-10 and TGF β 1 were also significantly increased, while that of TNF α was significantly decreased, in the supernatant from dendritic cells cultured in InWAX spent medium from *B. intestinalis* (containing bacterial cleaved ferulic acid, Supplementary Fig. 22) compared to dendritic cells cultured in InWAX medium without *B. intestinalis* (Fig. 8).....”

Comment 2

The immunology studies are not sufficiently controlled, especially given the elegant genetic tools used earlier. Isogenic strains with or without specific esterases are needed to control for the effects of other components of the spent media. Ideally, mono-colonized mice would be used to avoid the issues of oral delivery of microbial products. Even if this were done, I'd be concerned about downstream effects of these enzymes, since presumably they boost the efficiency of microbial fermentation leading to other end products that might have effects on host immunity independent of ferulic acid.

Response

We can understand the reviewer's concern and to address this issue, which was also pointed out by reviewer 2, we carried out the new experiment described above for the disease condition/state using dendritic cells. In this experiment, the following treatments were applied:

1. control medium containing the constituent sugars of arabinoxylan (xylose + arabinose)
2. control medium containing soluble wheat arabinoxylan (sWAX)
3. control medium containing Insoluble wheat arabinoxylan (InWAX)
4. Spent medium of *B. intestinalis* grown in control medium 1 above
5. Spent medium of *B. intestinalis* grown in control medium 2 above
6. Spent medium of *B. intestinalis* grown in control medium 3 above

The main point here is that soluble wheat arabinoxylan (sWAX) and insoluble wheat arabinoxylan (InWAX) have the same sugars (**xylose** and **arabinose**) in their polysaccharide structure. However, while **soluble wheat arabinoxylan has no ferulic acid** side chains, the **insoluble wheat arabinoxylan maintains its side chain ferulic acid**. Therefore, in the experiments described above, the microbes have access to the same sugars, i.e., arabinose and xylose, which they should ferment to the same end products or metabolites (in the spent medium). The only difference is in "6", where the bacterium has insoluble arabinoxylan with ferulic acid side chains, and which the bacterium cleaves and accumulates in the spent medium. Please, see below for the analysis of ferulic acid in the different spent media (data presented in **Supplementary Figure 22**, as shown below). Thus, it is only in the spent medium of *B. intestinalis* cultured on InWAX (i.e., Bi + InWAX) where there is a high level of ferulic acid. Furthermore, as expected, when we treated the LPS-stimulated dendritic cells (the disease model) with the 6 different spent media, only the (Bi + InWAX) spent media recapitulated the effects observed with the raw compound (ferulic acid), indicating that the bacteria-mediated release of ferulic acid from complex arabinoxylan (InWAX) is able to elicit the reported

effect of ferulic acid on the disease state. The data is presented below as Fig. 8 with an associated text.

Supplementary Fig. 22. Growth of *Bacteroides intestinalis* in minimal medium containing various substrates as sole carbon source. a HPLC-DAD chromatogram of *B. intestinalis* spent medium from 24h growth on monosaccharides (xylose+arabinose), soluble wheat arabinoxylan (sWAX) and insoluble wheat arabinoxylan (InWAX) as the sole carbon source showing release of ferulic acid into the medium only on InWAX. **b.** Quantification of ferulic acid released in the spent medium after 24h growth. The results are mean ± standard deviation of three replicates. Ferulic acid st.: Ferulic acid standard injected to determine elution time.

A detailed explanation of the disease model experiment. We treated dendritic cells with the control media (1,2,3) and the spent media (4,5,6) in the presence or absence of lipopolysaccharide (LPS). As a **positive control**, pure or commercial ferulic acid was also used to treat the dendritic cells in the presence and absence of LPS.

The concentrations of the anti-inflammatory cytokines interleukin-10 (IL-10) and transforming growth factorβ1 (TGFβ1) were significantly increased, whereas the

concentration of the pro-inflammatory cytokine tumor necrosis factor α (TNF α) was significantly decreased in the cell supernatant from dendritic cells treated with the **commercial ferulic acid** compared to the untreated controls (**Fig. 8**).

The concentrations of IL-10 and TGF β 1 were also significantly increased, while that of TNF α was significantly decreased, in the supernatant from dendritic cells cultured in InWAX spent medium from *B. intestinalis* (Bi + InWAX, see Supplementary Fig. S22 above) compared to dendritic cells cultured in InWAX medium without *B. intestinalis* (InWAX, **Supplementary Fig. 22**)(**Fig. 8**).

Thus, we were able to recapitulate the effect of commercial ferulic acid with the medium containing ferulic acid released from the insoluble wheat arabinoxylan by *B. intestinalis* (Bi +InWAX) demonstrating that the bacterial released ferulic acid is in a form that can elicit the reported effect of pure ferulic acid on a disease model.

Please see the description of the results on pages 26 and 27, lines 623 to 646, in the revised manuscript and Fig. 8 (see below). The text of the revised manuscript and the figure are also described below for your convenience.

*“..... In contrast to the healthy conditions examined above, under diseased conditions, including diabetes mellitus, cancer, and microbial infection, ferulic acid has been demonstrated to act as a potent anti-inflammatory factor⁴². We, therefore, evaluated the effect of ferulic acid in a disease model prepared in vitro by culturing mouse bone marrow-derived dendritic cells in the presence of lipopolysaccharide (LPS). A group of dendritic cells was stimulated with control media containing arabinose/xylose (Ara/xyl, the sugar components of arabinoxylan), simple or soluble wheat arabinoxylan (sWAX) or insoluble WAX (InWAX) in the presence or absence of LPS. Another group of dendritic cells was stimulated with the spent media of *B. intestinalis* grown in the Ara/xylose medium (Bi + Ara/xylose), *B. intestinalis* grown in sWAX (Bi + sWAX) or *B. intestinalis* grown in InWAX (Bi + InWAX) in the presence or absence of LPS. The difference between sWAX and InWAX is the absence of ferulic acid side chains in sWAX (Supplementary Fig. 1a,b). As expected, the concentrations of the anti-inflammatory cytokines interleukin-10 (IL-10) and transforming growth factor β 1 (TGF β 1) were significantly increased, whereas the concentration of the pro-inflammatory cytokine tumor necrosis factor α (TNF α) was significantly decreased in the cell supernatant from dendritic cells treated with ferulic acid compared to untreated controls (Fig. 8). The concentrations of IL-10 and TGF β 1 were also significantly increased, while that of TNF α was significantly decreased, in the supernatant from dendritic cells cultured in InWAX spent medium from *B. intestinalis* (containing bacterial cleaved ferulic acid, Supplementary Fig. 22) compared to dendritic cells cultured in InWAX medium without *B. intestinalis* (Fig. 8).....”*

We have also discussed these findings on pages 32 and 33, lines 766 to 787, and the text is presented below:

*“.....The beneficial effects of the compound have been attributed not only to its antioxidant function, but also to its modulatory activity on the immune system and the inflammatory response⁵⁷. Furthermore, the rich content of ferulic acid has been attributed to the beneficial effects of arabinoxylans in several disease states^{42,59}. Consistent with the regulatory activity of ferulic acid on host immunity and inflammation under pathological conditions, we found that both ferulic acid sources (i.e., the raw compound or in bacterial spent medium) increase the expression of anti-inflammatory cytokines (TGF β 1, IL-10) and decrease that of TNF α , a pro-inflammatory cytokine, from dendritic cells stimulated with LPS. This observation is consistent with an earlier report, also using a murine cell line, that noted that ferulic acid might have potential as an anti-inflammatory drug⁶⁰. Recent evidence also suggests that ferulic acid protects the host health condition by stimulating the immune system and physiological homeostasis⁴². Supplementation of ferulic acid in the diet increases the number of circulating leukocytes, lysosome activity and the serum levels of antioxidant enzymes in fish, enhances the activity of leukocytes in mice, and prevents diabetes mellitus and metabolic syndrome in mice and rats⁶¹⁻⁶⁶. Here, we found that administration of ferulic acid (the pure compound and also in the spent medium of the colonic bacterium *B. intestinalis*) enhances the number of monocytes and granulocytes and the concentration of circulating Th1 cell-derived cytokines in healthy mice, further supporting the protective role of ferulic acid under physiological or healthy conditions.”*

Fig 8 in the new manuscript is presented below for your convenience.

Fig. 8. The spent medium of *B. intestinalis* modulates the inflammatory response under pathological conditions. Bone marrow-derived cells were isolated from healthy C57BL/6 mice, cultured *in vitro* and differentiated to dendritic cells in the presence of Flt3. Dendritic cells were then cultured in medium containing arabinose/xylose (Ara/xylose), soluble wheat arabinoxylan (sWAX) or insoluble WAX (InWAX) or cultured in the presence of Ara/xylose (Bi + Ara/xylose), sWAX (Bi + sWAX) or InWAX (Bi + InWAX) spent medium from *B. intestinalis*. Maturation of dendritic cells was induced with lipopolysaccharide (LPS). Tumor necrosis factor α (TNF α), interleukin-10 (IL-10) and transforming growth factor β 1 (TGF β 1) were measured by enzyme immunoassays. Statistical analysis was performed using unpaired Student *t*-test, **p*<0.05.

Comment 3

I've also never been a huge fan of using Caco-2 cells (a cancer cell line) to study bacterial interactions with the immune system. The authors might consider DC or T cell-based assays, which are likely more physiologically relevant. The same issue is true for the HIEC-6, which are small intestinal not immune cells.

Response

As described above, we followed the reviewer's suggestion by carrying out the experiment with dendritic cells (or DC) and removing the data for the Caco-2 cells and the HIEC-6 cells.

Comment 4

Line 59 – typo – “nutrients acquisition”

Response

Please see **line 61**: “nutrient availability and acquisition in that environment”.

Comment 5

Line 299 – I'm not sure this data can be used to conclude that ferulic acid is not metabolized. Maybe accumulates during growth instead?

Response

We have changed the expression to accumulates. **Please, see line 567** “The cleaved ferulic acid accumulates during complex arabinoxylan fermentation”.

Comment 6

Figure 1b-d needs stats.

Response

Please, see page 56. We have split the analysis into the two proteins that form the transporter, i.e., SusC and SusD, and we have applied statistics to the data in Fig. 1b-g

Comment 7

Figure 2 – the upregulation is striking, how does this look genome-wide? Is this the most up-regulated PUL?

Response

Other genes were upregulated genome-wide; however, as a cluster, the genes in this region (PUL) were the most highly upregulated. In the Discussion section, we indicate that other PULs in the genome were upregulated; however, not to the extent of the EGE PUL. **Please, see line 173-198.** In addition, we have added a Table listing the top 50

upregulated proteins (Supplementary Table 1a,b,c on pages 131-136). The text is also presented below.

“...We carried out whole genome transcriptional analyses of the three Bacteroides spp. grown on complex arabinoxylan to determine if their EGE PUL associated genes will be up-regulated, as observed with the susC/susD gene pair, during growth on the complex arabinoxylan substrates. All three bacteria up-regulated expression of their respective EGE PUL on the complex substrate compared to the monosaccharide mixture. Bacteroides intestinalis showed the highest relative expression of its EGE PUL, followed by B. cellulosilyticus and then B. oleiciplenus (Fig.2a,b,c). Other GH and CE genes on the genomes of the three bacteria were also upregulated, although at a far lower level (Supplementary Table 1a,b,c), suggesting that the EGE PUL is the main PUL for complex arabinoxylan degradation. Thus, other enzymes outside of the EGE PUL likely participate in the degradation of the complex polysaccharide, since the nature of the arabinoxylans encountered in the diet is known to be more intricate than the complex arabinoxylans used in the present study. As an example, arabinoxylans and xylans are known to crosslink with cellulose, another plant cell wall component. This strategy of using PULs together with non-PUL associated enzymes, encoded in other regions of the genome, to degrade complex arabinoxylan is consistent with our findings on soluble arabinoxylan degradation, where two major PULs were highly upregulated, with several genes, some unassociated with PULs, also upregulated at lower levels²⁶. In fact, a gene encoding a GH9 family enzyme, known to be involved in cellulose degradation, is located in the EGE PUL of B. cellulosilyticus (Fig. 1a). This gene was upregulated together with the other EGE PUL associated genes during degradation of complex arabinoxylan (Fig. 2b). In the natural environment, a cellulose targeting enzyme would likely enhance accessibility of substrate to the arabinoxylan-degrading enzymes to facilitate degradation.”

Comment 8

Figure 3c – why are there multiple bands in some of the protein lanes?

Response

We provided an explanation for this observation and now that information appears from line 245-251 as follows:

“.....Recombinant Bi1038-CE6/CE1 degraded into two polypeptides (Fig. 3c) during purification, and we hypothesized that the two fragments derive from a proteolytic cleavage in a linker between its two modules (Fig. 3b,c). Expression of the recombinant protein in the presence of a protease inhibitor resulted in purification of the full-length polypeptide (Supplementary Fig. 4) and, therefore, supporting our hypothesis of a proteolytic cleavage.

Comment 9

Figure 3d – Any idea what’s going on with Bi1040????

Response

We have done extensive Molecular Dynamics Simulations to help explain the versatility of this enzyme. These results appear under **Conformational preferences of the esterases regulate substrate specificity**”, i.e., from Line 488-565. In brief, one of the contributing factors is, compared to the other esterases, Bi1040 has a large binding pocket volume and this likely can accommodate different large substrates and exhibit catalytic triad activity toward a diverse set of substrates. This should be of immense benefit to the microbe in removing ferulic acid in diverse linkages on arabinoxylan, leading to extensive hydrolysis of the polysaccharide to access its sugar units (xylose and arabinose) for metabolism.

Comment 10

Figure 5 – bacterial genetics would really help make this more definitive.?

Response

Unfortunately, there are no genetic systems available for manipulating any of the four primary *Bacteroides* spp used in the present work, and the EGE PUL is also not widely distributed. It is not found in any *Bacteroides* spp with a genetic manipulation system. We think that by carrying out the experiments above that showed that only the *B. intestinalis* spent medium from insoluble wheat arabinoxylan (the only medium containing ferulic acid) treatment could recapitulate the effect of commercial or pure ferulic acid makes it clear that the *Bacteroidetes* released ferulic acid from dietary fiber (i.e., complex arabinoxylan) likely imparts the effects of ferulic acid that have been reported in the literature. Importantly, we also recapitulated the reported effects with pure ferulic acid treatment (positive control) in the present study.

Comment 11

Figure 6 – a lot of these comparisons aren’t significant, especially for “WB”. The fact that changes are seen in spleen and lymph node are consistent with the concern that oral gavage leads to rapid absorption of these compounds, inducing systemic inflammation that might not be seen with the same level of compound in the colon.

Response

The observation that the wheat bran (WB) treatment did not lead to a statistically significant difference is reasonable, and we should have addressed this important observation, but we failed to do so. We have included an explanation for this observation in **lines 596-646**. The spent media used here are from wheat bran (WB) and **de-starched**

wheat bran (DWB). In the human diet, wheat bran is de-starched in the stomach through the enzyme amylase, and thus the products or remnants that flow to the colon are more similar to the DWB used in this experiment.

Also, note that when we measured the amount of ferulic acid released into the DWB spent medium, it was about twice the amount present in the WB spent medium. Thus, the higher amounts of ferulic acid released in the substrate that better mimics what may be seen in the human colon led to results more reflective of the result elicited by pure ferulic acid, i.e., increase monocytes and granulocytes, and the helper type 1 (Th1) cell-derived cytokines plasma IL-2 and plasma IL-12. Since this is the healthy model, the increase in the number of monocytes and granulocytes and the concentration of the circulating Th1 cell-derived cytokines in healthy mice supports the protective role of ferulic acid under physiological conditions. Our results are thus in line with reports in investigations of the effect of ferulic acid under healthy conditions, including with a fish model and a rat model, as described in the discussion section (**please see lines 760-787**).

Response to Queries of Reviewer #2:

The paper by Pereira et al describes the identification of a predicted carbohydrate esterase enriched PUL from a gut *Bacteroides* spp that is only expressed on 'complex' arabinoxylans (defined as the fraction of purified arabinoxylan that has ferulate cross links). The carbohydrate active enzymes from the *B. intestinalis* EGE PUL are biochemically and structurally characterised and several are shown to act together to release ferulic acid from complex xylans. Growth of *B. intestinalis* on complex xylans results in a build-up of free FA and the authors use cell lines and a mouse model to test the effect of the released FA on intestinal cell immune function.

While the proposed effect of FA on intestinal cells has been described, the finding that FA is released from dietary xylans by prominent gut bacteria is both novel and interesting.

I have however some comments to be addressed.

Response

We appreciate the reviewer for carefully reading our manuscript, for the very encouraging comments, and for the suggestions that have led to a significant improvement of our manuscript.

Comment 1

For both the cell line and mouse experiments the effect of spent media, while significant in some cases, seems very small making me question if it is real. In addition, a key control of spent media from Ara or Xyl and/or simple WAX grown cells (ie. no FA present) is missing – this is needed to be sure the effect seen is due to FA released

from xylans as proposed and not just say SCFAs and/or LPS that would be present in all spent media samples.

Response

The experiment suggested by the reviewer is very important, and we agree that such an experiment will make the results more convincing. Therefore, we have used dendritic cells (suggested as more relevant cell line by Reviewer 1) to carry out an *in vitro* experiment, where we include all the controls suggested by the reviewer. We are happy to report that we were able to recapitulate the effect of commercial (raw) ferulic acid only with the spent medium from complex arabinoxylan (i.e., insoluble wheat arabinoxylan + *B. intestinalis*) or the spent medium containing bacterial released ferulic acid. The experiment and results are also shown above for Reviewer 1 for his/her perusal. The final results are shown in Fig. 8 (above in the responses to Reviewer 1's queries and also on page 70 of the new manuscript). The text describing the work also appears from Line 596-646. We also show below, the analysis of ferulic acid in the spent media in the experiments suggested by the reviewer, i.e., **Supplementary Fig. 22 (page 117)**.

Supplementary Fig. 22. Growth of *Bacteroides intestinalis* in minimal medium containing various substrates as sole carbon source. a HPLC-DAD chromatogram of *B. intestinalis* spent medium from 24h growth on monosaccharides (xylose+arabinose), soluble wheat arabinoxylan (sWAX) and insoluble wheat arabinoxylan (InWAX) as the

sole carbon source showing release of ferulic acid into the medium only on InWAX insoluble. **b.** Quantification of ferulic acid released in the spent medium after 24h growth. The results are mean \pm standard deviation of three replicates. Ferulic acid st.: Ferulic acid standard injected to determine elution time.

Comment 2

2. Line 111-112. How do you know the EGE PUL is required for degradation of complex AX? What data is this statement based on?

Response

We have re-written the statement. **Line 138 – 140**

“Surprisingly, a putative esterase genes-enriched (EGE) PUL, that we hypothesize could enhance extensive hydrolysis of complex arabinoxylan, was not upregulated.”

Comment 3

3. In Table S7 it shows that at least one of the simple WAX PULs is also upregulated on complex WAX (InWAX - Bi04193-04205). The authors should comment on this and what this might mean for the degradation of InWAX as the MS is currently written from the perspective that only the EGE PUL is involved in degradation of complex WAX.

Response

We have included information in the manuscript to this effect on **line 173-198** (Results section).

“.....We carried out whole genome transcriptional analyses of the three *Bacteroides* spp. grown on complex arabinoxylan to determine if their EGE PUL associated genes will be up-regulated, as observed with the *susC/susD* gene pair, during growth on the complex arabinoxylan substrates. All three bacteria up-regulated expression of their respective EGE PUL on the complex substrate compared to the monosaccharide mixture. *Bacteroides intestinalis* showed the highest relative expression of its EGE PUL, followed by *B. cellulosilyticus* and then *B. oleiciplenus* (Fig.2a,b,c). Other GH and CE genes on the genomes of the three bacteria were also upregulated, although at a far lower level (Supplementary Table 1a,b,c), suggesting that the EGE PUL is the main PUL for complex arabinoxylan degradation. Thus, other enzymes outside of the EGE PUL likely participate in the degradation of the complex polysaccharide, since the nature of the arabinoxylans encountered in the diet is known to be more intricate than the complex arabinoxylans used in the present study. As an example, arabinoxylans and xylans are known to crosslink with cellulose, another plant cell wall component. This strategy of using PULs together with non-PUL associated enzymes, encoded in other regions of the genome, to degrade complex arabinoxylan is consistent with our findings on soluble arabinoxylan degradation, where two major PULs were highly upregulated, with several genes, some unassociated with PULs, also upregulated at lower levels²⁶. In fact, a gene encoding a GH9 family enzyme, known to be involved in cellulose

degradation, is located in the EGE PUL of *B. cellulosilyticus* (Fig. 1a). This gene was upregulated together with the other EGE PUL associated genes during degradation of complex arabinoxylan (Fig. 2b). In the natural environment, a cellulose targeting enzyme would likely enhance accessibility of substrate to the arabinoxylan-degrading enzymes to facilitate degradation.”

Comment 4

In relation to this in Table S7 it is shown that there are multiple esterases (both CE1 and CE6) in the other two simple AX PULs. Do the authors know what the roles of these enzymes are? Why have esterases in these xylan PULs if not to de-esterify xylans?

Response

This is an important question, and from our knowledge analyzing these enzymes, we think that these enzymes involved in simple AX PULs also have esterase activities; however, they may be limited in their activities based on the structure of the polysaccharide. As observed in the esterases analyzed in the present study, the range of substrates for Bi1040-FAE is very different from that of Bi1033-CE1 (both ferulic acid esterases), and Bi1039-CE1 and Bi1033-CE1, both of the same family have different activities. Thus, these organisms likely have evolved diverse esterases in response to the complexities that exist in the linkages found in the plant matter. In other words, depending on the configuration of the substrate, an esterase may be blocked or limited in its catalytic activity. We have previously published data also on GH enzymes, i.e., multiple GH3 enzymes that exhibit different capacities to cleave decorated oligosaccharides of xylose (please, see PMID: 20190048).

We have analyzed several esterases, and sometimes of the same family, and their activities are very different. An example is Bi1033-CE1 and Bi1039-CE1 in the present study where the activities are very different. We have also characterized many esterases from *B. intestinalis* (*unpublished data and PMID: 28669823*), and we think that the diversity and multiplicity of the esterases in these bacteria is a reflection of the diversity of ferulic acid linkages in the human diet or plant matter. We have a statement to this effect in the Discussion (**Line 739-741**):

“.....Furthermore, the importance of ferulic acid and its pervasiveness and diversity of linkages in the human diet is underscored by the abundance and also diversity of ferulic acid esterases in the colonic Bacteroides spp.^{17,29}.”

Comment 5

All locus tags should be further annotated throughout the MS (text and Figs) with Cazy family or activity for clarity (e.g. Bi1035-GH43) otherwise the reader has to constantly refer to Fig3a to have any idea what activity the locus tag refers to.

Response

This is a very good suggestion and we have made these changes throughout the entire manuscript, i.e., both text, figs and tables. See the example below (Page 82 in the new manuscript).

Comment 6

Lines 185-188. Clarify why only 1042 was tested against xylo-oligosaccs and 1043 against InWAX by HPAEC-PAD? Was it because they showed low levels of reducing sugar release against xylans compared to 1035 and 1041? Ideally test all enzymes against xylo-oligos and WAX by HPAEC-PAD or at least explain why only certain enzymes assayed.

Response

Please, see the new Fig. 4 (Page 62 of new manuscript). We have presented HPAEC-PAD traces for all the GH enzymes on insoluble wheat arabinoxylan (InWAX) and de-starched wheat bran (DWB). In this revised submission, we show the reactions of the four GH enzymes, i.e., BACINT_01035-GH43, BACINT_01041-GH43, BACINT_01042-GH3,

and BACINT_01043-GH43 on two types of arabinoxylan. The results clearly show that BACINT_01043-GH43 releases arabinose from both substrates, BACINT_01042-GH3 releases only xylose from both substrates, while BACINT_01035-GH43 and BACINT_01041-GH43 each releases mostly a range of xylo-oligosaccharides. From these analyses, it is clear that while BACINT_01035-GH43 and BACINT_01041-GH43 are endo-acting enzymes on arabinoxylans, BACINT_01042-GH3 is a terminal cleaving enzyme, releasing the monosaccharide xylose, and BACINT_01043-GH43 releasing the side chain arabinose. These observations are generally confirmed by the data in **Supplementary Fig. 6** (Page 82-83).

Comment 7

Line 190. The data does not show ‘...enzymes function to completely hydrolyze arabinoxylans.’ The data show that the enzymes are able to hydrolyse arabinoxylans, but not necessarily to completion (ie. only free monosaccharides, acetate and FA remaining). Please reword or present data that shows the enzymes hydrolyse AX to completion.

Response

We agree with the reviewer on his/her analysis. We now present a new analysis in **Fig. 4** (page 62-63) that shows the degradation of two arabinoxylans to their monosaccharides, arabinose and xylose (**Fig. 4f and 4l**). In **Fig. 3**, we also show the extensive release of ferulic acid, and with acetic acid esterase activity shown for Bi1038-CE6/CE1, we assume that all the enzymes together will lead to complete depolymerization. We have, however, modified the sub-title to read “The EGE PUL enzymes function to depolymerize arabinoxylans” (Please, see page 12 of the manuscript).

Comment 8

Line 195. How do the GH43s (Bi1035 and 41) release free FA from arabinoxylan as shown in Fig 3f and g? (see 1035 and 1041 only lanes – small amount, but above control, of FA released with all except 1041 vs InWAX). Are you proposing that the enzymes are dual acting GH/esterases? Please provide an explanation for this activity or if it is an artefact of the assay used.

Response

We appreciate the reviewer’s critique, and we think these small levels above the control are an artifact of the assay. We have provided a statement on **line 293-296** to this effect.

“.....Note that in some of the incubations with individual GH enzymes (Fig. 3f,g), small amounts of ferulic acid, above the control, appeared to be released. These are likely artefacts of the assay, as GH enzymes are not known to harbor esterase activities.”

Comment 9

9. Line 200. How do you know these enzymes are endo-acting? Is it just because they release the most reducing ends from xylan of all the GHs? This is very circumstantial evidence. This should be clarified and ideally present some data that directly shows endo activity (e.g. HPAEC-PAD or TLC showing range of oligosaccharides being produced during digestion of AX).

Response

We agree with the reviewer that this is a very important point that needed clarification. In the new manuscript, we show in the new Fig. 4a and 4g (Page 62-63) on insoluble wheat arabinoxylan (InWAX) and on de-starched wheat bran (DWB) that both Bi1035-GH43 and Bi1041-GH43 release oligosaccharides ranging from products close to xylobiose (X2) to xylohexaose (X6), i.e., by HPAEC-PAD. Thus, we confirm here that the two enzymes are endo-acting enzymes, unlike the other two enzymes (Bi1042-GH43 and Bi1043-GH43). Please see the text in the new manuscript, on lines 297-321.

Comment 10

10. Line 226. If the closest structural homologue of the GH43 is the double Ara specific *H.insolens* enzyme (3ZXL) have you tested to see if the Be enzyme is double or single Ara specific? The only data I can see is for InWAX (Fig S8d) which will have both double and singly substituted side chains so not clear from this.

Response

We appreciate the reviewer's insight and hope to devote more time to gain a deeper understanding of this interesting enzyme. We, however, think digging deeper into the GH43 of *B. eggerthii*, while our interest is actually in the structure of its appended esterase (Bi1040-FAE like polypeptide) could become a distraction for the reviewer and the reader. Please, note that all the enzymes being characterized here are from *B. intestinalis*, and that we only wanted to determine the structure of the FAE in *B. eggerthii* (Be) to gain an understanding of the versatility of the homologous Bi-1040 in *B. intestinalis*. If the reviewer thinks that this is absolutely essential, we will find the substrates to test the Be enzyme for the activity.

Comment 11

11. It looks like several of the signal peptide predictions are wrong (see Fig 3b for enzymes proposed to have signal peptide) and therefore the model (Fig S18) proposed for complex xylan breakdown incorrect. For example Bi1035 has a strongly predicted Type I signal sequence (see below - using either LipoP or the new SigP v5) which would mean in *Bacteroides* spp it was much more likely to be periplasmic than on the cell surface. Furthermore there is no experimental data presented regarding the enzyme localisation and so the whole model is based on these predictions.

These need to be corrected and the model changed. Do the analysis again and be more

specific

e.g. >Bi1035

MIMKTKAILLGTAAALMFTLTSEKAI|AQIGTPYIHDPSTIMECDGKY|YTFGTGGGGLISED
GWTWNGGGVVRPGGGAAPDAVKIGDRYLVAYSATGGGLGGGHSGKVLTMWNKTLDP
NSPDF
KYTEPIEVASSAYDEDCAIDAGLLLDPTDGRLWLSYGTYFGFIRLVELDPATGKRMEG
N
KEIDIAIDCEATDLIYRDGWYLLGTHGTCCDGPNSTYNIVVGRSRKVTGPYVDNMGRK
M
LEGGGKMVIAAGNRQTGPGHFGLFKVADGVEKMSCHF|EADFDRGGRSVLGIRPLLWK
NGW
PVAGEVFK|EGTYEIESERRGYALELAVDFVRMEYTRHR|FWEKDDTPV|VPLKPQTLEDV
IG
TWPQGKINVRIGDYMFRPHQKWTITAVPNGGGYLGGPYKIVIEGTNRALAATAGAEI
MT
VPEFTGAPEQLWRIDQLTDGTYRIMP|KKVPGTDKELVLV|SVGDSTPSL|GIFDMNSDNS
KW
NFRDH

LipoP output for Bi1035:

Sequence Spl score=14.7597 margin=14.960613 cleavage=26-27

Cut-off=-3

Sequence LipoP1.0:Best Spl 1 1 14.7597

Sequence LipoP1.0:Margin Spl 1 1 14.960613

Sequence LipoP1.0:Class CYT 1 1 -0.200913

Sequence LipoP1.0:Class TMH 1 1 -0.744742

Sequence LipoP1.0:Signal Cleav| 26 27 14.6568 # EKAIA|AQIGTP

Sequence LipoP1.0:Signal Cleav| 23 24 10.6123 # LTSEK|AIAQI

Sequence LipoP1.0:Signal Cleav| 24 25 7.30313 # TSEKA|IAQIG

Sequence LipoP1.0:Signal Cleav| 21 22 7.20032 # FTLTS|EKAIA

Sequence LipoP1.0:Signal Cleav| 22 23 4.65163 # TLTSE|KAIAQ

Sequence LipoP1.0:Signal Cleav| 25 26 2.9738 # SEKAI|AQIGT

Sequence LipoP1.0:Signal Cleav| 20 21 1.95987 # MFTLT|SEKAI

Sequence LipoP1.0:Signal Cleav| 28 29 1.24524 # AIAQI|GTPYI

SigP v5 output for Bi1035:

Protein type: Likelihood

Signal peptide (Sec/SPI): 0.9907

TAT signal peptide (Tat/SPI): 0.0048

Lipoprotein signal peptide (Sec/SPII): 0.0037

Most of the enzymes appear to be Type I signals and therefore periplasmic. The only

exception is Bi1041-Gh43 which is a predicted lipoprotein and so may be on the cell surface.

In this new model 1041 would be in the place of 1035 on the cell surface and most of the other enzymes in the periplasm, including the newly discovered FA esterase Bi1040. FA would therefore be released in the periplasm and have to be exported.

Bi1033-CE1 – SPI - periplasm

Bi1038-CE6/CE1 – SPI - periplasm

Bi1039-CE1 – SPI - periplasm

Bi1040-HP – SPI – periplasm. NB. N-term Met likely wrong here as database seq (starts MSLL) predicts no signal peptide (likely due to several Asn/Gln residues), but from downstream Met (MKLV) is well predicted Type I signal peptide.

>Bi1040

MSLLRFFLNNCQLFNKLKN

MKLVFGFVCALFYFMSFGQITQWTDINYANDSLEGHKLDIYLPDGGQTEY

KVVVLIYGSAWFANNMKQMAFQAMGKPLLDGGFAVVSINHRSSGDAKFPAQINDVKA

AVRFIRAHADEYR

LDTSEFIGITGFSSGGHLSSLAGTTNGVKVYKVGDTEMDIEGNVGDCTSFSSRVDAVVD

WFGPIDMTRMEN

CATTKGADSPEAALIGGTPAEHMDVL TLLNPMTYIDEKDPKFLVIHGDADTVVPHCQSV

FFKDTLSAKGR

LEEFITVPQGQHGPITFNEQTFKKMTEFFRKQAAMDLC PANITLVPHIRPGSKKNPPQK
QFYTTY

Bi1041-GH43 – SPII – cell surface

Bi1042-GH3 – SPI - periplasm

Bi1043-GH43 – SPI – periplasm

Response

We appreciate the reviewer's careful analysis of the sequences. We have re-analyzed the localizations, which agree with the reviewer's analysis. The model has, therefore, been modified and appears in the new manuscript as **Supplementary Fig. 23** (Page 118-119), where an alternative model is also considered.

Comment 12

12. Supplementary Fig 18. I have a few queries about the model (in addition to pt 11 above):

a. It is not clear on the fig what the role of each of the enzymes is. Please make clearer what each activity is – possibly by showing an arrow from the enzyme to the bond cleaved? Otherwise the reader is left trying to cross reference info from the rest of the paper to understand how you came to propose this model.

Response

This is a very good suggestion. We have incorporated this idea into the new proposed model. **Please, see Supplementary Fig. 23** (Please, see page **118-119**).

Comment 13

b. Why not show all enzymes? Where are Bi1033 and 1038? From your data (Fig 3e) 1038 releases the most acetate from xylan – should this not be shown?

Response

This is an excellent suggestion. All the enzymes are shown in the new proposed model. Please, see **Supplementary Fig. 23** (Please, see page **118-119**).

Comment 14

c. With regard to this what is the rationale for multiple CE1 and CE6 enzymes in the PUL if only need a single xylan esterase and FA esterase to de-esterify complex xylan? Some comment on this should be made.

Response

It is likely, and as can also be deduced from our data on Bi1040 activity, that the enzymes (esterases) work on different ferulic acid linked-substrates. As we note above, for a more efficient degradation, the organisms have evolved diverse ferulic acid esterases that work on different linkages to achieve extensive degradation. This hypothesis is supported by the variations in the linkages cleaved by the ferulic acid esterases characterized in the present study (Fig. 3d, **page 60-61** and supplementary Fig 2 and 3, **page 74-77**). We have also included a section on Molecular dynamics simulations (**line 488-565 or page 20-24**) to provide insights into the substrates that fit the active sites of the esterases.

Please, also see the statement on line 739-741 in the Discussion as follows:

“Furthermore, the importance of ferulic acid and its pervasiveness and diversity of linkages in the human diet is underscored by the abundance and also diversity of ferulic acid esterases in the colonic Bacteroides spp.^{17,29}.”

Comment 15

d. What is the outer membrane ‘sugar transporter’? This seems to have appeared from nowhere. What is the evidence for this and why do you think the bacterium needs this activity? (import of surface released Ara?).

Response

The model should not include such a transporter, and it has been removed. Transport should be through the SusC/SusD transporter as shown (Supplementary Fig. 23, **pages 118 -119**).

Comment 16

e. Line 1386. The identification of the branched oligos as the products sensed by the HTCS is likely correct based on previous studies (e.g. Martens et al 2011), but you should state this is speculation for this system.

Response

We have changed the title to clearly state that this is a prediction, i.e., “**A model predicting the regulation of the EGE PUL in *B. intestinalis***”. The model is currently Supplementary Fig. 23 (**pages 118-119**). We have also included the important reference “Martens et al. from PLoS Biology”. **Please see line 1707 or page 119**. Legend for the Supplementary Fig. 23.

Comment 17

13. Line 64 – what does ‘Unique consortium’ mean? Unique to what – humans or the colon? Rephrase this as I think you mean the consortium is unique to each individual at the species and strain level. But Bacteroidetes and Firmicutes common theme amongst most mammals.

Response

We have modified the statement. **Please see line 66-68**: “In humans, a microbial consortium comprised mostly of two bacterial phyla, the Bacteroidetes and Firmicutes, has evolved to degrade complex polysaccharides in the lower GIT”

Comment 18

14. Line 67. Clunky start. Change to ‘best studied in the Bacteroidetes’ ie. infers less studied in Firmicutes. This correction has been made.

Response

Please, see line 68-69. The sentence has been changed to “The process of polysaccharide degradation has been best studied in the Bacteroidetes”.

Comment 19

15. Line 75. Missing some key refs for HTCS as oligosaccharide sensors. Martens et al 2011 (PMID: 22205877) first showed the periplasmic domains of Bacteroides HTCS bind oligosaccharide ligands and this is the inducing signal for their cognate PUL, while Lowe et al 2012 (PMID: 22532667) showed the first structure of a Reg_{prop} class

HTCS bound to its target ligand and proposed a mechanism for signal transduction in this class of sensor-regulators.

Response

Thank you to the reviewer. These are very important references and information and references have been incorporated. Please, see line 78-88 as follows:

“...Biochemical, genetics, and structural analyses have been used to demonstrate that the sensor module of a Bacteroides thetaiotaomicron HTCS binds specifically to monomeric fructose to activate a fructan degradation PUL¹². Furthermore, Martens et al. reported that larger forms of the sensor module from diverse HTCS (i.e., ~700-900 amino acid-polypeptides versus ~300 amino acid-polypeptides in the fructose HTCS sensor) in B. thetaiotaomicron and Bacteroides ovatus recognize oligosaccharides derived from polysaccharides uniquely targeted by a cognate PUL for degradation³. The crystal structure of the sensor module of an HTCS together with its Y_Y_Y domain has provided the first look at a homodimeric structure of a Bacteroidetes HTCS and therefore allowed proposal of a mechanism for signal transduction in this class of sensor regulators¹³...”

Comment 20

16. Line 79. Add Ze et al 2012 (PMID: 22343308) and Cartmell et al 2018 (PMID: 30349080) as key refs on cross feeding in the microbiota.

Response

Please, see line 93. We have added these two significant manuscripts as follows:

“In orchestrating the initial enzymatic attack on polysaccharides in the GIT, members of the Bacteroidetes and the Firmicutes act as primary or keystone polysaccharide degraders, making nutrients available for their own metabolic processes and also to cross-feed the colonic microbiome members that lack the requisite hydrolytic enzymes^{4,14-19}. PMID:22343308 is reference 16 and PMID: 30349080 is reference 19.

Comment 21

17. Line 92. Missing Rogowski et al 2015 (PMID: 26112186). Key ref on xylan utilisation by Bacteroides.

Response

This reference has been added. See line 106 (reference 27), and also line 681.

Comment 22

18. Lines 95-97. Vague. What is ‘arabinoxylan lacking side chains’ – do you mean linear xylan? This is insoluble unless short dp. Also what are ‘...all the signatures for complex arabinoxylan degradation’? Please clarify both terms.

Response

The soluble arabinoxylan we used lack ferulic acid side chains and by “all the signatures” we meant all the enzymes especially diverse esterases (known and predicted) or side chain removing enzymes.

We have written this section to make it clearer (**line 109-114**)

“...Importantly, simple or soluble arabinoxylan (i.e., without ferulic acid side chains) induced expression of two unique PULs in B. intestinalis without upregulating a PUL containing several predicted esterases that likely enhance degradation of complex arabinoxylans (i.e., with ferulic acid side chains) by this colonic bacterium²⁶. This observation led us to hypothesize that the Bacteroidetes deploy different PULs for degradation of arabinoxylans of different complexities.”

Comment 23

19. Line 99 – what does ‘with primary degrader signatures’ mean?

Response

Here, we meant the organisms that contain in their genomes, the genes needed to initiate degradation of polysaccharides.

This sentence has been deleted and replaced with a new sentence (**Line 115-121**), as follows:

“.....In this study, we culture diverse members of the human colonic Bacteroidetes on arabinoxylans of different complexities and demonstrate that these bacteria have evolved to distinguish simple from complex arabinoxylans by deploying different PULs for their degradation. We identify an esterase genes-enriched (EGE) PUL that targets complex arabinoxylan degradation and systematically characterize the enzymes associated with the PUL to delineate the linkages cleaved by the different enzymes.”

Comment 24

20. Line 111. Define EGE PUL here – more than one predicted esterase? What family of esterases? Give locus tags.

Response

We have provided a definition for EGE PUL and also included locus tags. **Please, see line 141-151.**

Here, we define an EGE PUL as a PUL with more than two genes encoding either known or putative carbohydrate esterase family proteins. Thus, in the B. intestinalis EGE PUL, there are five putative esterases (BACINT_01033-CE1, BACINT_01034-putative

esterase, BACINT_01038-CE6/CE1, BACINT_01039-CE1, and BACINT_01040-putative esterase), and in the corresponding PUL in *B. cellulosilyticus* four putative esterases (BACCELL_02154-CE1, BACCELL_02153-putative esterase, BACCELL_02145-CE1, BACCELL_02144-CE1). Only two putative esterases were found in the EGE PUL of *B. oleiciplenus* (HMPREF9447_02532-CE1 and HMPREF9447_02531-putative esterase); however, due to the conservation of the critical glycoside hydrolase (GH) genes in the PUL, it was included in our analyses (Figs. 1a, 2).”

Comment 25

21. Line 118-120. Fig S1 only shows structures of simple AX (sWAX) and insoluble AX (InWAX), but the text refers to FAOX and DWB. The authors should describe the structures of these other two polysacchs for clarity.

Response

We agree with the reviewer that the structures will be very helpful to the reader. The FAOX structure is initially shown in **Supplementary Fig. 2 and 3 (pages 74 and 76)**. The DWB structure is not known, but it is proposed to be similar to InWAX. **Please, see line 158-162.** We have included the following information:

“.....The complex substrates were feruloylated arabinoxylan oligosaccharides (FAOX), de-starched wheat bran (DWB), and insoluble wheat arabinoxylan (InWAX). While the structure of the DWB is unknown, we assume that it is similar to InWAX (Supplementary Fig. 1b). In addition, we present the structures of the FAOX in Supplementary Figs. 2,3...”

Comment 26

22. Fig 2 legend Line 893. Clarify that Sus-like proteins are named after the prototypic system but are not all involved in starch utilisation. This can be confusing to non-specialist readers.

Response

We have updated the Figure 2 legend by including the information suggested by the reviewer. **Please, see line 1352-1355 (page 59).**

“...For the name Sus or Starch utilization system, note that although this name originally derives from the first functionally characterized homolog or prototypic system, this name is still maintained for these proteins although it is known that similar systems transport different polysaccharides, including xylan, pectin, mannan and others...”

Comment 27

23. Line 148. Refer to Fig 3b here. Also in Fig 3b define the signal peptides and state if Type I or Type II – see main comment on signal peptides.

Response

We have referred to Fig. 3b, after the last statement of this paragraph. **Please, see line 221.**

In **Fig. 3b**, we provide a column showing the predicted location of the polypeptide, i.e., extracellular or periplasmic. **See page 60.**

Please, see line 1365-1368. We have defined the signal peptides, i.e., type I and Type II in the legend as follows:

“The N-terminally appended dark boxes indicate the predicted signal peptide is Type I (SPI) and therefore a likely periplasmic localization, while the open box indicates predicted signal peptide Type II (SPII) and therefore a predicted outer membrane or extracellular localization”.

Comment 28

24. Line 155 – refer to Fig 3d at end of sentence (after FAXX).

Response

We have done so. **Please, refer to line 226-229.**

“.....Using a panel of feruloylated oligosaccharides²⁹, we showed that BACINT_01033-CE1 (or Bi1033-CE1) exhibits low ferulic acid cleavage activity on feruloylated monosaccharide (FA) and about 10 times higher activity on a feruloylated trisaccharide (FAXX) (Fig. 3d).”

Comment 29

25. Fig 3. Label locus tags with Cazy family for clarity e.g. Bi1033-CE1. This is needed throughout MS – see main comments.

Response

We have labeled all enzymes with their CAZy family designation in **Fig. 3**. **Please, see page 60.** We have also maintained this naming throughout the manuscript. This is a very good suggestion, since it makes the manuscript easy to follow.

Comment 30

26. Line 174. Add ref to Fig S5 at end of sentence.

Response

We have done so. **Please, see line 252-258.**

“Enzymatic assays with synthetic substrates, i.e., pNP-acetate and methyl-ferulate, also resulted in Bi1038-CE6/CE1 and Bi1039-CE1 exhibiting higher acetyl xylan esterase activities than Bi1033-CE1 and Bi1040-FAE (Supplementary Fig. 5).

Comment 31

27. Line 182 – add ref to data (Fig S6) here.

Response

We have added reference to Fig S6 after the sentence. **Please, see line 262-264.**

“.....The GH43 (Bi1035-GH43) encoded closest to the *susC/susD* gene pair showed far superior hydrolytic activities than the other enzymes (Supplementary Fig. 6a).”

Comment 32

28. Supplementary Fig 6a – annotate locus tags with Cazy family for clarity e.g. Bi1035-GH43.

Response

We have annotated each enzyme with its CAzy family designation for clarity throughout the manuscript, including **Supplementary Fig. 6. Please, see page 82 or below.**

Supplementary Fig 6. Hydrolytic activities of the glycoside hydrolases encoded in the EGE PUL of *B. intestinalis*. **a** The reducing ends released from xylan polysaccharide

substrates by the putative glycoside hydrolases in the *B. intestinalis* EGE PUL. **b** β -xylosidase activity of Bi1042-GH3 towards xylo-oligosaccharides. **c** Arabinoxylan polysaccharide-dependent α -arabinofuranosidase activity of Bi1043-GH43. The histogram shows the results of three independent measurements. Data are shown as the mean \pm standard deviation. Rye arabinoxylan (Megazyme) has a backbone of xylose linked in β -1,4 glycosidic linkages and arabinose side chains at the O-2 or O-3 positions or both. The ratio of arabinose to xylose is 40:60. Glucuronoxylan (Sigma) has a backbone of xylose linked in β -1,4 glycosidic linkages and glucuronate side chains at the O-2 or O-3 positions or both, which are commonly methylated at position 4. Wheat arabinoxylan (Megazyme) has a backbone of xylose linked in β -1,4 glycosidic linkages and arabinose side chains at the O-2 or O-3 positions or both. The ratio of arabinose to xylose is 38:62. X1: xylose, X2: xylobiose, X3: xylotriose, X4: xylotetraose, AXX: a trisaccharide of arabinose and two xylose residues, XAXX: a tetra-saccharide of xylose, arabinose and two xylose residues, and InWAX: insoluble wheat arabinoxylan (see Supplementary Fig. 1b). Statistical analysis was performed using analysis of variance with Tukey's test for mean separation. * $p < 0.05$, ** $p < 0.001$, *** $p < 0.0001$.

Comment 33

29. Supplementary Fig 6. Add some description of the structures of the different xylans to the legend. Also need more detail about the assays used here and the stats.

Response

We have provided descriptions for the structures of the different xylans in the legend for Supplementary Fig. 6 and also provided information on the statistics. **Please, See line 1496-1507 (page 83)** or as shown in the legend above.

Comment 34

30. Fig 3f and g. Very hard to follow with just the locus tags. You should colour code enzymes into classes (e.g. exo acting GH/ endo acting GH/esterase/FA esterase) to make it easier to understand.

Response

We thank the reviewer for the suggestion. **Please, see Page 60.** We have colour-coded the enzymes into endo-acting GH, exo-acting GH, alpha-arabinofuranosidase, Ferulic acid esterase, New ferulic acid esterase, and acetyl xylan esterase.

Comment 35

31. Lines 204-205. Define the robust activity of 1040 here for clarity.

Response

Please, see line 323-326: We have re-structured the sentence to make it clearer, as follows:

“Due to the extensive ferulic acid linkage-cleaving activity of Bi1040-FAE, we had interest in determining the three-dimensional structure of this esterase to gain insight into its enzymatic versatility.”

Comment 36

32. Line 206. Expression of the ‘recombinant’ Bi1040 was low (refer to Fig 3c in text).

Response

Please, see line 326-327. We make reference to the Fig. 3c.

However, expression of Bi1040-FAE was low (Fig. 3c), making crystallization a challenge.

Comment 37

33. Supplementary Fig 8d – why only test with arabino-oligos and WAX? Did you test on xylo-oligosaccharides?.

Response.

Please, see page 86. We have tested the enzyme for hydrolysis of xylo-oligosaccharides, and the enzyme did not show any activity (**Supplementary Fig. 8e**).

Comment 38

34. Line 216. What is structure of sugar beet pulp? Describe here or in legend to S11 and also explain why this was first tested here and not earlier i.e. with the *B. intestinalis* enzymes?

Response

We have provided information on the structural difference between arabinoxylans and sugar beet pulp in the text (**line 341-351, pages 14-15**) as follows:

“.....To further explore the versatility of the characterized ferulic acid esterases in the present study, we tested the enzymes for their capacity to release ferulic acid from sugar beet pulp (Supplementary Fig. 11), a natural substrate known to contain cellulose, hemicellulose and pectin, with the ferulic acid in more complex linkages in the pectin. Unlike the arabinoxylans, which generally have the ferulic acid esterified to the C-5 position of the arabinose in the polysaccharides (Supplementary Fig. 1b), in sugar beet pulp, the ferulic acid is linked to the core α -1,5-arabinan chains and the galactopyranosyl residues of the core β -1,4-linked type I galactan chains^{35,36}. Only Bi1040-FAE and BeGH43/FAE released ferulic acid from sugar beet pulp, further demonstrating the versatility of this new ferulic acid esterase.”

Please, also refer to the legend for **Supplementary fig. 11 (Line 1541-1549) or page 93.**

“Supplementary Fig. 11. Cleavage of ferulic acid from a natural substrate (sugar beet pulp). In substrates such as wheat bran, the ferulic acid is esterified to the C-5 position of arabinose in the arabinoxylan, a β -1,4-D-xylan to which α -L-arabinofuranosyl residues attached at position 2 or 3. By contrast, the feruloyl groups in sugar beet pulp are linked to the arabinofuranosyl residues of the main core of α -1,5 linked arabinan chains and to the galactopyranosyl residues of the main core of the β -1,4-linked type I galactan chains. The ferulic acid linkages in sugar beet pulp is therefore more complex than in wheat bran or wheat arabinoxylan. Here, we assessed the versatility of the new ferulic acid esterase for the capacity to release ferulic acid from the more complex substrate sugar beet pulp. The reaction was carried out by incubating 50 nM of Bi1040-FAE or BeGH43/FAE with 0.5% of a naturally occurring substrate, sugar beet pulp, for 2 hours at 37 °C in a buffer of pH 6.5. The end products of hydrolysis were analyzed by C18-HPLC. **a** ferulic acid release by Bi1040-FAE. **b** Ferulic acid release by BeGH43/FAE. Fa: ferulic acid. SBP: sugar beet pulp

Comment 39

35. Line 240. Describe reaction tannase carries out to clarify differences in specificities claimed.

Response

Please, refer to line 373 – 386.

“The FAE domain of BeGH43/FAE revealed structural similarity to a *Lactobacillus plantarum* tannase and an overlay of the S163A catalytically inactive mutant of the *L. plantarum* tannase complexed with gallic acid (PDB 4JUI, 2.0Å RMSD for 246 C α pairs) allowed the identification of the putative residues in BeGH43/FAE that discriminate substrate (Fig. 5a,b). Tannases are a family of esterases that target the galloyl ester bond in hydrolysable tannins to release gallic acid, and the *L. plantarum* enzyme was reported to display an α/β structure with a catalytic triad constituted of Ser/His/Asp³⁹. Despite differences in the specificities of these enzymes, there is considerable conservation of the active site architecture, including placement of the catalytic nucleophile (BeGH43/FAE-S634 and *L. plantarum* tannase S163A inactive mutant), the hydrophobic pocket created by Val744/Val745 (BeGH43/FAE) underneath of the phenolic substrate, and W563 (BeGH43/FAE) towards the exterior of the binding site (Fig. 5b). These are conserved as Ile206 and Y78 in the tannase structure.”

Comment 40

36. Line 242-245. Not clear which residues are Be and which tannase from the text. Make this clearer. Also line 242 - what is S163A? Is it the inactive mutant of the tannase? If so this should be clarified.

Response

Please, accept our apologies. We have clarified the information here. **Please, refer to line 380-386.**

“Despite differences in the specificities of these enzymes, there is considerable conservation of the active site architecture, including placement of the catalytic nucleophile (BeGH43/FAE-S634 and L. plantarum tannase S163A inactive mutant), the hydrophobic pocket created by Val744/Val745 (BeGH43/FAE) underneath of the phenolic substrate, and W563 (BeGH43/FAE) towards the exterior of the binding site (Fig. 5b). These are conserved as Ile206 and Y78 in the tannase structure.”

Comment 41

37. Line 254-255. Label on Fig 4 where the loop difference is between these enzymes. Also – is it near active site? Any speculation on why this region is different?

Response

This is a great question and in truth, we cannot tell the significance of this region from our crystal structures without substrate. We have included an additional panel as Figure 5d that indicates where this loop is with respect to the active site, highlighted from the overlay of ferulic acid in the active site of the *C. thermocellum* xylanase Z structure (PDB 1JT2). **(Please, see page 64)**

We have clarified this in the text **(Please, see line 396 – 404):**

“The most obvious structural difference between the two enzymes is the loop from residues 221-233 in Bi1039-CE1 with the more extended α -helix in loop spanning residues 220-240 in Bi1033-CE1 (Fig 5c, red box). In the monomeric structure of Bi1033, there is a chain break from 223-237 that could not be modeled, suggesting some flexibility. The helix in Bi1033-CE1 (residues 231-237) is proximal to the predicted active site, perhaps restricting access, whereas this loop is smaller and appears to leave the active site more open in Bi1039-CE1 (Fig 5d). However, the precise contribution of this loop to activity and substrate selectivity is unknown.”

Comment 42

38. Line 307 refer to Fig S15a after ‘acetate.’

Response

Please, see line 573 -576. We refer to the Fig, which is now **Supplementary Fig. 20a**.

“After 24 hours of incubation, ferulic acid amounts about doubled and tripled in the WB medium and the DWB medium, respectively (Supplementary Fig. 19a), with concomitant production of succinate, propionate and acetate (Supplementary Fig. 20a).”

Comment 43

39. Line 333. HIEC-6 cells are from small intestine. This is fine but should be stated as section heading states ‘colonic cells’.

Response

(Please, see line 609-631)

Based on the recommendation of the Reviewer 1, we have removed the experiments on the HIEC cells. We have included more experiments based on the recommended dendritic cells (Please, see line 623-646 and Fig. 8, page 70)

Comment 44

40. Line 346. Sentence ending ‘...either spent medium.’ Is confusing – change to ‘...spent medium from either DWB or DW grown cells.’

Response

Please, see line 609-631. The section has been deleted, including the confusing expression. The section has been replaced with experiments using dendritic cells. Please, refer to line 623-646.

Comment 45

41. I assume if Bi1040 is a new esterase family it will be submitted to Cazy.org?

Response

This is a very good suggestion. I have contacted Dr. Bernard Henrissat and his colleagues at CAZY and they have informed me that they are struggling to reliably assign families to esterases in general, and therefore they are planning to discontinue assigning new families to this group of polypeptides. They, however, indicated that they look forward to reading this manuscript when it is published. Hopefully, this issue will be resolved in the future, and this enzyme and its relatives can be given a family.

Comment 46

42. Line 376-379 should cite Rogowski et al 2015 as the ability of Bacteroides spp. to sense the complexity of the xylan present was shown here (glucuronoxylan vs arabinoxylan).

Response

Please, see line 680 - 686.

The important work of Rogowski et al is cited here as follows:

Consistent with our observation, earlier work by Rogowski and co-workers also demonstrated that Bacteroides spp are able to sense the complexity between glucuronoxylan and arabinoxylan²⁷. These findings collectively suggest that the Bacteroidetes PULs are finely tuned to sense complexity in different polysaccharides, a regulatory strategy that likely enhances energy conservation by eliminating inefficiencies in PUL expression.

Response to Queries of Reviewer #3:

Comment 1

The manuscript by Pereria and colleagues describes the functional dissection of a complex polysaccharide utilization locus involved in complex xylan degradation.

There have been a lot of similar papers in recent times - analysing what enzymes are activated for which substrate. It can be quite hard to be novel. But I found this paper enjoyable to read and informative. This is an unusual and novel system in which complexity is recognised and a panel of side-chain removing enzymes up-regulated.

As far as this referee is concerned, the paper is very close to publication-ready. It is well referenced, well written and technically well performed. I was especially asked to comment on structural aspects. The structural biology appears to have been carried out to a very high standard and is reported to expected current norm. Could not the mystery density be modelled as PEG? Partially distorted PEG fragments are very common in enzyme active sites and that was (in contrast to what is written?) a component of the crystallisation. If it looks like lysine and makes H bonds to –OH groups, then PEG might fit the bill better than an alkyl chain?

Response

We thank the reviewer for carefully reading our manuscript and also for the kind comments and the suggestion to try PEG. Indeed, we modeled in several potential small molecules in this space; however, none of these refined well, and left some aspect of the density unsatisfied. In order for the readers to visualize this density better, we are including two views of this as **Supplemental Figure panels 13a and b**. Please, see page 95.

We initially thought in some way that this density was ferulic acid, and that perhaps this substrate was added to the wrong protein and ended up unproductively wedged in the active site, but this did not refine well, as stated in the text. Therefore, to avoid inaccurate interpretations of the structure both in this paper and from individuals that download these coordinates, we have decided to keep this density unmodelled.

Comment 2

The precision in Supplemental table 10 displays no common sense (and might lead a reader to think that the authors didn't really understand what these numbers are or how they are generated?)

B values to 1/100th of an Angstrom squared (are the authors really claiming they can work at that level of precision)

R values to the fourth decimal place.

Angles to 1/100th of an Angstrom precision.

Response

Please, see page 137-138. We thank the reviewer for catching this embarrassing oversight. The final statistics were generated using the Table One subroutine of Phenix, which by default reports these statistics to many (in)significant figures. We have corrected this mistake.

Comment 3

I am not a fan of chemical structures drawn without their stereo-chemical shading. I would prefer them all with shading and perhaps following the nature group chemdraw template (available at Nat Chem Biol, may need to be modified somewhat for sugars where hydroxyls are close!). But that is up to the authors and editors to decide if it is worth it.

Response

We thank the reviewer for the suggestion. We have made the changes, **please see Supplementary Fig. 1, 2, 3, 8, and 10 (pages 72, 74, 76, 88, and 90, respectively).**

Comment 4

This is not an enzymology paper per se, and it makes no difference to any conclusion - so I shan't insist on this change. But enzymologists would much prefer k_{cat}/K_m pH profiles, which can at least be somewhat interpreted in terms of mechanism; which an elementary % activity vs pH cannot.

Response

We agree with the reviewer; however, the nature of the substrate does not lend itself easily to do such assays, since the substrate is so complex that it is difficult to clearly determine catalytic rounds. We sometimes use the reducing ends assay as the measure

to do the enzyme kinetics, and this is frowned upon by serious enzymologists, so in order not to create this situation, we have focused on demonstrating the enzymatic activity of the members (or enzymes) of the PUL and how they contribute to depolymerize the complex polysaccharide to simple sugars that can be fermented by the microbe. Figure 4 (**page 62**) provides insights into the linkages cleaved by each enzyme and how they cooperate to yield large end-products that when analyzed show release of the unit sugars, i.e., xylose and arabinose. The unit sugars should then be fermented by the bacterium to yield short-chain fatty acids, ATP, and cellular building blocks.

Comment 5

Line 212. I'm not a fan of the grammatical construction "which rather exhibited.." maybe that is common in USA? "instead" sounds so much nicer.

Response

Thank you. **Please, see line 335-338.** We have made the correction.

“Arabino-oligosaccharides and xylo-oligosaccharides were not cleaved by BeGH43/Hyp (Supplementary Fig. 8d,e), and the enzyme instead exhibited arabinofuranosidase activity by cleaving arabinose side-chains from InWAX (Supplementary Fig. 8d).”

REVIEWERS' COMMENTS

Reviewer #1 (Remarks to the Author):

Pereira et al. have greatly improved their study of arabinoxylan metabolism by *Bacteroides*. The manuscript could be even stronger following some edits for clarity and to avoid overstating the current results.

The weakest part remains the immunology section, which comes after a rather large body of work that probably would have been enough for publication without getting into the host effects. In the absence of a lot more data, I'd recommend removing the immunological effects from the title and abstract and adding the caveats of the current work and plan for future experiments to the discussion. Caveats include small number of mice/group, lack of replicate experiments, reliance on an in vitro disease model, lack of isogenic bacterial strains, potential contribution of other metabolites in the spent media, etc. This is especially true given the conflicting data shown here suggesting both immune activation and suppression depending on the context or model used. Furthermore, *Bacteroides* can influence immune function through many other mechanisms (e.g., PSA, SCFAs) and how much this particular activity matters relative to these other effects isn't addressed here.

Minor points:

Line 37: What do you mean by "major phyla"?

Lines 41, 110, 140: The abstract, intro, and results all mention the fact that the expected PUL was not upregulated, which seems unnecessarily confusing and not particularly surprising to me. How often do differential expression and a priori predictions match? Why is there a need to highlight this conflicting data here?

Lines 48-50: Please be more specific about the immune effects or remove this section.

Lines 62, 117: The intro repeatedly mention evolution, which is outside the scope of this study and not well supported by the provided references.

Line 90: I'm not sure the keystone species concept is meaningful at the phylum level. That's like saying Chordata matter for the ecology of a grassland.

Line 102: The 10% kcal value is based upon a specific diet and a very old citation.

Line 114: Different relative to what? Other genera, species, phyla?

Line 137: Overexpression refers to genetic manipulation or disease. Natural regulation is called up/down-regulation, which is more fitting for the current data.

Line 166: Not measuring transcription per se, but rather transcript levels. RNA stability can also be regulated.

Line 182: Differential expression does not imply activity.

Figure 1. ANOVA with multiple hypothesis correction would be more appropriate.

Reviewer #2 (Remarks to the Author):

In general in the revised MS Perriera et al have done a very good job of addressing most of my comments. However I think there is a key misinterpretation of the new data shown in Fig 8 that must be addressed.

The authors have carried out the control experiments as requested by myself (Comment 1) and Rev1, but to me they have not interpreted the data correctly. In the revised MS and response to myself and Rev1 the authors claim that the increased levels of TGFbeta and IL-10 and reduced TNFalpha in Bi+InWAX spent media compared to InWAX media alone treated cells shows that FA released by Bi is driving the effect. However the key experiment to compare is spent media from Bi on sWAX vs Bi on InWAX i.e. the only difference should be the release of FA in InWAX. Comparing to InWAX alone means that other products of bacterial fermentation (e.g. SCFAs) are not considered. In fact ideally ALL conditions (+LPS) should be compared to each other to analyse the data properly and not just Bi + InWAX vs InWAX alone as has been done here. From the data presented in the new Fig 8 it looks as if it might only be IL-10 that is significantly different in Bi InWAX vs all other conditions. This is fine as IL-10 is an important anti-inflammatory cytokine and so the authors can still comment on this and say their data shows a likely anti-inflammatory effect of microbiota driven release of FA, but this does not extend to TGFbeta and TNFalpha.

Reviewer #3 (Remarks to the Author):

Well, I have to say in all my career I have never seen such a detailed, well argued, and experimentally-supported response to reviewers.

That said. Something is going wrong in science generally if we are seeing 150 page manuscripts with 30-40 SI figures

Compared to the other referees, my comments were minor and they have mainly been addressed to my satisfaction. I don't think I have ever seen xylan (and others) drawn with the "away" shaded bonds. Ever. More normal just to shade, correctly, those coming towards you. It's correct as drawn, just unusual to the point of being unique.

I was also asked to comment on the molecular dynamics.

As far as I can tell these have been used to address a relatively minor comment about broad substrate specificity of some esterases compared to others. I shall start by saying that broad specificity esterases are surely the norm so I don't see the surprise here.

I think the study of binding pocket size is sound (although without knowing the size of what needs to be accommodated, I am not sure the comparative volumes are much use...and the changes seem quite small; comment please?). If I understood correctly (and this is quite a verbose section of the revision... the authors might consider shortening and focussing on key issues)

(a) there are different volumes which correlate with substrate accommodation

(b) simulations with substrates show that steric factors prevent larger substrates getting close to the nucleophilic serine in some enzymes

(c) in some enzymes the catalytic triad falls apart so the supporting Glu/His cannot perform their normal role.

(a) and (b) seem fine and sensible, I have to say I didn't understand the discussion of catalytic triad stability, nor do I have any intuition whether those observations would translate to the observed kinetics. That 64% are fine and viable triads over ? 2 microseconds ? I simply have no insight into whether that has any impact on kinetics measured on a seconds/minutes timescale.

64% may well be plenty ? 1% might be ? I cant tell.

in the abstract, as it is so important
We earlier discovered ? is poor English.
We discovered previously ?
In previous work, we discovered ?

Reviewer #4 (Remarks to the Author):

In this manuscript the authors present a comprehensive analysis of the degradation of arabinoxylans by human colonic bacterioidetes. I found the work very interesting, well written and exhaustive in terms of the different methodologies used to tackle the many different aspects of the problem at hand.

My report here pertains exclusively to the MD work as it is my field of expertise and will leave other reviewers to comment on the experimental work. I found the computational study well executed from both the methodological point of view and the data analysis presented via statistical methods rather than based on time evolutions. the results were also presented very clearly in both text and figures and in my opinion clearly support the conclusions. Based on my reading, in regard to the additional MD work in this resubmission, I support the publication of this work as it is.

REVIEWERS' COMMENTS

Reviewer #1 (Remarks to the Author):

Pereira et al. have greatly improved their study of arabinoxylan metabolism by *Bacteroides*. The manuscript could be even stronger following some edits for clarity and to avoid overstating the current results.

The weakest part remains the immunology section, which comes after a rather large body of work that probably would have been enough for publication without getting into the host effects. In the absence of a lot more data, I'd recommend removing the immunological effects from the title and abstract and adding the caveats of the current work and plan for future experiments to the discussion. Caveats include small number of mice/group, lack of replicate experiments, reliance on an in vitro disease model, lack of isogenic bacterial strains, potential contribution of other metabolites in the spent media, etc. This is especially true given the conflicting data shown here suggesting both immune activation and suppression depending on the context or model used. Furthermore, *Bacteroides* can influence immune function through many other mechanisms (e.g., PSA, SCFAs) and how much this particular activity matters relative to these other effects isn't addressed here.

Response: We appreciate the reviewer's diligence and supportive comments on the contents of the manuscript aside from the immunology section, which both reviewer 1 and 2 have expressed major concerns. Based on the suggestion of the senior editor, we have removed all the immunology work from the manuscript.

Minor points:

Line 37: What do you mean by "major phyla"?

Response: This statement appeared in the original abstract. The very nice and clear abstract suggested by the editor does not contain this expression.

Lines 41, 110, 140: The abstract, intro, and results all mention the fact that the expected PUL was not upregulated, which seems unnecessarily confusing and not particularly surprising to me. How often do differential expression and a priori predictions match? Why is there a need to highlight this conflicting data here?

Response: We appreciate the reviewers concerns and have made some modifications in regards to this statement in the manuscript. For line 41, which is in the abstract, the abstract itself has been extensively modified, and therefore this statement does not appear in the abstract anymore. We have deleted the sentence on line 140 and replaced it with the following sentence "In the present study, we discover that *B. intestinalis* and related colonic bacteria use a putative EGE PUL to extensively degrade complex arabinoxylans" (New line number 237-239).

We would, however, like to keep the sentence on line 110 as is, because we built our hypothesis for this work on that observation, based on our understanding of the functions of the PULs in the *Bacteroidetes*. We hope that the reviewer finds these revisions agreeable, especially since this statement or idea is no more being over-emphasized in the manuscript.

Lines 48-50: Please be more specific about the immune effects or remove this section.

Response: The immunology part has been removed, based on instructions by the editor.

Lines 62, 117: The intro repeatedly mention evolution, which is outside the scope of this study and not well supported by the provided references.

Response: We have removed the word “evolved” from the sentence on line 62, and instead re-structured the sentence using the expression “able to acquire” (**New line number 141**): “The human gut microbiome reflects this concept, as its members are able to acquire carbon and energy from host gastrointestinal tract (GIT)-derived glycans and the dietary components undegradable by the host”.

Similarly we rephrase the statement on previous line 117 using the expression “are able” (**New line number 202**). The sentence now reads: “In this study, we culture diverse members of the human colonic Bacteroidetes on arabinoxylans of different complexities and demonstrate that these bacteria are able to distinguish simple from complex arabinoxylans by deploying different PULs for their degradation.”

Line 90: I’m not sure the keystone species concept is meaningful at the phylum level. That’s like saying Chordata matter for the ecology of a grassland.

Response: We have removed the word “keystone” from the sentence on line 90. The sentence now reads: “In orchestrating the initial enzymatic attack on polysaccharides in the GIT, members of the Bacteroidetes and the Firmicutes act as primary polysaccharide degraders, making nutrients available for their own metabolic processes and also to cross-feed the colonic microbiome members that lack the requisite hydrolytic enzymes.” (**New line number 169-173**)

Line 102: The 10% kcal value is based upon a specific diet and a very old citation.

Response: We have removed this statement and also the reference. (**See New line number 181. Sentence is deleted**).

Line 114: Different relative to what? Other genera, species, phyla?

Response: We have modified the sentence to make it clearer to the reader as follows: “This observation led us to hypothesize that depending on the nature of the arabinoxylan, i.e., simple or complex, the Bacteroidetes deploy different PULs for degradation of the polysaccharide”. (**New line number 197-199**)

Line 137: Overexpression refers to genetic manipulation or disease. Natural regulation is called up/down-regulation, which is more fitting for the current data.

Response: We appreciate the comment and have changed the sentence to “....., we discovered that the bacterium **upregulates** a number of genes, mostly in two PULs containing either one or two carbohydrate esterase (CE) genes” (**New line number 236**)

Line 166: Not measuring transcription per se, but rather transcript levels. RNA stability can also be regulated.

Response: Thank you for the comment. We re-state the sentence to show that we are measuring the transcript level. The sentence now reads “Growth on the complex substrates led to ~25 – 100-fold increases in the **transcript level** of the *susC/susD* gene pair in the EGE PUL compared to growth on the constituent sugars”. (New line number 275-277)

Line 182: Differential expression does not imply activity.

Response: We have re-stated the sentence to indicate the data suggests the EGE PUL is important. The sentence now reads “Other GH and CE genes on the genomes of the three bacteria were also upregulated, although at a far lower level (Supplementary Data 1), suggesting that the EGE PUL is important for complex arabinoxylan degradation”. (New line number 290-294).

Figure 1. ANOVA with multiple hypothesis correction would be more appropriate.

Response: Thank you for the suggestion. We have done the analysis again with one-way Analysis of Variance with posttest comparisons based on Tukey’s test. We have also modified the information referencing this data to reflect the results. (New line number 281-282)

We have also modified the figure legend for Fig. 1b-g to reflect the new analysis.

Results now reads “In contrast, upregulation of transcription of the *susC/susD* gene pair in the EGE PUL was barely observed with growth of the three different bacteria on the simple or soluble arabinoxylan (sWAX). Furthermore, the differences in the expression of the EGE PUL between the simple and complex arabinoxylans were, in general, shown to be statistically significant (Fig. 1b,c,d,e,f,g). (New line number 277-282).

Reviewer #2 (Remarks to the Author):

In general in the revised MS Perriera et al have done a very good job of addressing most of my comments. However I think there is a key misinterpretation of the new data shown in Fig 8 that must be addressed.

The authors have carried out the control experiments as requested by myself (Comment 1) and Rev1, but to me they have not interpreted the data correctly. In the revised MS and response to myself and Rev1 the authors claim that the increased levels of TGFbeta and IL-10 and reduced TNFalpha in Bi+InWAX spent media compared to InWAX media alone treated cells shows that FA released by Bi is driving the effect. However the key experiment to compare is spent media from Bi on sWAX vs Bi on InWAX i.e. the only difference should be the release of FA in InWAX. Comparing to InWAX alone means that other products of bacterial fermentation (e.g. SCFAs) are not considered. In fact ideally ALL conditions (+LPS) should be compared to each other to analyse the data properly and not just Bi + InWAX vs InWAX alone as has been done here. From the data presented in the new Fig 8 it looks as if it might only be IL-10 that is significantly different in Bi InWAX vs all other conditions. This is fine as IL-10 is an important anti-inflammatory cytokine and so the authors can still comment on this and say

their data shows a likely anti-inflammatory effect of microbiota driven release of FA, but this does not extend to TGFbeta and TNFalpha.

Response: We thank the reviewer for all his suggestions, which have made the manuscript stronger and clearer to read. Based on the comments of reviewer 1 and reviewer 2, the editor has instructed us to remove all the immunology work from the manuscript. We clearly see the reviewer's point, and we will address this as we work to improve the immunology sections for publication elsewhere.

Reviewer #3 (Remarks to the Author):

Well, I have to say in all my career I have never seen such a detailed, well argued, and experimentally-supported response to reviewers.

That said. Something is going wrong in science generally if we are seeing 150 page manuscripts with 30-40 SI figures

Compared to the other referees, my comments were minor and they have mainly been addressed to my satisfaction. I don't think I have ever seen xylan (and others) drawn with the "away" shaded bonds. Ever. More normal just to shade, correctly, those coming towards you. It's correct as drawn, just unusual to the point of being unique.

Response: We appreciate the reviewer's suggestions and encouragement. By following the reviewer's suggestions throughout the review process, we think that the manuscript is now stronger, clearer and more easily accessible to the readership of Nature Communications.

We have also ensured that we have followed the reviewer's suggestions in regard to shading the bonds and modified the figures in Supplementary Fig.1, 2, 3, 9 and 10, i.e., the bonds towards the reader are the only ones shaded.

I was also asked to comment on the molecular dynamics.

As far as I can tell these have been used to address a relatively minor comment about broad substrate specificity of some esterases compared to others. I shall start by saying that broad specificity esterases are surely the norm so I don't see the surprise here.

Response: We appreciate the reviewers comment and we will keep this in mind.

I think the study of binding pocket size is sound (although without knowing the size of what needs to be accommodated, I am not sure the comparative volumes are much use...and the changes seem quite small; comment please?). If I understood correctly (and this is quite a verbose section of the revision... the authors might consider shortening and focussing on key issues)

(a) there are different volumes which correlate with substrate accommodation

(b) simulations with substrates show that steric factors prevent larger substrates getting close to the nucleophilic serine in some enzymes

(c) in some enzymes the catalytic triad falls apart so the supporting Glu/His cannot perform their normal role.

Response: We appreciate the reviewer's comments. Since many readers may not be very familiar with Molecular Dynamics Simulations (MDS), we thought we needed to provide more explanation to help the reader, and this unfortunately lengthened this section.

Also, the comparative pocket volume analysis is supported by the substrate binding simulation. Binding volume calculations show that mean binding pocket volume difference between BeGH43/FAE and Bi1039-CE1 is $\sim 16 \text{ \AA}^3$. Due to the larger binding volume, substrate FA can access the catalytic triad residues (HIS(Base) /SER(Nucleophile)) of the BeGH43/FAE (carbonyl carbon of FA binds closer to SER). For the case of Bi1039-CE1, it is hard for the FA to access catalytic triad due to the bulky residues (PHE184 and TYR295) which constrict the pocket size (Old Supplementary Fig. 17, and now **Fig. 8**).

(a) and (b) seem fine and sensible, I have to say I didn't understand the discussion of catalytic triad stability, nor do I have any intuition whether those observations would translate to the observed kinetics. That 64% are fine and viable triads over ~ 2 microseconds? I simply have no insight into whether that has any impact on kinetics measured on a seconds/minutes timescale. 64% may well be plenty? 1% might be? I can't tell.

Response: We appreciate the reviewer's query and agree with the reviewer that a direct estimate of the effect of catalytic triad stability on the observed kinetics would have strengthened the argument presented in the manuscript. However, the impact of the catalytic triad stability on the actual kinetics of the catalytic process cannot be assessed by classical MD simulation and is beyond the scope of the current study. The hypothesis that the enzyme kinetics is correlated with the population of the catalytically-competent conformation has been assessed for a variety of enzymes (Ma and Nussinov, *Current Opinion in Chemical Biology*, 14, 5, 652-659, 2010). Therefore, the stability analysis of catalytic triad could potentially explain the reason why Bi1033 protein shows more catalytic activity compared to Bi1039 for smaller substrate. Stable hydrogen bond between base and acid makes the base more electronegative and increase its propensity to deprotonate the nucleophile. Therefore, we argued that Bi-1033 has more catalytic activity compared to Bi-1039. Furthermore, we found that the percentage of the simulation snapshots with stable hydrogen bond between catalytic triad residues does not change after $1 \mu\text{s}$. Therefore, we performed $2 \mu\text{s}$ long simulations of each protein to obtain reliable estimate of the relative stability differences between Bi-1033 and Bi1039.

in the abstract, as it is so important

We earlier discovered? is poor English.

We discovered previously?

In previous work, we discovered?

Response: We appreciate the reviewer's comments, especially as it is related to the abstract. The abstract has been extensively modified, and we hope that the current version is acceptable.

Reviewer #4 (Remarks to the Author):

In this manuscript the authors present a comprehensive analysis of the degradation of arabinoxylans by human colonic bacterioidetes. I found the work very interesting, well written and exhaustive in terms of the different methodologies used to tackle the many different aspects of the problem at hand.

My report here pertains exclusively to the MD work as it is my field of expertise and will leave other reviewers to comment on the experimental work. I found the computational study well executed from both the methodological point of view and the data analysis presented via statistical methods rather than based on time evolutions. The results were also presented very clearly in both text and figures and in my opinion clearly support the conclusions. Based on my reading, in regard to the additional MD work in this resubmission, I support the publication of this work as it is.

Response: We appreciate the reviewer's very supportive comments of the entire work and especially the area directly related to his his/her field.